# Ordered Subgraph Aggregation Networks

**Chendi Qian**[*]
Department of Computer Science
TU Munich

**Gaurav Rattan**[*]
Department of Computer Science
RWTH Aachen University

**Floris Geerts**
Department of Computer Science
University of Antwerp

**Christopher Morris**
Department of Computer Science
RWTH Aachen University

**Mathias Niepert**
Department of Computer Science
University of Stuttgart

## Abstract

Numerous subgraph-enhanced graph neural networks (GNNs) have emerged recently, provably boosting the expressive power of standard (message-passing) GNNs. However, there is a limited understanding of how these approaches relate to each other and to the Weisfeiler–Leman hierarchy. Moreover, current approaches either use all subgraphs of a given size, sample them uniformly at random, or use hand-crafted heuristics instead of learning to select subgraphs in a data-driven manner. Here, we offer a unified way to study such architectures by introducing a theoretical framework and extending the known expressivity results of subgraph-enhanced GNNs. Concretely, we show that increasing subgraph size always increases the expressive power and develop a better understanding of their limitations by relating them to the established $k$-WL hierarchy. In addition, we explore different approaches for learning to sample subgraphs using recent methods for backpropagating through complex discrete probability distributions. Empirically, we study the predictive performance of different subgraph-enhanced GNNs, showing that our data-driven architectures increase prediction accuracy on standard benchmark datasets compared to non-data-driven subgraph-enhanced graph neural networks while reducing computation time.

## 1  Introduction

Graph-structured data are ubiquitous across application domains ranging from chemo- and bioinformatics [Barabasi and Oltvai, 2004, Jumper et al., 2021, Stokes et al., 2020] to image [Simonovsky and Komodakis, 2017] and social-network analysis [Easley and Kleinberg, 2010]. Numerous approaches for graph–based machine learning have been proposed, most notably those based on *graph kernels* [Borgwardt et al., 2020, Kriege et al., 2020] or using *graph neural networks* (GNNs) [Chami et al., 2020, Gilmer et al., 2017, Morris et al., 2021]. Here, graph kernels based on the 1-*dimensional Weisfeiler–Leman algorithm* (1-WL) [Weisfeiler and Leman, 1968], a simple heuristic for the graph isomorphism problem, and corresponding GNNs [Morris et al., 2019, Xu et al., 2019], have recently advanced the state-of-the-art in supervised vertex- and graph-level learning. However, the 1-WL and GNNs operate via local neighborhood aggregation, missing crucial patterns in the given data while more expressive architectures based on the $k$-*dimensional Weisfeiler–Leman algorithm* ($k$-WL) [Azizian and Lelarge, 2020, Maron et al., 2019, Morris et al., 2020b, 2021, 2022] may not scale to larger graphs.

---

[*]These authors contributed equally.

36th Conference on Neural Information Processing Systems (NeurIPS 2022).

Hence, several approaches such as Bevilacqua et al. [2021], Cotta et al. [2021], Li et al. [2020], Papp et al. [2021], Thiede et al. [2021], You et al. [2021] and Zhao et al. [2021] have enhanced the expressive power of GNNs, by removing, extracting, or marking (small) subgraphs, so as to allow GNNs to leverage more structural patterns within the given graph, essentially breaking symmetries induced by the GNNs' local aggregation function. We henceforth refer to these approaches as *subgraph-enhanced GNNs*.

**Present work** First, to bring some order to the multitude of recently proposed subgraph-enhanced GNNs, we introduce a theoretical framework to study these approaches' expressive power in a unified setting. Concretely,

- we introduce $k$-*ordered subgraph aggregation networks* ($k$-OSANs) and show that they capture most of the recently proposed subgraph-enhanced GNNs.
- We show that any $k$-OSAN is upper bounded by $(k + 1)$-WL in terms of expressive power and show that $k$-OSANs and $k$-WL are incomparable in terms of expressive power. Consequently, we obtain new upper bounds on the expressive power of recently proposed subgraph-enhanced GNNs.
- We show that increasing $k$, i.e., using larger subgraphs, always leads to an increase in expressive power, effectively showing that $k$-OSANs form a hierarchy.

Second, most approaches consider all subgraphs or use hand-crafted heuristics to select them, e.g., by deleting vertices or edges. Instead, we leverage recent progress in back-propagating through discrete structures using perturbation-based differentiation [Domke, 2010, Niepert et al., 2021] to sample subgraphs in a *data-driven* fashion, automatically adapting to the given data distribution. Concretely,

- we explore different strategies to sample subgraphs leveraging the I-MLE framework [Niepert et al., 2021], resulting in the data-driven $k$-OSAN architecture.
- We show, empirically, that data-driven $k$-OSANs increase prediction accuracy on standard benchmark datasets compared to non-data-driven subgraph-enhanced GNNs while vastly reducing computation time.

## 1.1 Related work

In the following, we discuss related work relevant to the present work; see Appendix A for an extended discussion.

**GNNs** Recently, GNNs [Gilmer et al., 2017, Scarselli et al., 2009] emerged as the most prominent graph representation learning architecture. Notable instances of this architecture include, e.g., Duvenaud et al. [2015], Hamilton et al. [2017] and Veličković et al. [2018], which can be subsumed under the message-passing framework introduced in Gilmer et al. [2017]. In parallel, approaches based on spectral information were introduced in, e.g., Defferrard et al. [2016], Bruna et al. [2014], Kipf and Welling [2017] and Monti et al. [2017]—all of which descend from early work in Baskin et al. [1997], Kireev [1995], Micheli and Sestito [2005], Merkwirth and Lengauer [2005], Micheli [2009], Scarselli et al. [2009] and Sperduti and Starita [1997].

**Limits of GNNs and more expressive architectures** Recently, connections between GNNs and Weisfeiler–Leman type algorithms have been shown [Azizian and Lelarge, 2020, Barceló et al., 2020, Chen et al., 2019b, Geerts et al., 2020, Geerts, 2020, Geerts and Reutter, 2022, Maehara and NT, 2019, Maron et al., 2019, Morris et al., 2019, 2022, Xu et al., 2019]. Specifically, Morris et al. [2019] and Xu et al. [2019] showed that the expressive power of any possible GNN architecture is limited by the 1-WL in terms of distinguishing non-isomorphic graphs.

Recent works have extended the expressive power of GNNs, e.g., by encoding vertex identifiers [Murphy et al., 2019, Vignac et al., 2020], using random features [Abboud et al., 2020, Dasoulas et al., 2020, Sato et al., 2020], homomorphism and subgraph counts [Barceló et al., 2021, Bouritsas et al., 2020, NT and Maehara, 2020], spectral information [Balcilar et al., 2021], simplicial and cellular complexes [Bodnar et al., 2021b,a], persistent homology [Horn et al., 2021], random walks [Tönshoff et al., 2021], graph decompositions [Talak et al., 2021], or distance [Li et al., 2020] and directional information [Beaini et al., 2020]. See Morris et al. [2021] for an in-depth survey on this topic.

**Subgraph-enhanced GNNs** Most relevant to the present work are *subgraph-enhanced GNNs*. Cotta et al. [2021] and Papp et al. [2021] showed how to make GNNs more expressive by removing one or more vertices from a given graph and using standard GNN architectures to learn vectorial representations of the

resulting subgraphs. The approaches either consider all possible subgraphs or utilize random sampling to arrive at more scalable architectures. Cotta et al. [2021] showed that by removing one or two vertices, such architectures can distinguish graphs the 1-WL and 2-WL, respectively, are not able to distinguish. Extensions and refinements of the above were proposed in Bevilacqua et al. [2021], Papp and Wattenhofer [2022], Thiede et al. [2021], You et al. [2021], Zhang and Li [2021] and Zhao et al. [2021], see Papp and Wattenhofer [2022] for an overview. For example, Bevilacqua et al. [2021] generalized several ideas discussed above and proposed the ESAN framework in which each graph is represented as a multiset of its subgraphs and processed them using an equivariant architecture based on the DSS architecture [Maron et al., 2020] and GNNs. The authors proposed several simple subgraph selection policies, e.g., edge removal, ego networks, or vertex removal, and showed that the architecture surpasses the expressive power of the 1-WL. Moreover, Frasca et al. [2022] presented a novel symmetry analysis unifying a series of subgraph-enhanced GNNs, allowing them to upper-bound their expressive power and to define a systematic framework to conceive novel architectures in this family. We note here that the above works, unlike the present one, mostly do not study the approaches' expressive power beyond (folklore or non-oblivious) 2-WL and do not compare at all to the (folklore or non-oblivious) 3-WL, while our analysis works for the whole $k$-WL hierarchy.

See Appendix A for a detailed overview of recent progress in differentiating through discrete structures.

## 2 Preliminaries

As usual, for $n \geq 1$, let $[n] \coloneqq \{1, \dots, n\} \subset \mathbb{N}$. We use $\{\!\!\{\dots\}\!\!\}$ to denote multisets, i.e., the generalization of sets allowing for multiple instances of each of its elements.

A *graph* $G$ is a pair $(V(G), E(G))$ with *finite* sets of *vertices* $V(G)$ and *edges* $E(G) \subseteq \{\{u, v\} \subseteq V(G) \mid u \neq v\}$. If not otherwise stated, we set $n \coloneqq |V(G)|$. For ease of notation, we denote the edge $\{u, v\}$ in $E(G)$ by $(u, v)$ or $(v, u)$. In the case of *directed graphs*, $E \subseteq \{(u, v) \in V \times V \mid u \neq v\}$. A *labeled graph* $G$ is a triple $(V, E, l)$ with *(vertex) coloring* or *label function* $l \colon V(G) \to \mathbb{N}$. Then $l(v)$ is a *label* of $v$ for $v$ in $V(G)$. The *neighborhood* of $v$ in $G$ is denoted by $N_G(v) \coloneqq \{u \in V(G) \mid \{v, u\} \in E(G)\}$ and the *degree* of a vertex $v$ is $|N_G(v)|$. For $S \subseteq V(G)$, the graph $G[S] = (S, E_S)$ is the *subgraph induced by $S$*, where $E_S \coloneqq \{(u, v) \in E(G) \mid u, v \in S\}$.

Two graphs $G$ and $H$ are *isomorphic* and we write $G \simeq H$ if there exists a bijection $\varphi \colon V(G) \to V(H)$ that preserves the adjacency relation, i.e., $(u, v)$ is in $E(G)$ if and only if $(\varphi(u), \varphi(v))$ is in $E(H)$. Then $\varphi$ is an *isomorphism* between $G$ and $H$. Moreover, we call the equivalence classes induced by the relation $\simeq$ *isomorphism types*. In the case of labeled graphs, we additionally require that $l(v) = l(\varphi(v))$ for $v$ in $V(G)$. We further define the atomic type $\mathsf{atp} \colon V(G)^k \to \mathbb{N}$ such that $\mathsf{atp}(\mathbf{v}) = \mathsf{atp}(\mathbf{w})$ for $\mathbf{v}, \mathbf{w} \in V(G)^k$ if and only if the mapping $\varphi \colon V(G)^k \to V(G)^k$ where $v_i \mapsto w_i$ induces a partial isomorphism, i.e., we have $v_i = v_j \iff w_i = w_j$ and $(v_i, v_j) \in E(G) \iff (\varphi(v_i), \varphi(v_j)) \in E(G)$. Let $\mathbf{v}$ be a *tuple* in $V(G)^k$ for $k > 0$, then $G[\mathbf{v}]$ is the *ordered $k$-vertex subgraph* induced by the multiset of elements of $\mathbf{v}$, where the vertices are labeled with integers from $[k]$ corresponding to their positions in $\mathbf{v}$. Moreover, let $\mathsf{t}(G[\mathbf{v}]) \coloneqq \mathbf{v}$, i.e., the $k$-tuple $\mathbf{v}$ underlying the ordered $k$-vertex subgraph $G[\mathbf{v}]$. We denote the set of all ordered $k$-vertex subgraphs of a graph $G$ by $G_k$. Finally, let $\mathcal{G}$ be the set of all vertex-labeled graphs.

**The $1$-WL and the $k$-WL** The 1-WL or color refinement is a simple heuristic for the graph isomorphism problem, originally proposed by Weisfeiler and Leman [1968].[2] Intuitively, the algorithm determines if two graphs are non-isomorphic by iteratively coloring or labeling vertices. Given an initial coloring or labeling of the vertices of both graphs, e.g., their degree or application-specific information, in each iteration, two vertices with the same label get different labels if the number of identically labeled neighbors is not equal. If, after some iteration, the number of vertices annotated with a specific label is different in both graphs, the algorithm terminates and a stable coloring (partition) is obtained. We can then conclude that the two graphs are not isomorphic. It is easy to see that the algorithm cannot distinguish all non-isomorphic graphs [Cai et al., 1992]. Nonetheless, it is a powerful heuristic that can successfully test isomorphism for a broad class of graphs [Babai and Kucera, 1979].

Formally, let $G = (V, E, l)$ be a labeled graph. In each iteration, $i > 0$, the 1-WL computes a vertex coloring $C_i^1 \colon V(G) \to \mathbb{N}$, which depends on the coloring of the neighbors. That is, in iteration $i > 0$, we

---

[2]Strictly speaking, the 1-WL and color refinement are two different algorithms. That is, the 1-WL considers neighbors and non-neighbors to update the coloring, resulting in a slightly higher expressive power when distinguishing vertices in a given graph, see Grohe [2021] for details. For brevity, we consider both algorithms to be equivalent.

set

$$C_i^1(v) \coloneqq \mathsf{RELABEL}\Big(\big(C_{i-1}^1(v), \{\!\!\{C_{i-1}^1(u) \mid u \in N_G(v)\}\!\!\}\big)\Big),$$

where RELABEL injectively maps the above pair to a unique natural number, which has not been used in previous iterations. In iteration 0, the coloring $C_0^1 \coloneqq l$. To test if two graphs $G$ and $H$ are non-isomorphic, we run the above algorithm in "parallel" on both graphs. If the two graphs have a different number of vertices colored $c$ in $\mathbb{N}$ at some iteration, the 1-WL *distinguishes* the graphs as non-isomorphic. Moreover, if the number of colors between two iterations, $i$ and $(i + 1)$, does not change, i.e., the cardinalities of the images of $C_i^1$ and $C_{i+1}^1$ are equal, the algorithm terminates. For such $i$, we define the *stable coloring* $C_\infty^1(v) = C_i^1(v)$ for $v$ in $V(G)$. The stable coloring is reached after at most $\max\{|V(G)|, |V(H)|\}$ iterations [Grohe, 2017].

Due to the shortcomings of the 1-WL or color refinement in distinguishing non-isomorphic graphs, several researchers [Babai, 1979, 2016, Immerman and Lander, 1990], devised a more powerful generalization of the former, today known as the $k$-*dimensional Weisfeiler-Leman algorithm* ($k$-WL); see Appendix B for a detailed description.

**Graph Neural Networks** Intuitively, GNNs learn a vectorial representation, i.e., a $d$-dimensional vector, representing each vertex in a graph by aggregating information from neighboring vertices. Formally, let $G = (V, E, l)$ be a labeled graph with initial vertex features $\mathbf{h}_v^{(0)} \in \mathbb{R}^d$ that are *consistent* with $l$. That is, each vertex $v$ is annotated with a feature $\mathbf{h}_v^{(0)} \in \mathbb{R}^d$ such that $\mathbf{h}_u^{(0)} = \mathbf{h}_v^{(0)}$ if $l(u) = l(v)$, e.g., a one-hot encoding of the labels $l(u)$ and $l(v)$. Alternatively, $\mathbf{h}_v^{(0)}$ can be an arbitrary real-valued feature vector or attribute of the vertex $v$, e.g., physical measurements in the case of chemical molecules. A GNN architecture consists of a stack of neural network layers, i.e., a composition of parameterized functions. Similarly to 1-WL, each layer aggregates local neighborhood information, i.e., the neighbors' features, around each vertex and then passes this aggregated information on to the next layer.

Following, Gilmer et al. [2017] and Scarselli et al. [2009], in each layer, $i > 0$, we compute vertex features

$$\mathbf{h}_v^{(i+1)} \coloneqq \mathsf{UPD}^{(i+1)}\Big(\mathbf{h}_v^{(i)}, \mathsf{AGG}^{(i+1)}\big(\{\!\!\{\mathbf{h}_u^{(i)} \mid u \in N_G(v)\}\!\!\}\big)\Big) \in \mathbb{R}^d,$$

where $\mathsf{UPD}^{(i+1)}$ and $\mathsf{AGG}^{(i+1)}$ may be differentiable parameterized functions, e.g., neural networks.[3] In the case of graph-level tasks, e.g., graph classification, one uses

$$\mathbf{h}_G \coloneqq \mathsf{READOUT}\big(\{\!\!\{\mathbf{h}_v^{(T)} \mid v \in V(G)\}\!\!\}\big) \in \mathbb{R}^d, \tag{1}$$

to compute a single vectorial representation based on learned vertex features after iteration $T$. Again, READOUT may be a differentiable parameterized function. To adapt the parameters of the above three functions, they are optimized end-to-end, usually through a variant of stochastic gradient descent, e.g., [Kingma and Ba, 2015], together with the parameters of a neural network used for classification or regression.

**The Weisfeiler–Leman hierarchy and permutation-invariant function approximation** The Weisfeiler–Leman hierarchy is a purely combinatorial algorithm for testing graph isomorphism. However, the graph isomorphism function, mapping non-isomorphic graphs to different values, is the hardest to approximate permutation-invariant function. Hence, the Weisfeiler–Leman hierarchy has strong ties to GNNs' capabilities to approximate permutation-invariant or equivariant functions over graphs. For example, Morris et al. [2019], Xu et al. [2019] showed that the expressive power of any possible GNN architecture is limited by 1-WL in terms of distinguishing non-isomorphic graphs. Azizian and Lelarge [2020] refined these results by showing that if an architecture is capable of simulating $k$-WL and allows the application of universal neural networks on vertex features, it will be able to approximate any permutation-equivariant function below the expressive power of $k$-WL; see also Chen et al. [2019b]. Hence, if one shows that one architecture distinguishes more graphs than another, it follows that the corresponding GNN can approximate more functions. These results were refined in Geerts and Reutter [2022] for color refinement and taking into account the number of iterations of $k$-WL.

---

[3]Strictly speaking, Gilmer et al. [2017] consider a slightly more general setting in which vertex features are computed by $\mathbf{h}_v^{(i+1)} \coloneqq \mathsf{UPD}^{(i+1)}\Big(\mathbf{h}_v^{(i)}, \mathsf{AGG}^{(i+1)}\big(\{\!\!\{(\mathbf{h}_v^{(i)}, \mathbf{h}_u^{(i)}, l(v, u)) \mid u \in N_G(v)\}\!\!\}\big)\Big)$.

# 3 Ordered subgraph Weisfeiler–Leman and MPNNs

In the following, we introduce a variant of 1-WL, denoted $k$-*ordered subgraph WL* ($k$-OSWL). Essentially, the $k$-OSWL labels or marks ordered subgraphs and then executes 1-WL on top of the marked graphs, followed by an aggregation phase. Although unordered subgraphs are also possible, ordered ones lead to more expressive architectures and also encompass the unordered case; see Appendix F.1 for a discussion. To make the procedure permutation-invariant, we consider all possible ordered subgraphs. Based on the ideas of $k$-OSWL, we then introduce $k$-OSANs, which can be seen as a neural variant of the former, allowing us to analyze various subgraph-enhanced GNNs.

## 3.1 Ordered subgraph WL

We now describe the algorithm formally. Let $G$ be a graph, and let $\mathbf{g} \in G_k$ be an ordered $k$-vertex subgraph. Then $k$-OSWL computes a vertex coloring, similarly to 1-WL, with the main distinction that it can use structural graph information related to the ordered subgraph $G[(v, \mathsf{t}(\mathbf{g}))]$, where $v$ is a vertex in $V(G)$.

More precisely, at each iteration $i \geq 0$, $k$-OSWL computes a coloring $C_i \colon V(G) \times G_k \to \mathbb{N}$ where we interpret elements $(v, \mathbf{g}) \in V(G) \times G_k$ as a vertex $v$ along with an ordered $k$-vertex subgraph. Given an ordered $k$-vertex subgraph $\mathbf{g} \in G_k$, initially, for $i = 0$, we set $C_0(v, \mathbf{g}) \coloneqq \mathsf{atp}(v, \mathsf{t}(\mathbf{g}))$, and for $i > 0$, we set

$$C_{i+1}(v, \mathbf{g}) \coloneqq \mathsf{RELABEL}\Big(\big(C_i(v, \mathbf{g}), \{\!\!\{ C_i(u, \mathbf{g}) \mid u \in \square \}\!\!\}\big)\Big),$$

where $\square$ is either $N_G(v)$ or $V(G)$. We compute the stable partition analogously to 1-WL. Finally, to compute a single color for a vertex $v$, we aggregate all ordered $k$-vertex subgraphs, i.e., we compute

$$C(v) \coloneqq \mathsf{RELABEL}\big(\{\!\!\{ C_\infty(v, \mathbf{g}) \mid \mathbf{g} \in G_k \}\!\!\}\big). \tag{2}$$

In other words, one can regard the $k$-OSWL as running 1-WL in parallel over $n^k$ graphs, one for each ordered $k$-vertex subgraph $\mathbf{g} \in G_k$, followed by combining the colors of each vertex in all these graphs. Furthermore, by restricting the number of considered subgraphs, the algorithm allows for more fine-grained control over the trade-off between scalability and expressivity. Note that 0-OSWL is equal to 1-WL. We also define a variation of the $k$-OSWL, denoted *vertex-subgraph $k$-OSWL*, which, unlike Equation (2), first computes a color $C(\mathbf{g})$ for each ordered $k$-vertex subgraph $\mathbf{g}$ by aggregating over vertices; see Appendix E for details.

We remark that in contrast to $k$-WL, which has to update the coloring of all $n^k$ ordered $k$-vertex subgraphs in a complicated manner, the computation of $k$-OSWL's coloring relies on the simple and easy-to-implement 1-WL. Furthermore, $k$-OSWL's computation can be either done in parallel or sequentially across all $n$ vertices and $n^k$ graphs. Despite its simplicity, in Section 3.2, we will show that the $k$-OSWL has high expressivity.

## 3.2 Ordered subgraph MPNNs

In the following, to study the expressivity of subgraph-enhanced GNNs, we introduce $k$-ordered subgraph MPNNs ($k$-OSANs), which can be viewed as neural variants of the $k$-OSWL. At initialization, $k$-OSANs learn two features for each element in $G_k$ and each vertex $v$

$$\mathbf{h}_{v,\mathbf{g}}^{(0)} \coloneqq \mathsf{UPD}(\mathsf{atp}(v, \mathsf{t}(\mathbf{g}))) \in \mathbb{R}^\mathsf{d}, \quad \text{and} \quad \boldsymbol{\pi}_{v,\mathbf{g}} \coloneqq \mathsf{UPD}_{\boldsymbol{\pi}}(\mathsf{atp}(v, \mathsf{t}(\mathbf{g}))),$$

where $\mathsf{UPD}$ and $\mathsf{UPD}_{\boldsymbol{\pi}}$ are differentiable, parameterized function, e.g., a neural network. Additional vertex features can be concatenated to the first feature. We use the second feature $\boldsymbol{\pi}_{v,\mathbf{g}}$ to select admissible ordered subgraphs for the vertex $v$; see below. Now in each layer $(i + 1)$, we update the feature of a vertex $v$ with regard to the $k$-ordered subgraph $\mathbf{g}$ as

$$\mathbf{h}_{v,\mathbf{g}}^{(i+1)} \coloneqq \mathsf{UPD}^{(i+1)}\Big(\mathbf{h}_{v,\mathbf{g}}^{(i)}, \mathsf{AGG}^{(i+1)}\big(\{\!\!\{ \mathbf{h}_{u,\mathbf{g}}^{(i)} \mid u \in \square \}\!\!\}\big)\Big),$$

where $\square$ is either $N_G(v)$ or $V(G)$. After $T$ such layers, for each vertex $v$, we then learn a joint feature over all $k$-ordered subgraphs, i.e., we apply subgraph aggregation

$$\mathbf{h}_v^{(T)} \coloneqq \mathsf{SAGG}\big(\{\!\!\{ \mathbf{h}_{v,\mathbf{g}}^{(T)} \mid \mathbf{g} \in G_k \text{ s.t. } \boldsymbol{\pi}_{v,\mathbf{g}} \neq \mathbf{0} \}\!\!\}\big). \tag{3}$$

Here, we leverage $\boldsymbol{\pi}_{v,\mathbf{g}} \neq \mathbf{0}$ to select a subset of the set of $k$-ordered subgraphs. Finally, analogous to GNNs, we use a READOUT layer to compute a single graph feature. Again, $\text{AGG}^{(i+1)}$, $\text{UPD}^{(i+1)}$, READOUT, and SAGG are differentiable, parameterized functions, e.g., neural networks.

We also define a variation of $k$-OSANs, denoted *vertex-subgraph $k$-OSANs*, which, unlike Equation (3), first compute a color for each ordered $k$-vertex subgraph; see Appendix E for details.

**Expressive power of $k$-OSANs** In the following, we study the expressive power of $k$-OSANs. The first result shows that any possible $k$-OSAN has at most the expressive power of $k$-OSWL in terms of distinguishing non-isomorphic graphs. Further, $k$-OSANs are in principle capable of reaching $k$-OSWL's expressive power. Hence, the $k$-OSWL upper bounds $k$-OSANs ability to represent permutation-invariant functions.

**Proposition 1.** For all $k \geq 1$, it holds that $k$-OSANs are upper bounded by $k$-OSWL in terms of distinguishing non-isomorphic graphs. Further, there exists a $k$-OSAN instance that has exactly the same expressive power as the $k$-OSWL.

The following result shows that any possible $k$-OSAN is upper-bounded by the $(k+1)$-WL in terms of distinguishing non-isomorphic graphs while the expressive power of $k$-OSANs and the $k$-WL are incomparable. That is, there exist non-isomorphic graphs $k$-WL cannot distinguish while $k$-OSANs can and vice versa.

**Proposition 2.** For all $k \geq 1$, it holds that $(k+1)$-WL is *stricly more* expressive than $k$-OSANs and there exist non-isomorphic graphs $k$-WL cannot distinguish while $k$-OSANs can and vice versa.

Finally, the following results shows that increasing the size of the subgraphs always leads to a strictly more expressive $k$-OSANs.

**Theorem 3.** For all $k \geq 1$, it holds that $(k+1)$-OSANs is *strictly more* expressive than $k$-OSANs.

**Subgraph-enhanced GNNs captured by $k$-OSANs** To exemplify the power and generality of $k$-OSANs, we show how $k$-OSANs cover most subgraph-enhanced GNNs; see Appendix F for a thorough overview. We say that $k$-OSANs *capture* a subgraph-enhanced GNN $G$ if there exists a $k$-OSAN instance that is at least as expressive as $G$.

The first results shows that $k$-OSANs capture $k$-marked GNNs ($k$-mGNNs) [Papp and Wattenhofer, 2022] and $k$-reconstruction GNNs ($k$-recGNNs) [Cotta et al., 2021]. For both approaches, the sets of $k$ vertices to be marked or deleted correspond to unordered $k$-vertex subgraphs. It then suffices to ensure that the update and aggregation functions in the $k$-OSANs treat the vertices in the selected $k$-vertex subgraphs as being marked or deleted.

**Proposition 4.** For $k \geq 1$, $k$-OSANs capture $k$-mGNNs and vertex-subgraph $k$-OSANs capture $k$-recGNNs.

Further, 1-OSANs capture identity-aware GNNs (idGNNs) [You et al., 2021], GNN As Kernel (kernelGNNs) [Zhao et al., 2021], and nested GNNs (nestedGNNs) [Zhang and Li, 2021]. Intuitively, in these approaches GNNs are used locally around each vertex. It thus suffices to ensure that the update and aggregation functions in the 1-OSANs use the selected single vertex subgraph to only pass messages locally.

**Proposition 5.** 1-OSANs capture idGNNs, kernelGNNs, and nestedGNNs.

Finally, $k$-OSANs capture the DS-GNNs with the vertex-deleted policy [Bevilacqua et al., 2021].[4]

**Proposition 6.** Vertex-subgraph $k$-OSANs capture DS-GNNs with the $k$-vertex-deleted policy.

We note that the above result can be further extended to accommodate the edge-deleted and ego-networks policy from Bevilacqua et al. [2021]; see Appendix F.

Importantly, by viewing existing subgraph-enhanced GNNs as $k$-OSANs we immediately gain insights into their expressive power. Previous results primarily focused on showing more expressivity than 1-WL.

---

[4]We note here that it is an open question if vertex-subgraph $k$-OSANs also capture the more general DSS-GNNs [Bevilacqua et al., 2021].

# 4 Data-driven Subgraph-enhanced GNNs

In the above section, we thoroughly investigated the expressive power of subgraph-enhanced GNNs. Specifically, we showed that they are strictly limited by the $(k+1)$-WL and that they can distinguish graphs which are not distinguishable by $k$-WL. As indicated by Proposition 1 and to reach maximal expressive power, however, we need to consider all possible ordered subgraphs, resulting in an exponential running time. Hence, in this section, we leverage the I-MLE framework [Niepert et al., 2021], to sample ordered subgraphs in a data-driven fashion. We first address the problem of learning the parameters of a probability distribution over ordered subgraphs using a GNN. Secondly, we show how to approximately sample from this intractable distribution. Subsequently, these subgraphs are used within a $k$-OSAN to compute a graph representation. Finally, we propose a gradient estimation scheme that allows us to use backpropagation in the resulting discrete-continuous architecture.

**Parameterizing probability distributions over subgraphs** Contrary to existing approaches, which often consider all possible subgraphs or sample a fraction of subgraphs uniformly at random, our method maintains a probability distribution over (ordered) subgraphs. Let $G$ be a graph where each vertex $v$ has an initial feature $\mathbf{h}_v^{(0)}$, which we stack row-wise over all vertices into the feature matrix $\mathbf{H} \in \mathbb{R}^{n \times d}$. Further, let $h_{\mathbf{W}_1} : \mathcal{G} \times \mathbb{R}^{n \times d} \to \mathbb{R}^{m \times n}$ be a permutation-equivariant function, e.g., a message-passing GNN, parameterized by $\mathbf{W}_1$, called *upstream model*, mapping a graph $G$ and its initial features $\mathbf{H}$ to a parameter matrix

$$\boldsymbol{\theta} := h_{\mathbf{W}_1}(G, \mathbf{H}) \in \mathbb{R}^{m \times n}.$$

Intuitively, each parameter $\theta_{ij}$ is an unnormalized prior probability of vertex $j$ being part of the $i$th sampled subgraph of $G$. Let $\boldsymbol{\theta}_i := (\theta_{i1}, \dots, \theta_{in})$ for $i \in [m]$. We use these to parameterize $m$ probability distributions $p(\boldsymbol{z}; \boldsymbol{\theta}_i)$, for $i \in [m]$, over vector encodings of ordered $k$-vertex subgraphs of $G$, i.e.,

$$p(\boldsymbol{z}; \boldsymbol{\theta}_i) := \begin{cases} \exp\left(\langle \boldsymbol{z}, \boldsymbol{\theta}_i \rangle - A(\boldsymbol{\theta}_i)\right) & \text{if } \boldsymbol{z} \in \mathcal{Z}, \\ 0 & \text{otherwise,} \end{cases} \tag{4}$$

where $\langle \cdot, \cdot \rangle$ is the standard inner product and $A(\boldsymbol{\theta}_i)$ is the *log-partition function* defined as $A(\boldsymbol{\theta}_i) := \log\left(\sum_{\boldsymbol{z} \in \mathcal{Z}} \exp\left(\langle \boldsymbol{z}, \boldsymbol{\theta}_i \rangle\right)\right)$. Furthermore, for a distribution over *unordered* $k$-vertex subgraphs, $\mathcal{Z}$ is the set of all binary $n$-component vectors with exactly $k$ non-zero entries indicating which vertices are part of a subgraph of $G$. Hence, there is a bijection between $\mathcal{Z}$ and the set of unordered $k$-vertex subgraphs of $G$. For a distribution over *ordered* $k$-vertex subgraphs, the set $\mathcal{Z}$ is the set of all vectors in $[k]^n$ with $k$ non-zero entries. For each non-zero entry $\boldsymbol{z}_i$ for $\boldsymbol{z} \in \mathcal{Z}$ it holds that $\boldsymbol{z}_i = j$ if and only if vertex $i$ has *rank* $k + 1 - j$ in the ordered subgraph, encoding the position in the ordered graph. This encoding is required for the gradient computation we perform later. For instance, in an ordered 5-vertex subgraph, if a node has rank 1 but should have rank 5 to obtain a lower loss, then the gradient of the downstream loss is proportional to $5 - 1 = 4$. Similarly, if a node $i$ is not part of the ordered subgraph, that is, $\boldsymbol{z}_i = 0$ but should be in position 1 of the ordered subgraph, then the gradient of a downstream loss is proportional to $0 - 5$. For any $i, j \in [k]$ with $i \neq j$ we have that $\boldsymbol{z}_i \neq \boldsymbol{z}_j$. Hence, again, there is a bijection between $\mathcal{Z}$ and the set of ordered $k$-vertex subgraphs of $G$.

**Efficient approximate sampling of subgraphs** Computing the log-partition function and sampling exactly from the probability distribution in Equation (4) is intractable for both ordered and unordered graphs. Since it is tractable, however, to compute a configuration with a highest probability, a maximum a posteriori (MAP) configuration $\boldsymbol{z}^*(\boldsymbol{\theta}_i)$, we can use perturb-and-MAP [Papandreou and Yuille, 2011, Niepert et al., 2021] to sample approximately. For unordered graphs, determining the top-$k$ values in $\boldsymbol{\theta}_i$ suffices, while for ordered graphs, we additionally require their rank. Therefore, the worst-case running time of computing $\boldsymbol{z}^*(\boldsymbol{\theta}_i)$ for ordered graphs of size $k$ is in $O(n + k \log k)$. That is, we first use a selection algorithm to find the $k$th largest element $E$ in the list of weights, taking time $\mathcal{O}(n)$, e.g., using the Quickselect algorithm. Now, we go through the list again and select all entries larger to $E$, taking time $\mathcal{O}(n)$. Finally, we sort the $k$ values.

Now, to use perturb-and-MAP to *approximately* sample the $i$-th ordered $k$-vertex subgraph $\mathbf{g}_i$ from the above probability distributions, we compute

$$\mathbf{g}_i := \mathtt{adj}\left(\boldsymbol{z}^*(\boldsymbol{\theta}_i + \boldsymbol{\epsilon}_i)\right) \quad \text{with} \quad \boldsymbol{\epsilon}_i \sim \rho(\boldsymbol{\epsilon}),$$

where $\rho(\boldsymbol{\epsilon})$ is a noise distribution such as the Gumbel distribution and $\mathtt{adj}$ converts the above vector encoding of the (ordered) subgraph to an $n \times n$ adjacency matrix as follows. The $j$th row or column encodes the vertex of the ordered subgraph with rank $j$ and its incident edges within the ordered subgraph. All other

entries are set to 0, i.e., they are masked out. We therefore sample a multiset of (ordered) subgraphs $S := \{\!\{ \boldsymbol{g}_1, ..., \boldsymbol{g}_m \}\!\} \subseteq G_k$, which act as the input to a $k$-OSAN instance $f_{\mathbf{W}_2}$, called *downstream model*, where $\boldsymbol{\pi}_{v,\mathbf{g}} \neq \mathbf{0}$ for $v \in V(G)$ if $\mathbf{g} \in S$, to compute the target outputs

$$f_{\mathbf{W}_2}(G, \mathbf{H}, \{\!\{ \boldsymbol{g}_1, ..., \boldsymbol{g}_m \}\!\}).$$

**Backpropagating through the subgraph distribution** Now that we have outlined a way to approximately sample subgraphs, we still need to learn the parameters $\boldsymbol{\omega} = (\mathbf{W}_1, \mathbf{W}_2)$ of the upstream and downstream model. Hence, given a set of examples $\{(G_j, \mathbf{H}_j, \hat{\boldsymbol{y}}_j)\}_{j=1}^N$, we are concerned with finding approximate solutions to $\min_{\boldsymbol{\omega}} 1/N \sum_j L(G, \mathbf{H}, \hat{\boldsymbol{y}}_j; \boldsymbol{\omega})$, where $L$ is the expected training error

$$L(G, \mathbf{H}, \hat{\boldsymbol{y}}_j; \boldsymbol{\omega}) := \mathbb{E}_{\boldsymbol{g}_i \sim p(\boldsymbol{z}; \boldsymbol{\theta}_i)} \left[ \ell \left( f_{\mathbf{W}_2} \left( G, \mathbf{H}, \{\!\{ \boldsymbol{g}_1, ..., \boldsymbol{g}_m \}\!\} \right), \hat{\boldsymbol{y}} \right) \right], \tag{5}$$

with $\boldsymbol{\theta} := h_{\mathbf{W}_1}(G, \mathbf{H})$ and $\ell \colon \mathcal{Y} \times \mathcal{Y} \to \mathbb{R}^+$ is a point-wise loss function. The challenge of training a model as defined in Equation (5) is to compute $\nabla_{\boldsymbol{\theta}} L(G, \mathbf{H}, \hat{\boldsymbol{y}}_j; \boldsymbol{\omega})$, i.e., the gradient with respect to the parameters $\boldsymbol{\theta}$ of the probability distribution for the expected loss. In this work, we utilize implicit maximum likelihood learning, a recent framework that allows us to use algorithmic solvers of combinatorial optimization problems as black-box components [Rolinek et al., 2020, Niepert et al., 2021]. A particular instance of the framework uses implicit differentation via perturbation [Domke, 2010]. We derive the gradient computation for a single ordered subgraph $\mathbf{g}_i$ to simplify the notation. We compute the gradients of the downstream loss with respect to parameters $\boldsymbol{\theta}_i$ as

$$\nabla_{\boldsymbol{\theta}_i} L(G, \mathbf{H}, \hat{\boldsymbol{y}}; \boldsymbol{\omega}) \approx \mathbb{E}_{\boldsymbol{\epsilon}_i \sim \rho(\boldsymbol{\epsilon})} \left[ 1/\lambda \left( \boldsymbol{z}^* \left( \boldsymbol{\theta}_i + \boldsymbol{\epsilon}_i \right) - \boldsymbol{z}^* \left( \boldsymbol{\theta}_i + \boldsymbol{\epsilon}_i - \lambda \widehat{\nabla} \right) \right) \right],$$

where $\lambda > 0$ and $\widehat{\nabla}_v$, the approximated gradient for a single vertex $v$, is defined as

$$\widehat{\nabla}_v := \mathbf{agg}\big(\{\!\{ \left[ \nabla_{\boldsymbol{g}_i} \ell \left( f_{\mathbf{W}_2} \left( G, \mathbf{H}, \{\!\{ \boldsymbol{g}_i \}\!\} \right) \right) \right]_{v,w} \mid w \in N_G(v) \}\!\}\big),$$

with $v \in V(G)$. Here, $\mathbf{agg}$ can be any aggregation function such as the element-wise sum or mean. Hence, to approximate the gradients with respect to the parameter $\boldsymbol{\theta}_{iv}$, which corresponds to vertex $v$ of the input graph, we aggregate the gradients of the downstream loss with respect to all edges $(v, w)$ incident to vertex $v$ in the *original* graph.

Hence, the above techniques allow us to efficiently learn to sample subgraphs, which are then fed into a $k$-OSAN to learn a graph representation while optimizing the parameters of all components in an end-to-end fashion.

## 5 Experimental evaluation

Here, we aim to empirically investigate the learning performance and efficiency of data-driven subgraph-enhanced GNNs, instances of the $k$-OSAN framework, compared to non-data-driven ones. Specifically, we aim to answer the following questions.

**Q1** Do data-driven subgraph-enhanced GNNs exhibit better predictive performance than non-data-driven ones?

**Q2** Does the graph structure of the subgraphs sampled affect predictive performance?

**Q3** Does data-driven sampling have an advantage in efficiency and predictive performance when used within state-of-the-art subgraph-enhanced GNNs?

All experimental results are fully reproducible from the source code provided at `https://github.com/ Spazierganger/OSAN`.

**Datasets** To compare our data-driven, subgraph-enhanced GNNs to non-data-driven ones and standard GNN baselines, we used the ALCHEMY [Chen et al., 2019a], the QM9 [Ramakrishnan et al., 2014, Wu et al., 2018], OGBG-MOLESOL [Hu et al., 2020], and the ZINC [Dwivedi et al., 2020, Jin et al., 2017] graph-level regression datasets; see Table 16 in Appendix C for dataset statistics and properties. In addition, we used the EXP dataset [Abboud et al., 2020] to investigate the additional expressive power of subgraph-enhanced GNNs over standard ones. Following Morris et al. [2020b], we opted not to use the 3D-coordinates of the ALCHEMY dataset to solely show the benefits of the data-driven subgraph-enhanced

Table 1: Results on large-scale regression datasets, data-driven versus non-data-driven subgraph sampling.

(a) Results for the OGBG-MOLESOL dataset.

| Method | Operat. | Type | # | # Subg. | RSMSE ↓ |
|---|---|---|---|---|---|
| Baseline | | | | | $1.193_{\pm 0.083}$ |
| Random | Delete | Vertex | 1 | 3 | $1.215_{\pm 0.095}$ |
| I-MLE | | | | | $1.053_{\pm 0.080}$ |
| Random | Delete | Vertex | 1 | 10 | $1.128_{\pm 0.055}$ |
| I-MLE | | | | | $0.984_{\pm 0.086}$ |
| Random | Delete | Vertex | 2 | 1 | $1.283_{\pm 0.080}$ |
| I-MLE | | | | | $0.968_{\pm 0.102}$ |
| Random | Delete | Vertex | 2 | 3 | $1.132_{\pm 0.020}$ |
| I-MLE | | | | | $1.081_{\pm 0.021}$ |
| Random | Delete | Vertex | 5 | 3 | $0.992_{\pm 0.115}$ |
| I-MLE | | | | | $1.115_{\pm 0.076}$ |
| Random | Delete | Vertex | 5 | 10 | $1.186_{\pm 0.154}$ |
| I-MLE | | | | | $1.137_{\pm 0.053}$ |
| Random | Select | Vertex | 10 | 1 | $1.128_{\pm 0.022}$ |
| I-MLE | | | | | $1.099_{\pm 0.099}$ |
| Random | Delete | Edge | 1 | 3 | $1.240_{\pm 0.029}$ |
| I-MLE | | | | | $1.106_{\pm 0.069}$ |
| Random | Delete | Edge | 1 | 10 | $1.152_{\pm 0.046}$ |
| I-MLE | | | | | $1.056_{\pm 0.071}$ |
| Random | Delete | Edge | 3 | 3 | $1.084_{\pm 0.076}$ |
| I-MLE | | | | | $1.052_{\pm 0.049}$ |
| Random | Delete | Edge | 3 | 10 | $1.099_{\pm 0.071}$ |
| I-MLE | | | | | $1.077_{\pm 0.079}$ |
| Random | Delete | 2-Ego | – | 3 | $1.071_{\pm 0.062}$ |
| I-MLE | | | | | $0.959_{\pm 0.184}$ |

(b) Result for the ALCHEMY dataset.

| Method | Operat. | Type | # | # Subg. | MAE ↓ |
|---|---|---|---|---|---|
| Baseline | | | | | $11.12_{\pm 0.69}$ |
| Random | Delete | Vertex | 1 | 3 | $13.26_{\pm 0.41}$ |
| I-MLE | | | | | $8.78_{\pm 0.28}$ |
| Random | Delete | Vertex | 1 | 10 | $12.11_{\pm 0.21}$ |
| I-MLE | | | | | $8.87_{\pm 0.12}$ |
| Random | Delete | Vertex | 2 | 3 | $12.66_{\pm 0.28}$ |
| I-MLE | | | | | $9.01_{\pm 0.27}$ |
| Random | Delete | Vertex | 5 | 3 | $10.29_{\pm 0.30}$ |
| I-MLE | | | | | $9.22_{\pm 0.06}$ |
| Random | Delete | Edge | 1 | 3 | $11.66_{\pm 0.63}$ |
| I-MLE | | | | | $10.80_{\pm 0.31}$ |
| Random | Delete | Edge | 2 | 3 | $10.79_{\pm 0.64}$ |
| I-MLE | | | | | $10.56_{\pm 0.44}$ |
| Random | Delete | Edge | 5 | 3 | $9.15_{\pm 0.12}$ |
| I-MLE | | | | | $9.08_{\pm 0.28}$ |
| Random | Select | Vertex | 5 | 3 | $11.48_{\pm 0.60}$ |
| I-MLE | | | | | $9.22_{\pm 0.14}$ |
| Random | Select | Edge | 5 | 3 | $8.99_{\pm 0.24}$ |
| I-MLE | | | | | $8.95_{\pm 0.29}$ |
| Random | Delete | 1-Ego | – | 3 | $14.98_{\pm 0.49}$ |
| I-MLE | | | | | $11.15_{\pm 1.09}$ |
| Random | Select | 5-Ego | – | 3 | $14.97_{\pm 0.23}$ |
| I-MLE | | | | | $13.83_{\pm 1.06}$ |

GNNs regarding graph structure. All datasets, excluding EXP and OGBG-MOLESOL, are available from Morris et al. [2020a].[5]

**Neural architectures and experimental protocol** For all datasets and architectures, we used the competitive GIN layers [Xu et al., 2019] for the baselines and the downstream models. For data with (continuous) edge features, we used a 2-layer MLP to map them to the same number of components as the vertex features and combined them using summation. We describe the upstream and downstream models' architecture used for each dataset in the following. We stress here that we always used the same hyperparameters for the downstream model and the baselines.

**Sampling subgraphs** Since the number of unordered $k$-vertex subgraphs is considerably smaller than the number of ordered $k$-vertex subgraphs, we opted to consider unordered $k$-vertex subgraphs; see also Appendix F.1. Further, since vertex-subgraph $k$-OSANs, see Appendix E, are easier to implement efficiently and are closer to DS-GNNs variant of ESAN [Bevilacqua et al., 2021], we opted to use them for the empirical evaluation. In addition, we used a simple GNN architecture for the upstream model to compute initial features for the subgraphs for ease of implementation. We experimented with selecting and deleting a various number of vertices, edges, and subgraphs induced by $k$-hop neighborhoods ($k$-Ego) for all datasets; see Appendix C for details.

**Upstream models** For all datasets and experiments, we used a GCN model [Kipf and Welling, 2017] consisting of three GCN layers, with batch norm and ReLU activation after each layer. We set the hidden dimensions to that of the downstream model one. The model either outputs the vertex or edge embeddings according to the task. We computed edge embeddings based on the vertex features of the incident vertices after the last layers and the edge attributes provided by the dataset.

When sampling multiple subgraphs with I-MLE, they tend to have similar structures. In other words, I-MLE learns similar distributions in different channels of the neural network. This phenomenon is not in our favor, as we need to cover the original full graph as much as possible. To mitigate this issue, we propose an auxiliary loss for the diversity of subgraphs. We calculate the cosine similarity between the selected vertex or edge masks of every two subgraphs and try to minimize the average similarity value. We tune the weight for the auxiliary loss on the log scale, e.g., 0.1, 1, 10, and so on.

**Downstream and baseline models** See Appendix A for a detailed description of the architecture used for the downstream and baseline models, and how we processed subgraphs. For processing the subgraphs, we performed similar steps like ESAN. We first applied intra-subgraph aggregation for the vertices within each subgraph and obtained graph embeddings for each subgraph. After that, we performed inter-subgraph mean

---

[5]https://chrsmrrs.github.io/datasets/

pooling to obtain a single embedding vector for the original graph. It is worth noting that ESAN does not exclude the vertices deleted during graph pooling but removes the adjacent edges of those nodes. In our experiments, we masked out the deleted or unselected nodes.

See Appendix C for further details on the experiments.

## 5.1 Results and discussion

In the following, we answer the research questions **Q1** to **Q3**.

**A1** See Tables 1, 2a, 4 and 5 (in the appendix). On all five datasets, the subgraph-enhanced GNN models based on I-MLE beat the random baseline, excluding edge sampling configurations on the ALCHEMY and the QM9 dataset; see Tables 1b and 2a. For example, on the OGBG-MOLESOL the average gain over the random baseline is over 11%. Similar improvements can be observed over the other four datasets. Moreover, the results on the EXP dataset, see Appendix C, clearly indicate that the added expressivity of the (data-driven) subgraph-enhanced GNNs translates into improved predictive performance. The data-driven subgraph-enhanced GNNs improve the accuracy of the non-subgraph-enhanced GNN by almost 50% in all configurations while improving over the random subgraph-enhanced GNN baseline by almost 6%. The data-driven subgraph-enhanced GNNs also clearly improve over the (non-subgraph-enhanced) GNN baseline on four out of five datasets.

**A2** See Tables 1 and 2a (in the appendix). Deleting or selecting subgraphs leads to a clear boost in predictive performance across datasets over the random baseline while also improving over the non-subgraph-enhanced GNN baseline. Further, on all datasets, learning to delete or select $k$-hop neighborhood subgraphs for $k \in \{2, 3\}$ leads to a clear boost over the random as well as non-subgraph-enhanced baselines. However, the number of deleted vertices seems to affect the predictive performance. For example, on the OGBG-MOLESOL dataset, going from deleting one 2-vertex subgraph to one 10-vertex subgraph leads to a drop in performance. Hence, the drop in performance of the latter is in contrast to our theoretical findings, i.e., larger subgraphs lead to improved expressivity, indicating that more work should be done to understand subgraph-enhanced GNNs' generalization ability. Interestingly, deleting edges did not perform as well as deleting vertices or other subgraphs. We speculate that a more powerful edge embedding method is needed here, which computes edge features directly instead of learning them from vertex features.

**A3** See Table 2b (in the appendix). The I-MLE-based ESAN severely speeds up the computation time. That is, across all configurations, we achieve a significant speed-up. For some configurations, e.g., sampling three vertices, the I-MLE based ESAN is more than 3.5 times faster than the non-data-driven ESAN while taking about the same time as the simple random baseline. We stress here that the ESAN implementation provided by Bevilacqua et al. [2021] precomputes subgraphs in a preprocessing step, which is not possible when learning to sample subgraphs using ESAN. Regarding predictive performance, the I-MLE based ESAN is slightly behind the non-data-driven one, although always better than the non-subgraph enhanced GNN baseline; see Table 5 in Appendix C.

## 6 Conclusion

We introduced the $k$-OSAN framework to study the expressive power of recently introduced subgraph-enhanced GNNs. We showed that any such architecture is strictly less powerful than the $(k + 1)$-WL while being incomparable to the $k$-WL in representing permutation-invariant functions over graphs. Further, to circumvent random or heuristic subgraph selection, we devised a data-driven variant of $k$-OSANs which learn to select subgraph for a given data distribution. Empirically, we verified that such data-driven subgraph selection is superior to previously used random sampling in predictive performance. Further, when compared to state-of-the-art models, we showed promising performance in terms of computation time while still providing good predictive performance. We believe that our paper provides a first step in unifying combinatorial insights on the expressive power of GNNs with data-driven insights.

## Acknowledgments and Disclosure of Funding

CM is partially funded a DFG Emmy Noether grant (468502433) and RWTH Junior Principal Investigator Fellowship under the Excellence Strategy of the Federal Government and the Länder. GR is funded by the DFG Research Grants Program–RA 3242/1-1–411032549. MN acknowledges funding by the German Research Foundation under Germany's Excellence Strategy–EXC 2075.

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
