# A   Related work (expanded)

**GNNs** Recently, GNNs [Gilmer et al., 2017, Scarselli et al., 2009] emerged as the most prominent graph representation learning architecture. Notable instances of this architecture include, e.g., [Duvenaud et al., 2015, Hamilton et al., 2017, Veličković et al., 2018], which can be subsumed under the message-passing framework introduced in [Gilmer et al., 2017]. In parallel, approaches based on spectral information were introduced in, e.g., [Defferrard et al., 2016, Bruna et al., 2014, Kipf and Welling, 2017, Monti et al., 2017]—all of which descend from early work in [Baskin et al., 1997, Kireev, 1995, Micheli and Sestito, 2005, Merkwirth and Lengauer, 2005, Micheli, 2009, Scarselli et al., 2009, Sperduti and Starita, 1997]. Recent extensions and improvements to the GNN framework include approaches to incorporate different local structures (around subgraphs), e.g., [Abu-El-Haija et al., 2019, Flam-Shepherd et al., 2020, Jin et al., 2019, Niepert et al., 2016, Xu et al., 2018], novel techniques for pooling vertex representations to perform graph classification, e.g., [Bianchi et al., 2020, Cangea et al., 2018, Gao and Ji, 2019, Grattarola et al., 2021, Ying et al., 2018, Zhang et al., 2018], incorporating distance information [You et al., 2019], non-Euclidean geometry approaches [Chami et al., 2019], and more efficient GNNs, e.g., [Fey et al., 2021, Li et al., 2021]. Furthermore, recently, empirical studies on neighborhood aggregation functions for continuous vertex features [Corso et al., 2020a], edge-based GNNs that leverage physical knowledge [Anderson et al., 2019, J. Klicpera, 2020, Klicpera et al., 2021], studying over-smoothing and over-squashing phenomena [Alon and Yahav, 2020, Bodnar et al., 2022, Addanki et al., 2021], and sparsification methods [Rong et al., 2020] emerged. Surveys of recent advancements in GNN techniques can be found, e.g., in Chami et al. [2020], Wu et al. [2019], Zhou et al. [2018].

**Limits of GNNs and more expressive architectures** Recently, connections of GNNs to Weisfeiler–Leman type algorithms have been shown [Azizian and Lelarge, 2020, Barceló et al., 2020, Chen et al., 2019b, Geerts et al., 2020, Geerts, 2020, Geerts and Reutter, 2022, Maehara and NT, 2019, Maron et al., 2019, Morris et al., 2019, 2022, Xu et al., 2019]. Specifically, [Morris et al., 2019, Xu et al., 2019] showed that the expressive power of any possible GNN architecture is limited by the 1-WL in terms of distinguishing non-isomorphic graphs.

Triggered by the above results, a large set of papers proposed architectures to overcome the expressivity limitations of 1-WL. Morris et al. [2019] introduced $k$-*dimensional* GNNs which rely on a message-passing scheme between subgraphs of cardinality $k$. Similar to [Morris et al., 2017], the paper employed a local, set-based (neural) variant of the 1-WL. Later, this was refined in [Azizian and Lelarge, 2020, Maron et al., 2019] by introducing $k$-*order folklore graph neural networks*, which are equivalent to the folklore or oblivious variant of the $k$-WL [Grohe, 2021, Morris et al., 2021] in terms of distinguishing non-isomorphic graphs. Subsequently, Morris et al. [2020b] introduced neural architectures based on a local version of the $k$-WL, which only considers a subset of the original neighborhood, taking sparsity of the underlying graph (to some extent) into account. Chen et al. [2019b] connected the theory of universal approximations of permutation-invariant functions and the graph isomorphism viewpoint and introduced a variation of the 2-WL. Geerts and Reutter [2022] introduced a higher-order message-passing framework that allows us to obtain upper bounds of extension of GNNs in terms of $k$-WL.

Recent works have extended the expressive power of GNNs, e.g., by encoding vertex identifiers [Murphy et al., 2019, Vignac et al., 2020], using random features [Abboud et al., 2020, Dasoulas et al., 2020, Sato et al., 2020], homomorphism and subgraph counts [Barceló et al., 2021, Bouritsas et al., 2020, NT and Maehara, 2020], spectral information [Balcilar et al., 2021], simplicial and cellular complexes [Bodnar et al., 2021b,a], persistent homology [Horn et al., 2021], random walks [Tönshoff et al., 2021], graph decompositions [Talak et al., 2021], or distance [Li et al., 2020] and directional information [Beaini et al., 2020]. See Morris et al. [2021] for an in-depth survey on this topic.

**Differentiating through discrete structures** Recently, numerous papers have aimed to combine discrete random variables and (continuous) neural network components and addressed the resulting gradient estimation problem. Most existing approaches used various types of relaxation of discrete distributions. For instance, Maddison et al. [2017] and Jang et al. [2017] proposed the Gumbel-softmax distribution to relax categorical random variables. REBAR [Tucker et al., 2017] combined the Gumbel-softmax trick with the score-function estimator but is tailored to categorical distributions. Recent work on relaxed gradient estimators derived several extensions of the softmax trick [Paulus et al., 2020]. However, the Gumbel-softmax distribution is only directly applicable to categorical variables. For more complex distributions, one has to come up with tailor-made relaxations or use the straight-through or score function estimators, see, e.g., Grover et al. [2019], Kim et al. [2016]. Further, Grathwohl et al. [2018], Tucker et al.

[2017] developed parameterized control variates based on continuous relaxations for the score-function estimator. In this work, we have to sample and select sparse, discrete, and complex substructures of a given input graph. Due to the resulting exponential number of possible substructures, we cannot use the Gumbel-softmax trick for categorical distributions. On the other hand, the requirement to sample sparse and discrete substructures does not allow us to utilize relaxations. Therefore, we use I-MLE, a recently proposed general-purpose framework to combine neural and discrete components [Niepert et al., 2021].

## B  Weisfeiler–Leman algorithm (expanded)

Due to the shortcomings of the 1-WL or color refinement in distinguishing non-isomorphic graphs, several researchers [Babai, 1979, 2016, Immerman and Lander, 1990], devised a more powerful generalization of the former, today known as the *k-dimensional Weisfeiler-Leman algorithm* ($k$-WL).[6,7]

Intuitively, to surpass the limitations of the 1-WL, the algorithm colors ordered subgraphs instead of a single vertex. More precisely, given a graph $G$, it colors the tuples from $V(G)^k$ for $k \geq 1$ instead of the vertices. By defining a neighborhood between these tuples, we can define a coloring similar to the 1-WL. Formally, let $G$ be a graph, and let $k \geq 2$. In each iteration $i \geq 0$, the algorithm, similarly to the 1-WL, computes a *coloring* $C_i^k \colon V(G)^k \to \mathbb{N}$. In the first iteration, $i = 0$, the tuples $\mathbf{v}$ and $\mathbf{w}$ in $V(G)^k$ get the same color if they have the same atomic type, i.e., $C_0^k(\mathbf{v}) \coloneqq \mathsf{atp}(\mathbf{v})$. Now, for $i > 0$, $C_{i+1}^k$ is defined by

$$C_{i+1}^k(\mathbf{v}) \coloneqq \mathsf{RELABEL}\big(C_i^k(\mathbf{v}), M_i(\mathbf{v})\big),$$

with $M_i(\mathbf{v})$ the multiset

$$M_i(\mathbf{v}) \coloneqq \{\!\{(C_i^k(\phi_1(\mathbf{v}, w)), \ldots, C_i^k(\phi_k(\mathbf{v}, w))) \mid w \in V(G)\}\!\}$$

and where

$$\phi_j(\mathbf{v}, w) \coloneqq (v_1, \ldots, v_{j-1}, w, v_{j+1}, \ldots, v_k).$$

That is, $\phi_j(\mathbf{v}, w)$ replaces the $j$-th component of the tuple $\mathbf{v}$ with the vertex $w$. Hence, two tuples are *adjacent* or *j-neighbors*, with respect to a vertex $w$, if they are different in the $j$th component (or equal, in the case of self-loops). Again, we run the algorithm until convergence, i.e.,

$$C_i^k(\mathbf{v}) = C_i^k(\mathbf{w}) \iff C_{i+1}^k(\mathbf{v}) = C_{i+1}^k(\mathbf{w}),$$

for all $\mathbf{v}$ and $\mathbf{w}$ in $V(G)^k$ holds, and call the partition of $V(G)^k$ induced by $C_i^k$ the stable partition. For such $i$, we define $C_\infty^k(\mathbf{v}) = C_i^k(\mathbf{v})$ for $\mathbf{v}$ in $V(G)^k$. Hence, two tuples $\mathbf{v}$ and $\mathbf{w}$ with the same color in iteration $(i-1)$ get different colors in iteration $i$ if there exists a $j$ in $[k]$ such that the number of $j$-neighbors of $\mathbf{v}$ and $\mathbf{w}$, respectively, colored with a certain color is different. We set $C_\infty^k(v) \coloneqq C_\infty^k(v, \ldots, v)$ and refer to this as the color of the vertex $v$.

To test whether two graphs $G$ and $H$ are non-isomorphic, we run the $k$-WL in "parallel" on both graphs. Then, if the two graphs have a different number of vertices colored $c$ in $\mathbb{N}$, the $k$-WL *distinguishes* the graphs as non-isomorphic. By increasing $k$, the algorithm becomes more powerful in distinguishing non-isomorphic graphs, i.e., for each $k \geq 1$, there are non-isomorphic graphs distinguished by $(k+1)$-WL but not by $k$-WL [Cai et al., 1992].

**The Weisfeiler–Leman hierarchy and permutation-invariant function approximation**  The Weisfeiler–Leman hierarchy is a purely combinatorial algorithm for testing graph isomorphism. However, the graph isomorphism function, mapping non-isomorphic graphs to different values, is the hardest to approximate permutation-invariant function. Hence, the Weisfeiler–Leman hierarchy has strong ties to GNNs' capabilities to approximate permutation-invariant or equivariant functions over graphs. For example, Morris et al. [2019], Xu et al. [2019] showed that the expressive power of any possible GNN architecture is limited by 1-WL in terms of distinguishing non-isomorphic graphs. Azizian and Lelarge [2020] refined these results by showing that if an architecture is capable of simulating $k$-WL and allows the application of universal neural networks on vertex features, it will be able to approximate any

---

[6]In [Babai, 2016] László Babai mentions that he first introduced the algorithm in 1979 together with Rudolf Mathon from the University of Toronto.

[7]In this paper $k$-WL corresponds to original version [Babai, 1979, 2016, Immerman and Lander, 1990] which is sometimes referred to as the "folklore" version in the literature. It corresponds to the "oblivious" $(k+1)$-WL version often used in the graph learning community [Grohe, 2021].

permutation-equivariant function below the expressive power of $k$-WL; see also Chen et al. [2019b]. Hence, if one shows that one architecture distinguishes more graphs than another, it follows that the corresponding GNN can approximate more functions. These results were refined in Geerts and Reutter [2022] for color refinement and taking into account the number of iterations of $k$-WL.

## C  Datasets, details on the experiments, and additional experimental results

In the following, we outline details on the experiments.

**Additional details on the upstream model**   When sampling multiple subgraphs with I-MLE, they tend to have similar structures. In other words, I-MLE learns similar distributions in different channels of the neural network. This phenomenon is not in our favor as we need to cover the original full graph as much as possible. To mitigate this issue, we propose an auxiliary loss for the diversity of subgraphs. We calculate the KL-divergence between the selected vertex or edge masks and an all-one vector and try to minimize the value. We tune the weight for the auxiliary loss on the log scale, e.g., $0.1$, $1$, $10$, and so on.

**Downstream and baseline models**   For the larger molecular regression tasks ALCHEMY, QM9, and ZINC, we closely followed the hyperparameters found in Chen et al. [2019a], Gilmer et al. [2017], and Dwivedi et al. [2020] respectively, and used GIN layers. That is, for ZINC, we used four GIN layers with a hidden dimension of 256 followed by batch norm and a 4-layer MLP for the joint regression of the target after applying ESAN mean pooling. Moreover, we report results on ZINC dataset with PNA model architecture. We mainly follow the configurations of [Corso et al., 2020b] and the official implementation of [Fey and Lenssen, 2019]. For the number of hidden dimensions, where we used 128 instead of 75. For ALCHEMY and QM9, we used six layers with 64 (hidden) features and a set2seq layer [Vinyals et al., 2016] for graph-level pooling, followed by a 2-layer MLP for the joint regression of the twelve targets. Moreover, following [Chen et al., 2019a, Gilmer et al., 2017], we normalized the targets of the training split to zero mean and unit variance and report re-normalized testing scores. We used a single model to predict all targets and report (mean) MAE. For the GNN baseline for the QM9 dataset, we computed edge-wise $\ell_2$ distances based on the 3D coordinates and concatenated them to the edge features. We note here that our intent is not the beat state-of-the-art, physical knowledge-incorporating architectures, e.g., DimeNet [J. Klicpera, 2020] or Cormorant [Anderson et al., 2019], but to solely show the benefits of data-driven subgraph-enhanced GNNs. Further, to compare to ESAN, we used the same architecture as Bevilacqua et al. [2021]. For the EXP dataset, we processed the raw dataset following Abboud et al. [2020] and used six GIN layers, each with a hidden dimension of 32, followed by mean pooling and one linear layer immediately after mean pooling. For the OGBG-MOLESOL and OGBG-MOLBACE datasets, we followed OGB's [Hu et al., 2020] official GIN model architecture without virtual vertices, i.e., five GIN layers each with 300 hidden dimensions, and mean pooling as the final layer. For the PROTEINS dataset, we followed the DS-GNN setting of ESAN paper, i.e., using 32 hidden dimensions, 4 hidden layers, and mean pooling as the last layer.

**Sampling subgraphs**   Since the number of unordered $k$-vertex subgraphs is considerably smaller than the number of ordered $k$-vertex subgraphs, we opted to consider unordered $k$-vertex subgraphs; see also Appendix F.1. Further, since vertex-subgraph $k$-OSANs, see Appendix E, are easier to implement efficiently and are closer to ESAN [Bevilacqua et al., 2021], we opted to use them for the empirical evaluation. In addition, we used a simple GNN architecture for the upstream model to compute initial features for the subgraphs for ease of implementation. We experimented with selecting and deleting a various number of vertices, edges, and subgraphs induced by $k$-hop neighborhoods ($k$-Ego) for all datasets. We outline the subgraph sampling methods for each dataset below.

For ALCHEMY, we opted for learning to delete three vertices or edges. We also looked at sampling three subgraphs on five vertices or edges. Finally, we looked at sampling three 3-hop neighborhood subgraphs. For QM9, we opted for learning to delete one vertex or edge, learning to select three subgraphs with ten vertex or edges, and sampling ten 3-hop neighborhood subgraphs. For the OGBG-MOLESOL dataset, we looked at learning to delete one vertex three and ten times, two vertices one time, two subgraphs on five vertices, and one subgraph on ten vertices. Further, we looked at deleting one edge ten times and three 2-hop neighborhood subgraphs. For ZINC, we opted to learn to delete vertices three and ten times. In addition, we investigated deleting two vertices three times. We also investigated learning to delete edges three and ten times. Further, we looked at selecting three subgraphs on 20 vertices or edges. Finally, we

Table 2: Additional experimental results on large-scale regression datasets and comparison to non-data-driven ESAN [Bevilacqua et al., 2021].

(a) Results for the QM9 dataset

| | Operat. | Type | # | # Subg. | MAE $\downarrow$ |
|---|---|---|---|---|---|
| Baseline | | | | | 21.92 ±4.37 |
| Random
I-MLE | Delete | Vertex | 1 | 3 | 15.46 ±1.05
9.14 ±0.60 |
| Random
I-MLE | Delete | Vertex | 3 | 3 | 22.29 ±4.07
9.30 ±0.32 |
| Random
I-MLE | Delete | Vertex | 3 | 10 | 24.81 ±2.01
12.43 ±0.12 |
| Random
I-MLE | Delete | Vertex | 5 | 3 | 30.12 ±1.27
11.35 ±0.41 |
| Random
I-MLE | Select | Vertex | 10 | 3 | 22.69 ±3.05
11.88 ±0.52 |
| Random
I-MLE | Delete | Edge | 3 | 3 | 14.85 ±0.35
9.72 ±0.23 |
| Random
I-MLE | Delete | Edge | 5 | 3 | 13.69 ±0.28
10.08 ±0.36 |
| Random
I-MLE | Select | Edge | 10 | 3 | 14.02 ±0.99
11.58 ±0.46 |
| Random
I-MLE | Delete | 1-Ego | – | 3 | 22.20 ±3.01
21.19 ±1.38 |
| Random
I-MLE | Select | 3-Ego | – | 5 | 64.76 ±5.74
27.28 ±5.30 |
| Random
I-MLE | Select | 3-Ego | – | 10 | 19.64 ±1.38
14.93 ±0.83 |
| Random
I-MLE | Select | 5-Ego | – | 3 | 39.67 ±0.22
34.98 ±1.52 |

(b) I-MLE with ESAN on the ZINC dataset.

| Method | Operat. | Type | # | # Subg. | MAE $\downarrow$ | Time in s |
|---|---|---|---|---|---|---|
| ESAN
I-MLE
Random | Delete | Vertex | 1 | All Vertexs
2 | 0.171 ±0.010
0.177 ±0.016
0.214 ±0.007 | 11.86 ±0.110
3.449 ±0.082
2.910 ±0.071 |
| ESAN
I-MLE
Random | Delete | Edge | 1 | All edges
3 | 0.172 ±0.008
0.222 ±0.003
0.214 ±0.008 | 12.260 ±0.120
3.425 ±0.070
2.842 ±0.063 |
| ESAN
I-MLE
Random | Delete | Edge | 2 | All edges
3 | –
0.171 ±0.009
– | –
4.538 ±0.091
– |
| ESAN
I-MLE
Random | Select | 3-Ego | – | All 3-ego nets
10 | 0.126 ±0.006
0.181 ±0.010
0.188 ±0.004 | 6.825 ±0.021
3.907 ±0.015
4.502 ±0.043 |

looked at sampling three 7-hop neighborhood subgraphs. For the EXP dataset, we learned to delete three vertices or edges.

**Comparison to ESAN** To investigate how our data-driven sampling approaches compares to state-of-the-art architectures, we integrated I-MLE-based subgraph sampling into ESAN [Bevilacqua et al., 2021] (DS-GNNs)), and compared to ESAN using all subgraphs of a given size on the ZINC dataset. In addition, we also compared to a simple random model sampling subgraphs uniformly and at random, using the same configurations as the data-driven ones. To compare computation time between our method and ESAN, we measured the time on the test set. The timing consisted of data batch retrieval, subgraph sampling, downstream model forward propagation, upstream model forward propagation (in our method), and loss calculation. Like ESAN, we repeated the inference five times and voted for the majority.

**Additional experimental details** For ZINC, we used the subset of 12 000 graphs provided in [Dwivedi et al., 2020]. For ALCHEMY and QM9, we used a subset of 12 000 graphs sampled uniformly at random, we used the splits provided in [Morris et al., 2020b]. All of the above datasets consists of a training split of 10 000 graphs, and a validation and test split of 1 000 graphs, respectively. For the other datasets, we used the officially provided splits.

We repeated all experiments five times and report average scores and standard deviations. All experiments were conducted on a workstation with one GPU card with 32GB of GPU memory.

We used two separate instances of an Adam optimizers [Kingma and Ba, 2015] for the upstream and downstream models, both with default hyper-parameters and no weight decay. For the upstream model, we did not use learning rate decay. For the ZINC, ALCHEMY and QM9 datasets, we trained for at least 700 epochs and leveraged early stopping with a patience of 100 afterwards. The learning rate for the downstream model decays twice by 0.316 at the 400 and 600 epochs. For the EXP dataset, we trained for 350 epochs with a decay rate of 0.5 every 50 epochs, following the setup of ESAN [Bevilacqua et al., 2021]. For the OGBG-MOLESOL and OGBG-MOLBACE datasets, we trained for 100 epochs, following the default setting of [Hu et al., 2020]. For the PROTEINS dataset, we trained for 400 epochs, decaying the learning rate by 0.316 twice at 150 and 300 epoch.

**Additional experimental results** In addition to the results already shown in the main paper, we exhibit some additional results for different datasets and training settings. See Table 3 for results on PROTEINS. We sampled by deleting one and three vertices three times, and deleting three edges three times. See Table 4 for results on EXP. We examine the accuracy by selecting three subgraphs with one node or edge deletion. See Table 5 for results on ZINC using a GIN model. See Table 6 for results using PNA. See Table 7 for results on OGBG-MOLBACE. For ease of implementation, we use unordered subgraph aggregation by default. Here, we show results for additional experimental results comparing between ordered and

Table 3: Results for the PROTEINS dataset.

| Method | | | | ROCAUC ↑ |
|---|---|---|---|---|
| Baseline | | | | $0.775_{\pm 0.034}$ |
| GIN [Xu et al., 2019] | | | | $0.762_{\pm 0.028}$ |
| GIN + ID-GNN [You et al., 2021] | | | | $0.754_{\pm 0.027}$ |
| DropEdge [Rong et al., 2020] | | | | $0.735_{\pm 0.045}$ |
| PPGN [Maron et al., 2019] | | | | $0.772_{\pm 0.047}$ |
| CIN [Bodnar et al., 2021a] | | | | $0.770_{\pm 0.043}$ |
| OPERAT. | TYPE | # | # SUBG. | |
| Random Delete | Vertex | 1 | 3 | $0.760_{\pm 0.011}$ |
| I-MLE | | | | $0.775_{\pm 0.014}$ |
| Random Delete | Vertex | 3 | 3 | $0.769_{\pm 0.019}$ |
| I-MLE | | | | $0.783_{\pm 0.012}$ |
| Random Delete | Edge | 3 | 3 | $0.764_{0.024}$ |
| I-MLE | | | | $0.780_{0.013}$ |

Table 4: Results for the EXP dataset.

| Method | | | | Accuracy ↑ |
|---|---|---|---|---|
| Baseline | | | | $0.522_{\pm 0.003}$ |
| GIN [Xu et al., 2019] | | | | $0.511_{\pm 0.021}$ |
| GIN + ID-GNN [You et al., 2021] | | | | $1.000_{\pm 0.000}$ |
| OPERAT. | TYPE | # | # SUBG. | |
| Random Delete | Vertex | 1 | 3 | $0.943_{\pm 0.002}$ |
| I-MLE | | | | $1.000_{\pm 0.000}$ |
| Random Delete | Edge | 1 | 3 | $0.946_{\pm 0.002}$ |
| I-MLE | | | | $0.999_{\pm 0.001}$ |

unordered aggregation methods. Table 8 and Table 9 show results for OGBG datasets, while Table 10 and Table 11 show results for ZINC. To show the auxiliary loss described in Appendix C makes a difference, we carry out ablations studies using no auxiliary loss; see Table 12, Table 13 for results.

Finally, we designed more sophisticated subgraph selection methods. For vertices, we solve an Integer Linear Programming problem (ILP). The objective goal is to select subgraphs, maximizing the sum of the corresponding weights, while the constraints are that each vertex in the original graph much be selected at least once. We compare this combinatorial-optimization-based selection method with unordered I-MLE. For edges, we grow a Maximum Spanning Tree (MST) on each graph. We repeat this several times to get different subgraph instances. We compare this method to MST-based selection strategy using uniformly sampled edge weights; see Table 14 and Table 15 for results.

## D $k$-OSWL and $k$-OSAN: Omitted proofs

In the following, we outline the proofs from the main paper.

### D.1 Equivalence $k$-OSWL and $k$-OSAN

Let $\mathcal{G}$ be the set of all vertex-labeled graphs and $F$ be a set of permutation-invariant functions over $\mathcal{G}$, e.g., the functions expressible by some GNN architecture. Then, following Azizian and Lelarge [2020], we define an equivalence relation $\rho$ where for graphs $G$ and $H$ in $\mathcal{G}$

$$(G, H) \in \rho(F) \iff \text{for all } f \in F, f(G) = f(H)$$

holds. When $F$ is replaced by an architecture's name, we mean the set of function expressible with that architecture.

**Proposition 7** (Proposition 1 in the main text). For all $k \geq 1$, it holds that

$$\rho(k\text{-OSANs}) = \rho(k\text{-OSWL}).$$

To show the above result, we show the inclusions $\rho(k\text{-OSANs}) \subseteq \rho(k\text{-OSWL})$ and $\rho(k\text{-OSWL}) \subseteq \rho(k\text{-OSANs})$ in Lemma 8 and Lemma 9, respectively.

**Lemma 8.** For all $k \geq 1$, it holds that

$$\rho(k\text{-OSWL}) \subseteq \rho(k\text{-OSANs}).$$

Table 5: Results for the ZINC dataset with GIN model.

| Method | | | | MAE ↓ |
|---|---|---|---|---|
| PNA [Corso et al., 2020a] | | | | $0.188_{\pm 0.004}$ |
| Baseline | | | | $0.207_{\pm 0.006}$ |
| PNA [Corso et al., 2020b] | | | | $0.188_{\pm 0.004}$ |
| DGN [Beaini et al., 2020] | | | | $0.168_{\pm 0.003}$ |
| GIN [Xu et al., 2019] | | | | $0.252_{\pm 0.017}$ |
| HIMP [Fey et al., 2020] | | | | $0.151_{\pm 0.006}$ |
| GNS [Bouritsas et al., 2022] | | | | $0.108_{\pm 0.018}$ |
| CIN [Bodnar et al., 2021a] | | | | $0.094_{\pm 0.004}$ |
| | OPERAT. | TYPE | # | # SUBG. |
| Random I-MLE | Delete | Vertex | 1 | 1 | $0.378_{\pm 0.004}$ $0.287_{\pm 0.015}$ |
| Random I-MLE | Delete | Vertex | 1 | 3 | $0.283_{\pm 0.003}$ $0.194_{\pm 0.007}$ |
| Random I-MLE | Delete | Vertex | 1 | 10 | $0.234_{\pm 0.005}$ $0.217_{\pm 0.003}$ |
| Random I-MLE | Delete | Vertex | 3 | 3 | $0.265_{\pm 0.003}$ $0.184_{\pm 0.006}$ |
| Random I-MLE | Delete | Vertex | 3 | 10 | $0.275_{\pm 0.010}$ $0.240_{\pm 0.003}$ |
| Random I-MLE | Delete | Vertex | 10 | 10 | $0.210_{\pm 0.006}$ $0.204_{\pm 0.004}$ |
| Random I-MLE | Delete | Edge | 3 | 1 | $0.382_{\pm 0.004}$ $0.325_{\pm 0.019}$ |
| Random I-MLE | Delete | Edge | 3 | 3 | $0.192_{\pm 0.002}$ $0.176_{\pm 0.006}$ |
| Random I-MLE | Delete | Edge | 3 | 10 | $0.187_{\pm 0.002}$ $0.180_{\pm 0.006}$ |
| Random I-MLE | Delete | Edge | 10 | 3 | $0.173_{\pm 0.007}$ $0.162_{\pm 0.002}$ |
| Random I-MLE | Delete | Edge | 10 | 10 | $0.169_{\pm 0.013}$ $0.155_{\pm 0.004}$ |
| Random I-MLE | Select | Vertex | 20 | 3 | $0.384_{\pm 0.011}$ $0.313_{\pm 0.016}$ |
| Random I-MLE | Select | Edge | 20 | 3 | $0.274_{\pm 0.012}$ $0.261_{\pm 0.014}$ |
| Random I-MLE | Delete | 1-Ego | – | 3 | $0.330_{\pm 0.002}$ $0.208_{\pm 0.010}$ |
| Random I-MLE | Delete | 1-Ego | – | 10 | $0.285_{\pm 0.006}$ $0.260_{\pm 0.041}$ |
| Random I-MLE | Select | 7-Ego | – | 3 | $0.464_{\pm 0.023}$ $0.257_{\pm 0.004}$ |

Table 6: Results for the ZINC dataset with PNA model.

| Method | | | | MAE ↓ |
|---|---|---|---|---|
| PNA [Corso et al., 2020a] | | | | $0.188_{\pm 0.004}$ |
| GIN [Xu et al., 2019] | | | | $0.252_{\pm 0.017}$ |
| DGN [Beaini et al., 2020] | | | | $0.168_{\pm 0.003}$ |
| Baseline | | | | $0.174_{\pm 0.003}$ |
| | OPERAT. | TYPE | # | # SUBG. |
| Random I-MLE | Delete | Vertex | 1 | 3 | $0.260_{\pm 0.001}$ $0.168_{\pm 0.005}$ |
| Random I-MLE | Delete | Vertex | 1 | 10 | $0.227_{\pm 0.004}$ $0.154_{\pm 0.008}$ |
| Random I-MLE | Delete | Vertex | 3 | 3 | $0.226_{\pm 0.007}$ $0.172_{\pm 0.001}$ |
| Random I-MLE | Delete | Vertex | 3 | 10 | $0.255_{\pm 0.004}$ $0.164_{\pm 0.001}$ |
| Random I-MLE | Delete | Edge | 3 | 3 | $0.180_{\pm 0.007}$ $0.159_{\pm 0.008}$ |
| Random I-MLE | Delete | Edge | 10 | 3 | $0.174_{\pm 0.009}$ $0.161_{\pm 0.003}$ |
| Random I-MLE | Delete | 1-Ego | – | 3 | $0.325_{\pm 0.001}$ $0.167_{\pm 0.005}$ |

*Proof.* We show that if $k$-OSWL does not distinguish two vertices $v$ and $w$ in a graph $G$, then any $k$-OSAN will also not distinguish them. That is, any $k$-OSAN will compute the same feature for the two vertices, which implies the result.

Table 7: Results for the OGBG-MOLBACE dataset.

| Method | | | | | ROCAUC ↑ |
|---|---|---|---|---|---|
| Baseline | | | | | $0.714 \pm 0.058$ |
| | OPERAT. | TYPE | # | # SUBG. | |
| Random I-MLE | Delete | Vertex | 1 | 10 | $0.719 \pm 0.039$ $0.723 \pm 0.066$ |
| Random I-MLE | Delete | Vertex | 3 | 3 | $0.742 \pm 0.025$ $0.771 \pm 0.038$ |
| Random I-MLE | Delete | Vertex | 3 | 5 | $0.730 \pm 0.026$ $0.763 \pm 0.030$ |
| Random I-MLE | Delete | Vertex | 3 | 10 | $0.716 \pm 0.032$ $0.757 \pm 0.019$ |
| Random I-MLE | Delete | Vertex | 10 | 3 | $0.761 \pm 0.026$ $0.791 \pm 0.008$ |
| Random I-MLE | Delete | Edge | 1 | 3 | $0.724 \pm 0.056$ $0.735 \pm 0.046$ |
| Random I-MLE | Delete | Edge | 5 | 3 | $0.732 \pm 0.026$ $0.756 \pm 0.041$ |
| Random I-MLE | Delete | Edge | 10 | 3 | $0.772 \pm 0.028$ $0.777 \pm 0.024$ |
| Random I-MLE | Delete | Edge | 10 | 10 | $0.754 \pm 0.018$ $0.784 \pm 0.022$ |
| Random I-MLE | Delete | 1-Ego | – | 3 | $0.709 \pm 0.023$ $0.757 \pm 0.023$ |
| Random I-MLE | Select | 5-Ego | – | 3 | $0.768 \pm 0.039$ $0.777 \pm 0.027$ |

Table 8: Results for the OGBG-MOLESOL dataset using ordered and unordered subgraphs.

| Method | | | | | RSMSE ↓ |
|---|---|---|---|---|---|
| Baseline | | | | | $1.193 \pm 0.083$ |
| | OPERAT. | TYPE | # | # SUBG. | |
| Random I-MLE unordered I-MLE ordered | Delete | Vertex | 1 | 3 | $1.215 \pm 0.095$ $1.053 \pm 0.080$ $0.835 \pm 0.079$ |
| Random I-MLE unordered I-MLE ordered | Delete | Vertex | 2 | 3 | $1.132 \pm 0.020$ $1.081 \pm 0.021$ $0.850 \pm 0.106$ |
| Random I-MLE unordered I-MLE ordered | Delete | Vertex | 5 | 3 | $0.992 \pm 0.115$ $1.115 \pm 0.076$ $0.853 \pm 0.043$ |

Table 9: Results for the OGBG-MOLBACE dataset using ordered and unordered subgraphs.

| Method | | | | | AUCROC ↑ |
|---|---|---|---|---|---|
| Baseline | | | | | $0.714 \pm 0.058$ |
| | OPERAT. | TYPE | # | # SUBG. | |
| Random I-MLE unordered I-MLE ordered | Delete | Vertex | 3 | 3 | $0.742 \pm 0.025$ $0.771 \pm 0.038$ $0.761 \pm 0.011$ |
| Random I-MLE unordered I-MLE ordered | Delete | Vertex | 3 | 10 | $0.716 \pm 0.032$ $0.757 \pm 0.019$ $0.776 \pm 0.032$ |

Table 10: Results for the ZINC dataset using ordered and unordered subgraphs.

| Method | | | | | MAE ↓ |
|---|---|---|---|---|---|
| Baseline | | | | | $0.207 \pm 0.006$ |
| | OPERAT. | TYPE | # | # SUBG. | |
| Random I-MLE unordered I-MLE ordered | Delete | Vertex | 1 | 3 | $0.283 \pm 0.003$ $0.194 \pm 0.007$ $0.187 \pm 0.004$ |

Let us make precise what we will show. Let $v$ and $w \in V(G)$ such that $C(v) = C(w)$. We recall that $C(v) := \mathsf{RELABEL}\big(\{\!\{C_\infty(v, \mathbf{g}) \mid \mathbf{g} \in G_k\}\!\}\big)$ and $C(w) := \mathsf{RELABEL}\big(\{\!\{C_\infty(w, \mathbf{g}) \mid \mathbf{g} \in G_k\}\!\}\big)$. For $C(v) = C(w)$ to hold, we therefore need that

$$C_i(v) := \{\!\{C_i(v, \mathbf{g}) \mid \mathbf{g} \in G_k\}\!\} = \{\!\{C_i(w, \mathbf{g}) \mid \mathbf{g} \in G_k\}\!\} =: C_i(w) \qquad (6)$$

for all iterations $i$ of the $k$-OSWL.

Table 11: Results for the ZINC dataset using ordered and unordered subgraphs with PNA model.

| Method | | | | | MAE ↓ |
|---|---|---|---|---|---|
| | OPERAT. | TYPE | # | # SUBG. | |
| Random | | | | | $0.260 \pm 0.001$ |
| I-MLE unordered | Delete | Vertex | 1 | 3 | $0.168 \pm 0.005$ |
| I-MLE ordered | | | | | $0.182 \pm 0.005$ |
| Random | | | | | $0.227 \pm 0.004$ |
| I-MLE unordered | Delete | Vertex | 1 | 10 | $0.154 \pm 0.008$ |
| I-MLE ordered | | | | | $0.181 \pm 0.010$ |
| Random | | | | | $0.226 \pm 0.007$ |
| I-MLE unordered | Delete | Vertex | 3 | 3 | $0.172 \pm 0.008$ |
| I-MLE ordered | | | | | $0.186 \pm 0.003$ |
| Random | | | | | $0.255 \pm 0.004$ |
| I-MLE unordered | Delete | Vertex | 3 | 10 | $0.164 \pm 0.001$ |
| I-MLE ordered | | | | | $0.175 \pm 0.008$ |

Table 12: Results for the OGBG-MOLESOL dataset, auxiliary loss ablation.

| Method | | | | | RSMSE ↓ |
|---|---|---|---|---|---|
| Baseline | | | | | $1.193 \pm 0.083$ |
| | OPERAT. | TYPE | # | # SUBG. | |
| Random | | | | | $1.215 \pm 0.095$ |
| I-MLE | Delete | Vertex | 1 | 3 | $1.053 \pm 0.080$ |
| I-MLE ablation | | | | | $1.120 \pm 0.092$ |
| Random | | | | | $1.132 \pm 0.020$ |
| I-MLE | Delete | Vertex | 2 | 3 | $1.081 \pm 0.021$ |
| I-MLE ablation | | | | | $1.137 \pm 0.146$ |
| Random | | | | | $0.992 \pm 0.115$ |
| I-MLE | Delete | Vertex | 5 | 3 | $1.115 \pm 0.076$ |
| I-MLE ablation | | | | | $1.247 \pm 0.126$ |

Table 13: Results for the ZINC dataset, auxiliary loss ablation.

| Method | | | | | MAE ↓ |
|---|---|---|---|---|---|
| Baseline | | | | | $0.207 \pm 0.006$ |
| | OPERAT. | TYPE | # | # SUBG. | |
| Random | | | | | $0.283 \pm 0.003$ |
| I-MLE | Delete | Vertex | 1 | 3 | $0.194 \pm 0.007$ |
| I-MLE ablation | | | | | $0.194 \pm 0.004$ |
| Random | | | | | $0.265 \pm 0.003$ |
| I-MLE | Delete | Vertex | 3 | 3 | $0.184 \pm 0.006$ |
| I-MLE ablation | | | | | $0.184 \pm 0.004$ |
| Random | | | | | $0.192 \pm 0.002$ |
| I-MLE | Delete | Edge | 3 | 3 | $0.176 \pm 0.006$ |
| I-MLE ablation | | | | | $0.178 \pm 0.008$ |
| Random | | | | | $0.169 \pm 0.013$ |
| I-MLE | Delete | Edge | 10 | 10 | $0.155 \pm 0.004$ |
| I-MLE ablation | | | | | $0.162 \pm 0.001$ |

Table 14: Results for the OGBG-MOLESOL dataset using different selection methods.

| Method | | | | | RSMSE ↓ |
|---|---|---|---|---|---|
| Baseline | | | | | $1.193 \pm 0.083$ |
| | OPERAT. | TYPE | # | # SUBG. | |
| Random | | | | | $1.215 \pm 0.095$ |
| I-MLE unordered | Delete | Vertex | 1 | 3 | $1.053 \pm 0.080$ |
| I-MLE covered | | | | | $1.074 \pm 0.115$ |
| Random | | | | | $1.132 \pm 0.020$ |
| I-MLE unordered | Delete | Vertex | 2 | 3 | $1.081 \pm 0.021$ |
| I-MLE covered | | | | | $1.081 \pm 0.068$ |
| Random | | | | | $0.992 \pm 0.115$ |
| I-MLE unordered | Delete | Vertex | 5 | 3 | $1.115 \pm 0.076$ |
| I-MLE covered | | | | | $0.946 \pm 0.058$ |
| Random | | | | | $1.095 \pm 0.021$ |
| I-MLE | MST | Edge | – | 3 | $1.070 \pm 0.005$ |

We next turn to $k$-OSANs. Let us denote by $\mathbf{h}_{v,\mathbf{g}}^{(i)}$ the vertex feature of $v$ for $\mathbf{g} \in G_k$ computed in layer $i$ of a $k$-OSAN. We define

$$\mathbf{h}_v^{(i)} := \mathsf{SAGG}\big(\{\!\{\mathbf{h}_{v,\mathbf{g}}^{(i)} \mid \mathbf{g} \in G_k \text{ s.t. } \boldsymbol{\pi}_{v,\mathbf{g}} \neq \mathbf{0}\}\!\}\big).$$

We now show

$$C_i(v) = C_i(w) \implies \mathbf{h}_v^{(i)} = \mathbf{h}_w^{(i)}, \tag{7}$$

for $i \geq 0$. To do so, we first show the following result.

Table 15: Results for the OGBG-MOLBACE dataset using different selection methods.

| Method | | | | | AUCROC ↑ |
|---|---|---|---|---|---|
| Baseline | | | | | $0.714_{\pm 0.058}$ |
| | OPERAT. | TYPE | # | # SUBG. | |
| Random | | | | | $0.742_{\pm 0.025}$ |
| I-MLE unordered | Delete | Vertex | 3 | 3 | $0.771_{\pm 0.038}$ |
| I-MLE covered | | | | | $0.765_{\pm 0.032}$ |
| Random | MST | Edge | – | 3 | $0.740_{\pm 0.034}$ |
| I-MLE | | | | | $0.758_{\pm 0.025}$ |
| Random | MST | Edge | – | 10 | $0.741_{\pm 0.025}$ |
| I-MLE | | | | | $0.763_{\pm 0.027}$ |

Table 16: Dataset statistics and properties for graph-level prediction tasks, $^\dagger$—Continuous vertex labels following Gilmer et al. [2017], the last three components encode 3D coordinates.

| Dataset | Properties | | | | | |
|---|---|---|---|---|---|---|
| | Number of graphs | Number of classes/targets | ∅ Number of vertices | ∅ Number of edges | Vertex labels | Edge labels |
| ALCHEMY | 202 579 | 12 | 10.1 | 10.4 | ✓ | ✓ |
| QM9 | 129 433 | 12 | 18.0 | 18.6 | ✓(13+3D)$^\dagger$ | ✓(4) |
| ZINC | 249 456 | 1 | 23.1 | 24.9 | ✓ | ✓ |
| EXP | 1 200 | 2 | 44.5 | 55.2 | ✓ | ✗ |
| OGBG-MOLESOL | 1 128 | 1 | 13.3 | 13.7 | ✓ | ✓ |
| OGBG-MOLBACE | 1 513 | 2 | 34.1 | 36.9 | ✓ | ✓ |
| PROTEINS | 1 113 | 2 | 39.1 | 72.8 | ✓ | ✗ |

**Claim 1.** *It holds that*

$$C_i(v, \mathbf{g}) = C_i(w, \mathbf{g}') \implies \mathbf{h}_{v,\mathbf{g}}^{(i)} = \mathbf{h}_{w,\mathbf{g}'}^{(i)} \text{ and } \boldsymbol{\pi}_{v,\mathbf{g}} = \boldsymbol{\pi}_{w,\mathbf{g}'} \tag{8}$$

*for all $\mathbf{g}$ and $\mathbf{g}' \in G_k$ and $i \geq 0$.*

*Proof.* We proof the result by induction on the number of iterations or layers $i$. The base case, $i = 0$, follows by definition of the initial coloring of the $k$-OSWL and the initial features of $k$-OSANs, that is, both are dictated solely by the atomic type. The same holds for $\boldsymbol{\pi}_{v,\mathbf{g}}$ and $\boldsymbol{\pi}_{w,\mathbf{g}'}$, which remain unchanged for layers $i > 0$.

Assume Equation (8) holds for the first $i$ iteration and further assume $C_{i+1}(v, \mathbf{g}) = C_{i+1}(w, \mathbf{g}')$ holds. Hence, $C_i(v, \mathbf{g}) = C_i(w, \mathbf{g}')$ and $\{\!\{C_i(u, \mathbf{g}) \mid u \in \square\}\!\} = \{\!\{C_i(u, \mathbf{g}') \mid u \in \square\}\!\}$. We now define the multi-sets

$$M_{v,\mathbf{g}}^{(i+1)} := \{\!\{\mathbf{h}_{u,\mathbf{g}}^{(i)} \mid u \in \square\}\!\} \quad \text{and} \quad M_{w,\mathbf{g}'}^{(i+1)} := \{\!\{\mathbf{h}_{u,\mathbf{g}'}^{(i)} \mid u \in \square\}\!\}.$$

By the above, we know that $M_{v,\mathbf{g}}^{(i+1)} = M_{w,\mathbf{g}'}^{(i+1)}$ and $\mathbf{h}_{v,\mathbf{g}}^{(i)} = \mathbf{h}_{w,\mathbf{g}'}^{(i)}$. Therefore, regardless of the concrete choice of $\mathsf{UPD}^{(i+1)}$ and $\mathsf{AGG}^{(i+1)}$, $\mathbf{h}_{v,\mathbf{g}}^{(i+1)} = \mathbf{h}_{w,\mathbf{g}'}^{(i+1)}$. $\square$

We are now ready to show Equation (7). Hence, we assume $C_i(v, \mathbf{g}) = C_i(v, \mathbf{g}')$ for $i \geq 0$ holds. Hence, by assumption, the two multisets of Equation (6) are (element-wise) equal. Hence, there exists a bijection $\theta \colon \{(v, \mathbf{g}) \mid \mathbf{g} \in G_k\} \to \{(w, \mathbf{g}) \mid \mathbf{g} \in G_k\}$ such that $C_0(v, \mathbf{g}) = C_0(\theta(v, \mathbf{g}))$. By leveraging Claim 1, we now construct a bijection $\varphi$ with the same domain and co-domain as $\theta$ such that $\mathbf{h}_{v,\mathbf{g}}^{(i)} = \mathbf{h}_{\varphi(v,\mathbf{g})}^{(i)}$, implying Equation (7).

We construct the bijection $\varphi$ as follows. Take $(v, \mathbf{g}) \in V(G) \times G_k$, let $\theta(v, \mathbf{g}) = (w, \mathbf{g}')$, and set $\varphi(v, \mathbf{g}) = \theta(v, \mathbf{g}) = (w, \mathbf{g}')$. Since $C_i(v, \mathbf{g}) = C_i(\theta(v, \mathbf{g}))$, Claim 1 implies that $\mathbf{h}_{v,\mathbf{g}}^{(i)} = \mathbf{h}_{\varphi(v,\mathbf{g})}^{(i)}$ and $\boldsymbol{\pi}_{v,\mathbf{g}} = \boldsymbol{\pi}_{\varphi(v,\mathbf{g})}$. Hence, by the existence of the bijection $\varphi$, we have that

$$\{\!\{\mathbf{h}_{v,\mathbf{g}}^{(i)} \mid \mathbf{g} \in G_k \text{ s.t. } \boldsymbol{\pi}_{v,\mathbf{g}} \neq \mathbf{0}\}\!\} = \{\!\{\mathbf{h}_{w,\mathbf{g}}^{(i)} \mid \mathbf{g} \in G_k \text{ s.t. } \boldsymbol{\pi}_{w,\mathbf{g}} \neq \mathbf{0}\}\!\}.$$

Hence, the feature vector $\mathbf{h}_v$ is equal to $\mathbf{h}_w$. $\square$

**Lemma 9.** *For all $k \geq 1$, it holds that*

$$\rho(k\text{-OSANs}) \subseteq \rho(k\text{-OSWL}).$$

*Proof.* We argue that there exists a canonical $k$-PMPMN that can simulate the $k$-OSWL. By setting UPD to the identity function and a constant, non-zero function, we can simulate the initial labeling of the

$k$-OSWL. For the other iterations, we need to show that there exist instances of UPD$^{(i)}$, AGG$^{(i)}$ for $i > 0$, and SAGG that are injective, faithfully distinguishing non-equal multisets. The existence of such instances follows directly from the proof of Theorem 2 in [Morris et al., 2019]. $\qquad\square$

The above two lemmas directly imply Proposition 1.

## D.2 Separation results

**Theorem 10** (Theorem 3 in the main text). *For all $k \geq 1$ it holds that*

$$\rho(k + 1\text{-OSANs}) \subsetneq \rho(k\text{-OSANs}).$$

*Proof.* First, $\rho(k + 1\text{-OSANs}) \subseteq \rho((k)\text{-OSANs})$, is a direct consequence of the results in [Geerts and Reutter, 2022] showing that $(k + 1)$-MPNNs are bounded by $(k + 1)$-WL and that $k$-OSANs are a restricted class of $(k + 1)$-MPNNs. The strictness follows by Proposition 1 and Theorem 14. $\qquad\square$

**Construction of Fürer grid-graphs** We restate the following construction due to Fürer [2001]. Let $h$ and $n$ be fixed positive integers such that $n \gg h + 1$. Fix a *global* graph $G_n^h$, defined to be a $h \times n$ grid graph. Define a graph $X(G_n^h)$ as follows.

1. For each vertex $v \in V(G_n^h)$,
   - let degree of $v$ be $d$,
   - let $E_v$ be the set of edges incident to $v$,
   - replace $v$ by a *vertex cloud* $C_v$ of $2^{d-1}$ vertices of the form $(v, S)$ such that $S$ is an even subset of $E_v$.

2. For each edge $e = \{u, v\} \in E(G_n^h)$,
   - for each $(u, S) \in C_u$ and $(v, T) \in C_v$, add an edge between $(u, S)$ and $(v, T)$ if
     - both $S$ and $T$ contain $e$, or
     - both $S$ and $T$ do not contain $e$.

The graph $Y(G_n^h)$ is defined exactly as $X(G_n^h)$, with the following exception. Fix an edge $\{u^*, v^*\}$ of the global graph $G_n^h$. In the second step above (Item 2), we use a different rule for this edge $\{u^*, v^*\}$,

- for each $(u^*, S) \in C_{u^*}$ and $(v^*, T) \in C_{v^*}$, add an edge between $(u^*, S)$ and $(v^*, T)$ if
  - exactly one out of $S$ and $T$ contains $e$

The edge $\{u^*, v^*\}$ is said to be *twisted*. Equivalently, $Y(G_n^h)$ is the graph obtained from the graph $X(G_n^h)$ by performing a bipartite-complement operation on the bipartite graph between the vertex clouds $C_{u^*}$ and $C_{v^*}$.

For a vertex $v$ in $X_k$ or $Y_k$, let $\bar{v}$ denote the vertex $x$ in $G_n^h$ such that $v \in C_x$ (also called a meta-vertex in [Fürer, 2001]). We assign a fresh color say $c_v$ to the the vertex cloud $C_v$ for each $v \in V(G_n^h)$, imposing an initial coloring on the graphs $X(G_n^h)$ and $Y(G_n^h)$. It is easy to see that this coloring is stable under Color Refinement.

**Construction and Properties of $X_k$ and $Y_k$** To ease notation, we set $B = G_n^{k+1}$ as our *base graph* where $n \gg k + 1$. For $k \in \mathbb{N}$, we set $X_k = X(B)$ and $Y_k = Y(B)$. Fürer showed that the graphs $X_k$ and $Y_k$ are non-isomorphic yet $k$-WL-indistinguishable. It was also shown that $(k + 1)$-WL can distinguish these graphs after at least $n$ rounds. The proof technique relies on *trapping the twist* using $k + 2$ pebbles in a Spoiler-Duplicator game [Immerman and Lander, 1990].

Moreover, let $Z$ be a graph obtained by twisting some $\ell$ distinct edges of $B$, similar to how $Y_k$ is obtained from $B$ after a single twist. Then, it was shown that $Z$ is isomorphic to $X_k$ if $\ell$ is even, and $Z$ is isomorphic to $Y_k$ if $\ell$ is odd.

**Twists and Shields**  Let $u \in V(X_k)$. Let $\boldsymbol{u}$ in $V(X_k)^k$. Let $B \backslash (\bar{\boldsymbol{u}}, \bar{u})$ denote the graph obtained by deleting the vertices in $(\bar{\boldsymbol{u}}, \bar{u})$ in the base graph $B$. Let $e = (x, y)$ be the edge of $B$ which was twisted to obtain $Y_k$. Assume that at least one of its endpoints of $e$ is not in $(\bar{\boldsymbol{u}}, \bar{u})$. The *twisted component* of $B$ w.r.t $(\bar{\boldsymbol{u}}, \bar{u})$, denoted by $\mathsf{TC}(\bar{\boldsymbol{u}}, \bar{u})$, is the unique component of $B \backslash (\bar{\boldsymbol{u}}, \bar{u})$ which contains the twisted edge.

Let $N_{\mathsf{TC}}[\bar{\boldsymbol{u}}, \bar{u}]$ be the neighborhood of vertices in $(\boldsymbol{u}, u)$ into the twisted component, i.e., the set of vertices $v \in \mathsf{TC}(\bar{\boldsymbol{u}}, \bar{u})$ which are incident to $(\bar{\boldsymbol{u}}, \bar{u})$. Then, a twisted component is a *shield* if it satisfies the following two properties:

- the twisted edge is not incident to any of the vertices in $(\bar{\boldsymbol{u}}, \bar{u})$ and $N_{\mathsf{TC}}[\bar{\boldsymbol{u}}, \bar{u}]$, and
- the twisted edge lies on some cycle in $\mathsf{TC}(\bar{\boldsymbol{u}}, \bar{u}) \backslash N_{\mathsf{TC}}[\bar{\boldsymbol{u}}, \bar{u}]$.

In this case, we also call $(\boldsymbol{u}, u)$ to be *shielding* for $Y_k$. The motivation behind these conditions is as follows. The first condition ensures that the twist is at distance at least two from the individualized vertices. The second condition ensures that the twist cannot be trapped using just two pebbles.

**Proposition 11.** Suppose that $(\boldsymbol{u}, u)$ is shielding for $Y_k$. If we run color refinement on the disjoint union of $(X_k, \boldsymbol{u})$ and $(Y_k, \boldsymbol{u})$, the stable color of $u$ in $(X_k, \boldsymbol{u})$ is identical to the stable color of $u$ in $(Y_k, \boldsymbol{u})$.

*Proof.* Since $(\boldsymbol{u}, u)$ is shielding for $Y_k$, the twisted edge lies on a cycle $C$ inside $\mathsf{TC}(\bar{\boldsymbol{u}}, \bar{u}) \backslash N_{\mathsf{TC}}[\bar{\boldsymbol{u}}, \bar{u}]$. Hence, every vertex of $C$ is at distance at least two from the vertices in $(\boldsymbol{u}, u)$. We invoke the usual Spoiler-Duplicator games of Immerman-Lander to argue the desired claim [Cai et al., 1992].

We show that a Duplicator can always move around the twist such that it is never caught by the Spoiler. This game uses $k$ pairs of fixed pebbles corresponding to $\boldsymbol{u}$ in each graph, and two pairs of movable pebbles which are placed on $u$ in $X_k$ and $Y_k$ respectively. Recall that color refinement can be simulated using a 2-pebble Spoiler-Duplicator game [Immerman and Lander, 1990]. Since the $k$ fixed pebbles are influential only in their neighbourhood, the Duplicator strategy is to move the twist around in the cycle $C$, so that the twist is always at a distance of at least two from the fixed pebbles. This renders the fixed pebbles useless for the Spoiler. Since there are only two movable pebbles, the Duplicator can always move the twist around in the cycle $C$ and hence avoid a situation where the Spoiler can trap the twist with the two movable pebbles. $\square$

**Shielding Twists**  Let $u \in V(X_k)$ and $\boldsymbol{u} \in V(X_k)^k$. Next we show that if $(\boldsymbol{u}, u)$ is not shielding for $Y_k$, we can do a series of twisting operations on the graph $Y_k$ to obtain an isomorphic graph $Y_k'$ such that $(\boldsymbol{u}, u)$ is shielding for $Y_k'$.

**Proposition 12.** If $(\boldsymbol{u}, u)$ is not shielding for $Y_k$, there exists $\boldsymbol{v} \in V(X_k)^k$ such that $(\boldsymbol{v}, u)$ is shielding for $Y_k$. Hence, if we run Color Refinement on the disjoint union of $(X_k, \boldsymbol{u})$ and $(Y_k, \boldsymbol{v})$, the stable color of $u$ in $(X_k, \boldsymbol{u})$ is identical to the stable color of $u$ in $(Y_k, \boldsymbol{v})$.

*Proof.* Since $n \gg k$, there exists at least one component in $B \backslash (\bar{\boldsymbol{u}}, \bar{u})$ such that it contains a grid $G_{3 \times 3}$ of dimension $3 \times 3$ as an induced subgraph, where $G_{3 \times 3}$ does not have any edges to $(\boldsymbol{u}, u)$. Let $C^*$ be the lexicographically least such component in $B \backslash (\bar{\boldsymbol{u}}, \bar{u})$. Our goal is to use an automorphism $\theta$ of $Y_k$ to transfer the twist to this grid $G_{3 \times 3}$ inside the component $C^*$ such that $\theta$ *fixes* $u$, i.e. $\theta(u) = u$. This would mean that $(\boldsymbol{u}, u)$ is shielding for $Y_k^\theta$ with $C^*$ as the shield. Hence, we set $\boldsymbol{v} = \boldsymbol{u}^{\theta^{-1}}$ so that $(\boldsymbol{v}, u)$ is shielding for $Y_k$.

To achieve this transformation, for every $\bar{\boldsymbol{u}} \in V(B)^k$, we fix a shortest path $P^{\bar{\boldsymbol{u}}}$ from one of the ends of the twisted edge to the central vertex of the grid $G_{3,3}$ such that $P$ avoids $u$. We twist all the edges on the path $P^{\bar{\boldsymbol{u}}}$. If the length of the path $P$ is odd, we twist one more edge in $G_{3,3}$ so as to ensure that $P^{\bar{\boldsymbol{u}}}$ has even length. The resulting graph $Y_k'$ is isomorphic to $Y_k$ via a unique isomorphism $\theta$. Since the path $P$ avoids $u$, the isomorphism $\theta$ fixes $u$. Hence, $(\boldsymbol{u}, u)$ is shielding for $Y_k'$, and therefore $(\boldsymbol{u}^{\theta^{-1}}, u)$ is shielding for $Y_k$. Hence, proved. $\square$

Observe that the association $\boldsymbol{u} \mapsto \boldsymbol{v}$ in the proof of the above claim is bijective, as follows. Suppose there exists $\boldsymbol{w} \mapsto \boldsymbol{v}$ such that $\boldsymbol{u} \neq \boldsymbol{w}$. Now, $\boldsymbol{u}$ and $\boldsymbol{w}$ must have same initial color type, since the used isomorphisms preserve vertex clouds, i.e. $\bar{\boldsymbol{u}} = \bar{\boldsymbol{w}}$. Hence, the same path $P^{\boldsymbol{u}}$ is used for both $\boldsymbol{u}$ and $\boldsymbol{v}$ in the base graph $B$. For a fixed path $P^{\boldsymbol{u}}$ of even length, there is a unique isomorphism $\theta$ which twists all the edges in $P$ to yield the graph $Y_k'$. Hence, it must be the case that $\boldsymbol{u} = \boldsymbol{w} = \boldsymbol{v}^{\theta^{-1}}$.

**Lemma 13.** For $k \in \mathbb{N}$, $k$-OSWL cannot distinguish graphs $X_k$ and $Y_k$.

*Proof.* Let $\mathcal{X}$ denote the disjoint union of graphs $(X_k, \boldsymbol{u})$, $\boldsymbol{u} \in V(X_k)^k$. Let $\mathcal{Y}$ denote the disjoint union of graphs $(Y_k, \boldsymbol{v})$, $\boldsymbol{v} \in V(Y_k)^k$. It suffices to show the equality of the following nested multisets

$$\{\!\{\{\!\{\mathsf{CR}(\mathcal{X}, u^{\boldsymbol{u}}) \,|\, \boldsymbol{u} \in V(X_k)^k\}\!\} \,|\, u \in V(X_k)\}\!\} = \{\!\{\{\!\{\mathsf{CR}(\mathcal{Y}, v^{\boldsymbol{v}}) \,|\, \boldsymbol{v} \in V(Y_k)^k\}\!\} \,|\, v \in V(Y_k)\}\!\},$$

where $u^{\boldsymbol{u}}$ denotes the vertex $u$ in the constituent $(X_k, \boldsymbol{u})$ of $\mathcal{X}$. Similarly, $v^{\boldsymbol{v}}$ denotes the vertex $v$ in the constituent $(X_k, \boldsymbol{v})$ of $\mathcal{Y}$.

Observe that the graphs $X_k$ and $Y_k$ have the same vertex set. We claim that for every $u \in V(X_k)$, the corresponding vertex $u \in V(Y_k)$ satisfies

$$\{\!\{\mathsf{CR}(\mathcal{X}, u^{\boldsymbol{u}}) \,|\, \boldsymbol{u} \in V(X_k)^k\}\!\} = \{\!\{\mathsf{CR}(\mathcal{Y}, u^{\boldsymbol{v}}) \,|\, \boldsymbol{v} \in V(Y_k)^k\}\!\}.$$

Indeed, this follows immediately from Proposition 11 and Proposition 12 along with the fact that the association in Proposition 12 is bijective (see the discussion subsequent to Proposition 12). Hence, proved. $\quad\square$

**Theorem 14.** For $k \in \mathbb{N}$, there exist graphs $X_k$ and $Y_k$ such that they are distinguishable by $(k+1)$-WL but not distinguishable by $k$-OSWL.

*Proof.* Immediate from Lemma 13. $\quad\square$

Next we compare the expressive power of $k$-WL and $k$-OSWL.

**Lemma 15.** For $k \in \mathbb{N}$, there exist graphs $X_k$ and $Y_k$ such that they are distinguishable by $k$-OSWL but not distinguishable by $k$-WL.

*Proof.* We set $X_k$ and $Y_k$ to be CFI-gadgets $G_{k+1}$ and $H_{k+1}$ which are known to be indistinguishable by $k$-WL (see Section E.2 for the definition of these gadgets). It remains to show that they can be distinguished by $k$-OSWL. Recall that $X_k$ contains a colorful distance-two-clique $Q$ of size $k+2$ while $Y_k$ does not contain such an object. We place $k$ fixed pebbles on some $k$ vertices of $Q$, and let $x, y$ be the remaining two vertices in $Q$. It is clear that upon two rounds of color refinement, the vertices $x$ and $y$ see all individualized colors corresponding to the fixed pebbles. Moreover, the individualized pebbles also see all the individualized colors of other pebbles.

On the other hand, doing such an operation on $Y_k$ will never yield such colors, since this would otherwise ensure a colorful distance-two-clique in $Y_k$. Hence, there does not exist any $x' \in V(Y_k)$ and $\boldsymbol{v} \in V(Y_k)^k$ such that color refinement on the disjoint union of $(X_k, \boldsymbol{u})$ and $(Y_k, \boldsymbol{v})$ yields the same colors for $x$ and $x'$. Therefore for any choice of $x' \in V(Y_k)$ it holds that the following multisets for vertices $x \in V(X_k)$ and $x'$, obtained by aggregation over all ordered subgraphs, satisfy

$$\{\!\{\mathsf{CR}(\mathcal{X}, x^{\boldsymbol{u}}) \,|\, \boldsymbol{u} \in V(X_k)^k\}\!\} \neq \{\!\{\mathsf{CR}(\mathcal{Y}, (x')^{\boldsymbol{v}}) \,|\, \boldsymbol{v} \in V(Y_k)^k\}\!\}.$$

which implies that the aggregated multisets over all vertices

$$\{\!\{\{\!\{\mathsf{CR}(\mathcal{X}, u^{\boldsymbol{u}}) \,|\, \boldsymbol{u} \in V(X_k)^k\}\!\} \,|\, u \in V(X_k)\}\!\} \neq \{\!\{\{\!\{\mathsf{CR}(\mathcal{Y}, v^{\boldsymbol{v}}) \,|\, \boldsymbol{v} \in V(Y_k)^k\}\!\} \,|\, v \in V(Y_k)\}\!\}.$$

Hence, $k$-OSWL distinguishes $X_k$ and $Y_k$. $\quad\square$

The following theorem shows that the algorithms $k$-OSWL, $k \in \mathbb{N}$, form a hierarchy of increasingly powerful isomorphism tests.

**Theorem 16.** For $k \in \mathbb{N}$, $k$-OSWL has strictly less expressive power than $(k+1)$-OSWL.

*Proof.* The proof follows immediately from Theorem 14 and Lemma 15; see below. $\quad\square$

# E  Vertex-subgraph $k$-OSWL and $k$-OSAN: Omitted Proofs

In this section we consider a variant of $k$-OSWL, denoted vertex-subgraph $k$-OSWL, in which the construction of the multi-sets used to define the color of graph is defined differently. As before, we define $C_i(v, \mathbf{g})$ and $C_\infty(v, \mathbf{g})$ for $v \in V(G)$ and $\mathbf{g} \in G_k$. Then, instead of computing a single color for a vertex $v$, we compute a single color for $\mathbf{g} \in G_k$. We do this by aggregating over all vertex in $G$, i.e, we compute

$$C(\mathbf{g}) := \mathsf{RELABEL}\big(\{\!\{C_\infty(v, \mathbf{g}) \mid v \in V(G)\}\!\}\big).$$

Finally, we use

$$\mathsf{RELABEL}\big(\{\!\{C(\mathbf{g}) \mid \mathbf{g} \in G_k\}\!\}\big)$$

to obtain the color $C(G)$ of $G$. The neural counterpart, vertex-subgraph $k$-OSANs, are defined in a similar way. That is, $\mathbf{h}_{v,\mathbf{g}}^{(i)}$ is defined as for $k$-OSANs but we now define

$$\mathbf{h}_{\mathbf{g}}^{(T)} := \mathsf{AGG}\big(\{\!\{\mathbf{h}_{v,\mathbf{g}}^{(T)} \mid v \in V(G)\}\!\}\big)$$
$$\mathbf{h}_G := \mathsf{SAGG}\big(\{\!\{\mathbf{h}_{\mathbf{g}}^{(T)} \mid \mathbf{g} \in V(G)^k, v \in V(G)\, \boldsymbol{\pi}_{v,\mathbf{g}} \neq \mathbf{0}\}\!\}\big).$$

Again, AGG and SAGG are differentiable, parameterized functions, e.g., neural networks.

## E.1  Equivalence of vertex-subgraph $k$-OSWL and vertex-subgraph $k$-OSAN

**Proposition 17.** For all $k \geq 1$, vertex-subgraph $k$-OSANs and vertex-subgraph $k$-OSWL have the same distinguishing power.

The proof consists in showing that (i) vertex-subgraph $k$-OSANs cannot distinguish more graphs than vertex-subgraph $k$-OSWL (Lemma 18); and (ii) vertex-subgraph $k$-OSWL cannot distinguish more graphs than vertex-subgraph $k$-OSANs (Lemma 19).

**Lemma 18.** For all $k \geq 1$, it holds that $\rho(k\text{-OSWL}) \subseteq \rho(k\text{-OSANs})$.

*Proof.* Consider graphs $G$ and $H$ in $\rho(k\text{-OSWL})$. By definition, this implies that

$$\mathsf{RELABEL}\big(\{\!\{C(\mathbf{g}) \mid \mathbf{g} \in G_k\}\!\}\big) = \mathsf{RELABEL}\big(\{\!\{C(\mathbf{q}) \mid \mathbf{q} \in H_k\}\!\}\big) \tag{9}$$

holds. We next show $C(\mathbf{g}) = C(\mathbf{q})$ for $\mathbf{g} \in G_k$ and $\mathbf{q} \in H_k$ implies that any $k$-OSAN computes the same features for $k$-ordered subgraphs $\mathbf{g}$ and $\mathbf{q}$. Combined with Equation (9) this implies that any $k$-OSAN assigns the same feature to $G$ and $H$. Assume $C(\mathbf{g}) = C(\mathbf{q})$, hence $C(\mathbf{g}) := \mathsf{RELABEL}\big(\{\!\{C_\infty(v, \mathbf{g}) \mid v \in V(G)\}\!\}\big)$ and $C(\mathbf{q}) := \mathsf{RELABEL}\big(\{\!\{C_\infty(w, \mathbf{q}) \mid w \in V(H)\}\!\}\big)$. Hence, for $C(\mathbf{g}) = C(\mathbf{q})$ to hold, we need that

$$C_i(\mathbf{g}) := \{\!\{C_i(v, \mathbf{g}) \mid v \in V(G)\}\!\} = \{\!\{C_i(w, \mathbf{q}) \mid w \in V(H)\}\!\} =: C_i(\mathbf{q}) \tag{10}$$

for all iterations $i$ of the $k$-OSWL. On the $k$-OSAN side we define

$$\mathbf{h}_{\mathbf{g}}^{(i)} := \mathsf{AGG}\big(\{\!\{\mathbf{h}_{v,\mathbf{g}}^{(i)} \mid v \in V(G)\big)$$

and similarly for $\mathbf{h}_{\mathbf{q}}^{(i)}$. We now show

$$C_i(\mathbf{g}) = C_i(\mathbf{q}) \implies \mathbf{h}_{\mathbf{g}}^{(i)} = \mathbf{h}_{\mathbf{q}}^{(i)}. \tag{11}$$

Indeed, $C_i(\mathbf{g}) = C_i(\mathbf{q})$ and Equation (10) imply that there exists a bijection $\theta : V(G) \to V(H)$ such that $C_i(v, \mathbf{g}) = C_i(\theta(v), \mathbf{q})$. Claim 1 implies that $\mathbf{h}_{v,\mathbf{g}}^{(i)} = \mathbf{h}_{\theta(w),\mathbf{q}}^{(i)}$ and thus $\theta$ can be used to define a bijection between the multisets defining $\mathbf{h}_{\mathbf{g}}^{(i)}$ and $\mathbf{h}_{\mathbf{q}}^{(i)}$. Hence, $\mathbf{h}_{\mathbf{g}}^{(i)} = \mathbf{h}_{\mathbf{q}}^{(i)}$ as desired. $\square$

**Lemma 19.** For all $k \geq 1$, it holds that $\rho(k\text{-OSANs}) \subseteq \rho(k\text{-OSWL})$.

This is shown in precisely the same way as Lemma 9.

## E.2  CFI-Gadgets

The comparison with $k$-WL and separation results are derived from a graph construction, also outlined in Morris et al. [2020b, Appendix C.1.1]. They provide an infinite family of graphs $(G_k, H_k)$, $k \in \mathbb{N}$, such that (a) $(k-1)$-WL does not distinguish $G_k$ and $H_k$ but (b) $k$-WL distinguishes $G_k$ and $H_k$. In the following, we recall some relevant results from their paper.

**Construction of $G_k$ and $H_k$.** Let $K_{k+1}$ denote the complete graph on $k+1$ nodes (there are no self-loops). We index the nodes of $K_{k+1}$ from 0 to $k$. Let $E(v)$ denote the set of edges incident to $v$ in $K_{k+1}$. Clearly, $|E(v)| = k$ for all $v \in V(K_{k+1})$. We define the graph $G_k$ as follows.

1. For the node set $V(G_k)$, we add
   (a) $(v, S)$ for each $v$ in $V(K_{k+1})$ and for each *even* subset $S$ of $E(v)$.
   (b) two nodes $e^1$ and $e^0$ for each edge $e$ in $E(K_{k+1})$.

2. For the edge set $E(G_k)$, we add
   (a) an edge $(e^0, e^1)$ for each $e$ in $E(K_{k+1})$,
   (b) an edge between $(v, S)$ and $e^1$ if $v$ in $e$ and $e$ in $S$,
   (c) an edge between $(v, S)$ and $e^0$ if $v$ in $e$ and $e$ not in $S$.

For $v \in V(K_{k+1})$, the set of vertices of the form $(v, S)$ defined in Item 1 are assigned a common color $C_v$. They form what we call a *vertex-cloud* corresponding to the vertex $v$. Similarly, for $e \in E(K_{k+1})$, the two vertices $e^0$ defined in Item 1 are assigned a common color $C_e$. They form what we call an *edge-cloud* corresponding to the edge $e$. A *vertex-cloud vertex* is a vertex of the form $(v, S)$ as defined above. An *edge-cloud vertex* is a vertex of the form $e^0$ or $e^1$ as defined above.

We define the graph $H_k$, in a similar manner to $G_k$, with the following exception. In step 1(a), for the node 0 in $V(K_{k+1})$, we choose all *odd* subsets of $E(0)$. Clearly, both graphs have $(k) \cdot 2^k + \binom{k+2}{2} \cdot 2$ nodes. The above construction of graphs $(G_k, H_k)$ is essentially the application of the classic Cai-Fürer-Immerman construction to a $(k+1)$-clique: we refer to these graphs as *CFI-gadgets* henceforth.

**Distance-two cliques.** We say that a set $S$ of nodes form a *distance-two-clique* if the distance between any two nodes in $S$ is exactly two. A distance-two-clique $S$ is *colorful* if (a) every vertex of $S$ is of vertex-cloud kind, and (b) no two vertices in $S$ belong to the same vertex cloud. Clearly, each vertex in a colorful distance-two-clique has a unique initial color. The following lemma is a mild strengthening of a lemma from Morris et al. [2020b]: the proof is a straightforward derivation from the proof of their lemma.

**Lemma 20** ([Morris et al., 2020b])**.** The following holds for the graphs $G_k$ and $H_k$ defined above.

- There exists a set of $k+1$ vertex-cloud vertices in $G_k$ such that they form a colorful distance-two-clique of size $(k+1)$.

- There does not exist a set of $k+1$ vertex-cloud vertices in $H_k$ such that they form a colorful distance-two-clique of size $(k+1)$.

Hence, $G_k$ and $H_k$ are non-isomorphic.

Further, they showed the following results regarding the power and limitations of Weisfeiler-Leman vis-a-vis such graphs.

**Lemma 21** ([Morris et al., 2020b])**.** The $(k-1)$-WL does not distinguish $G_k$ and $H_k$.

**Lemma 22** ([Morris et al., 2020b])**.** The $k$-WL does distinguish $G_k$ and $H_k$.

### E.3 Separation results: Comparison of vertex-subgraph $k$-OSWL and $k$-WL.

We now compare the relative expressive power of the $k$-ordered subgraph Weisfeiler-Leman and the standard Weisfeiler-Leman. We remark that, by definition, 0-OSWL = 1-WL, so in the remainder of this section we consider $k$-OSWL for $k > 0$.

We show that $k$-OSWL is bounded in distinguishing power by $k+1$-WL (Lemma 24), yet there are are graphs that can be distinguished by $k+1$-WL but not by $k$-OSWL (Proposition 26). Moreover, $k$-OSWL can distinguish graphs which cannot be distinguished by $k$-WL (Lemma 27). As a consequence, As a consequence, the algorithms $k$-OSWL, $k \in \mathbb{N}$, form a strict hierarchy of vertex-refinement algorithms.

**Lemma 23.** Let $k \in \mathbb{N}$. Then $k$-OSWL is strictly less expressive than $(k+1)$-OSWL.

*Proof.* We have shown that (a) $k$-OSWL is strictly less expressive than $(k+1)$-WL, (b) there exist $(k+1)$-WL-indistinguishable graphs which are distinguished by $(k+1)$-OSWL. Hence, we obtain the desired claim. $\square$

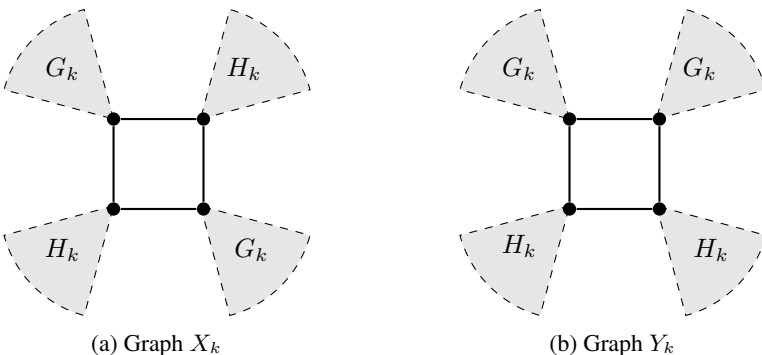

(a) Graph $X_k$                    (b) Graph $Y_k$

Figure 1: Pair of graphs which are $(k+1)$-WL distinguishable but $k$-OSWL indistinguishable. The graphs $G_k$ and $H_k$ are CFI gadgets. Shaded sector represents uniform adjacency to the backbone vertex.

**Lemma 24.** Let $k \in \mathbb{N}$. Then $k$-OSWL is strictly less expressive than $(k+1)$-WL.

We start by describing a construction of a new family of graphs $(X_k, Y_k)$, $k \in \mathbb{N}$, based on the CFI gadgets $G_k$ and $H_k$.

**Construction of the graphs $X_k$ and $Y_k$**    The graph $X_k$ is defined as follows. Let $C$ be a *backbone* cycle of length four with vertices $u_1, u_2, u_3, u_4$, each colored "red". We attach the CFI gadgets $G_k, H_k$ to each of these four vertices as follows. By "attaching a gadget $F$ to a vertex $u$", we mean that every vertex of the gadget $F$ is made adjacent to the backbone vertex $u$. Conversely, a backbone vertex $u$ *points* to a gadget $F$ if $F$ is attached to $u$. Going back to our construction of $X_k$, we attach a copy of $G_k$ each to $u_1$ and $u_3$. We also attach a copy of $H_k$ each to $u_2$ and $u_4$. All the gadget vertices retain their original colors.

The graph $Y_k$ is defined similarly to $X_k$ with the following exception. We attach a copy of $G_k$ each to consecutive vertices $u_1$ and $u_2$, while we attach a copy of $H_k$ each to consecutive vertices $u_3$ and $u_4$. Hence, $X_k$ and $Y_k$ only differ in the cyclic ordering of the attached gadgets.

Observe that the backbone vertices in $X_k$ and $Y_k$ are colored 'red' initially. The gadget vertices inherit their colors from the construction of graphs $G_k$ and $H_k$. These colors are either vertex cloud colors, say $\{C_i : i \in [k+1]\}$, or the edge clouds colors $\{C_{ij} : \{i, j\} \in \binom{k+1}{2}\}$. We call these two kinds of colors along with the red color as the *basic* colors. Let the basic color of a vertex $u$ be denoted by $\beta(u)$.

**Proposition 25.** For $k \in \mathbb{N}$, $(k+1)$-WL distinguishes the graphs $X_k$ and $Y_k$.

*Proof.* It suffices to define a $(k+2)$-variable sentence $\varphi$ in first-order logic with counting quantifiers (FOC) such that $\varphi(X_k) \neq \varphi(Y_k)$ (Indeed, Cai et al. [1992] establishes a precice correspondence between $k$-WL and FOC sentences using $(k+1)$ variables). Intuitively, the sentence $\varphi$ expresses that there is a backbone vertex which has two different backbone vertices, each of which pointing to a CFI-gadget containing a colorful distance-two clique of size $(k+1)$.

We first note that a distance-two-clique of size $(k+1)$ over vertex-cloud vertices is definable as a FOC-formula on $(k+2)$ variables. Indeed, let $C_i$, $i \in [k+1]$ be unary color predicates for vertex clouds, and $C_{ij}$, $\{i, j\} \in \binom{k+1}{2}$ unary predicates for edge clouds. Then,

$$\mathsf{DTC}(x_1, \ldots, x_{k+1}) := \bigwedge_{\substack{i,j \in [k+1] \\ i \neq j}} \mathsf{C}_i(x_i) \wedge \mathsf{C}_j(x_j) \wedge \exists x_{k+2} \left( \mathsf{C}_{ij}(x_{k+2}) \wedge (E(x_{k+2}, x_i) \wedge E(x_{k+2}, x_j)) \right)$$

is a formula that is satisfied by a $(k+1)$-tuple of vertices in a graph when they form a colorful distance-two clique with colors based on vertex and edge clouds.

We proceed to the description of $\varphi$. Let $\exists X$ denote the chain of $k+1$ quantifiers $\exists x_1 \cdots \exists x_{k+1}$. Since the backbone vertices in $X_k$ and $Y_k$ receive a distinct color (red), let $\mathsf{BB}(x)$ denote the unary predicate encoding this condition. The $(k+2)$-variable formula $\mathsf{POINT}_{G_k}(x)$ encodes whether a backbone vertex $x$ points to a $G_k$-gadget, by requiring the existence of a distance-two-clique of the kind stated in Lemma 20.

$$\mathsf{POINT}_{G_k}(x_{k+2}) := \mathsf{BB}(x_{k+2}) \wedge \exists X \left( \mathsf{DTC}(x_1, \ldots, x_{k+1}) \wedge \bigwedge_{i \in [k+1]} E(x_{k+2}, x_i) \right)$$

Then the desired sentence

$$\varphi := \exists x_{k+1} \big( \mathsf{BB}(x_{k+1}) \wedge \exists^{=2} x_{k+2} \big( \mathsf{BB}(x_{k+2}) \wedge E(x_{k+2}, x_{k+1}) \wedge \mathsf{POINT}_{G_k}(x_{k+2}) \big) \big)$$

It is know clear that $X_k$ satisfies $\varphi$: it has a backbone vertex $u_3$ which has exactly two backbone neighbours $u_2$ and $u_4$, each of which point to a $G_k$-gadget. Since $G_k$ contains a distance-two-clique of size $(k+1)$ while $H_k$ does not contain a distance-two-clique of size $(k+1)$, $X_k$ satisfies $\varphi$. On the other hand, $Y_k$ does not satisfy $\varphi$ because it does not have any backbone vertex with two such backbone neighbours. $\quad\square$

**Cyclic Types** Given a vertex tuple $\boldsymbol{z} = (z_1, \ldots, z_k) \in V(X_k)^k$, define its *cyclic type* as follows. For $i \in [4]$, let $S_i$ denote the set of all $j \in [k]$ such that the vertex $z_j$ is either equal to or is attached to the backbone vertex $u_i$. Call $|S_i|$ to be the *weight* of the backbone vertex $u_i$. This associates a cyclic sequence $S_{\boldsymbol{z}} = (S_1, S_2, S_3, S_4)$ with $\boldsymbol{z}$: by cyclic sequence, we mean that only the cyclic ordering of the sets matters, e.g., the cyclic sequence $(S_2, S_3, S_4, S_1)$ is equal to the cyclic sequence $(S_1, S_2, S_3, S_4)$. The cyclic type of $\boldsymbol{z} = (z_1, \ldots, z_k)$ is then defined by the tuple $(\beta(z_1), \ldots, \beta(z_k))$ of basic colors and the cyclic sequence $S_{\boldsymbol{z}}$. The same procedure can be used to define the cyclic type of a vertex tuple $\boldsymbol{z} \in V(Y_k)^k$. Further, a cyclic type is said to be *simple* if the weight of every backbone vertex is at most $k-2$. If the cyclic type is not simple, there is a unique backbone vertex of weight at least $k-1$. We call such a vertex as a *leader*. If the weight of the leader is exactly $k-1$, there exists a *follower* vertex of weight one. A backbone vertex of weight zero is called *weightless*.

Instead of usual color refinement (CR), we define a *skewed* color refinement (SCR) on the graphs $(X_k, \boldsymbol{u})$ and $(Y_k, \boldsymbol{v})$, as follows: in the first stage, we exhaustively and exclusively refine the class of backbone vertices. This refinement uses color information from both backbone and non-backbone vertices. In the second stage, we do the usual Color Refinement on the resulting graph. Using standard arguments, it is easy to show that both CR and SCR both converge to the same stable coloring.

**Proposition 26.** *For $k \in \mathbb{N}$, $k$-OSWL cannot distinguish the graphs $X_k$ and $Y_k$.*

*Proof.* It suffices to show a partition $\mathcal{P}_X$ of $V(X_k)^k$ into $m$ classes $P_1, \ldots, P_m$ and a partition $\mathcal{P}_Y$ of $V(Y_k)^k$ into $m$ classes $Q_1, \ldots, Q_m$ such that for each $i \in [m]$ it holds that (a) $|P_i| = |Q_i|$, and (b) graphs $(X_k, \boldsymbol{u})$ and $(Y_k, \boldsymbol{v})$ are indistinguishable under color refinement, for all $\boldsymbol{u} \in P_i$ and for all $\boldsymbol{v} \in Q_i$. Here $(X_k, \boldsymbol{u})$ stands for a copy of the graph $X_k$ in which the vertex $\boldsymbol{u}_i$ receives a distinct initial color $i$, for each $i \in [k]$. Similarly, $(Y_k, \boldsymbol{v})$ stands for a copy of the graph $Y_k$ in which the vertex $\boldsymbol{u}_i$ receives a distinct initial color $i$, for each $i \in [k]$.

For the partition $\mathcal{P}_X$, we first partition the tuples in $V(X_k)^k$ into sets $P_\tau$ according to their cyclic type $\tau$. Next, if a cyclic type $\tau$ is not simple, we further partition the set $P_\tau$ depending on whether the leader vertex points to a $G_k$-gadget or a $H_k$-gadget. We obtain a corresponding partition $\mathcal{P}_Y$ following the same process for $Y_k$.

Instead of usual color refinement (CR), we do a *skewed* color refinement (SCR) on the graphs $(X_k, \boldsymbol{u})$ and $(Y_k, \boldsymbol{v})$: in first stage, we exhaustively and exclusively refine the class of backbone vertices, and in second stage we do the usual Color Refinement on the resulting graph. Using standard arguments, it is easy to show that both CR and SCR both converge to the same stable coloring.

Let $\tau$ be a simple cyclic type. It is easy to verify that the number of tuples of type $\tau$ in $V(X_k)^k$ and $V(Y_k)^k$ are equal. Suppose that $\boldsymbol{u} \in V(X_k)^k$ and $\boldsymbol{v} \in V(Y_k)^k$ have the cyclic type $\tau$. After the first stage of SCR, the two graphs are indistinguishable because of the cyclic types being equal. After the second stage of SCR, the backbone vertices do not get refined any further: since each gadget has at most $k-2$ individualized vertices, it is not possible to identify whether it is a $G_k$ gadget or a $H_k$ gadget with color refinement (otherwise $k$-WL would also distinguish $G_k$ and $H_k$). Hence, Color Refinement does not distinguish $(X_k, \boldsymbol{u})$ and $(Y_k, \boldsymbol{v})$.

Otherwise, let $\tau$ be a non-simple type. Let $\boldsymbol{u} \in V(X_k)^k$ and $\boldsymbol{v} \in V(Y_k)^k$ of type $\tau$ such that their leader vertices point to a $G_k$-gadget. Again, it is easy to verify that the number of such tuples in $V(X_k)^k$ and $V(Y_k)^k$ are equal. Again, we do a skewed color refinement on the graphs $(X_k, \boldsymbol{u})$ and $(Y_k, \boldsymbol{v})$. After the first stage of SCR, the two graphs are again indistinguishable because of the cyclic types being equal. After the second stage of SCR, the backbone vertices again do not get refined any further for the following reason. The leader and the follower vertices are already in singleton color classes. If the weightless vertices are not already distinguished after stage one, they will not get distinguished any further because the gadgets attached to them do not have any individualized vertices and hence they cannot be distinguished by Color

Refinement (i.e. 1-WL). Hence, CR does not distinguish $(X_k, \boldsymbol{u})$ and $(Y_k, \boldsymbol{v})$. A similar argument works when both leader vertices point to a $H_k$-gadget. This finishes our case analysis. $\qquad\square$

We conclude by comparing $k$-WL and $k$-OSWL.

**Lemma 27.** For each $k \in \mathbb{N}$ there exist $k$-WL-indistinguishable graphs which are distinguished by $k$-OSWL.

*Proof.* We show that the CFI-gadget graphs $G_{k+1}$ and $H_{k+1}$ can be distinguished by $k$-OSWL. Since Lemma 21 implies that these graphs cannot be distinguished by $k$-WL this suffices.

More specifically, we will identify a $k$-ordered subgraph $\mathbf{g}$ in $V(G_{k+1})^k$ for which $C(\mathbf{g})$ is different from any $C(\mathbf{q})$ with $\mathbf{q}$ in $V(H_{k+1})^k$.

Let $\{v_1, \ldots, v_k, v_{k+1}, v_{k+2}\}$ be a colorful distance-two-clique in $G_{k+1}$. Recall that each $v_i$ lies in a distinct vertex cloud. We set $\mathbf{g} = (v_1, \ldots, v_k)$. For each pair $v_i, v_j$ with $i, j \in [k]$, there exist a vertex $v_{ij}$ in an edge cloud, such that $(v_i, v_{ij})$ and $(v_{ij}, v_j)$ are edges. This information is captured by $C(v_{ij}, \mathbf{g})$ and hence also by $C(\mathbf{g})$. In other words, $C(\mathbf{g})$ will reflect that the vertices in $\mathbf{g}$ form a colorful distance-two clique of size $k$. We now argue that $C(\mathbf{g})$ also reflects that there is a distance-two clique of size $k + 2$ in $G_{k+1}$.

Indeed, observe that $C(v_{k+1}, \mathbf{g})$ contains information that $v_{k+1}$ is connected to all vertices in $\mathbf{g}$ at distance two, and similarly for $C(v_{k+2}, \mathbf{g})$. Moreover, since $C(v_{k+1}, \mathbf{g})$ reflects that $v_{k+2}$ is at distance two from $v_{k+1}$. In other words, $C(\mathbf{g})$ indeed reflects that there is a colorful distance-two cliques of size $k + 2$ in $G_{k+1}$. By Lemma 20, $C(\mathbf{q})$ cannot reflect this since $H_{k+1}$ does not contain a colorful distance-two cliques of size $k + 2$. $\qquad\square$

# F  Subgraph-enhanced GNNs as $k$-OSANs, Proofs of Propositions 4 to 6.

## F.1  Unordered vs. ordered subgraphs

We specified $k$-OSANs using ordered $k$-vertex subgraphs $G[\mathbf{v}]$ with $\mathbf{v} \in V(G)^k$. The order information is encoded in $G[\mathbf{v}]$ by means of the vertex labels in $[k]$ of the vertices in $\mathbf{v}$. In the unordered case, we would simply consider $G[\mathbf{v}]$ without any labels. That is, $k$-OSANs using ordered $k$-vertex subgraphs can simulate any $k$-OSAN using unordered $k$-vertex subgraphs.

As an example of how ordered $k$-vertex subgraphs can be used, consider $\mathbf{v} = (v_1, v_2, v_2, v_3) \in V(G)^4$ and assume that the vertices $v_1, v_2$ and $v_3$ form a 3-clique in $G$. In the ordered case $G[\mathbf{v}]$ is simply the 3-clique, in the ordered case, $G[\mathbf{v}]$ is the 3-clique in which $v_1$ is labeled with 1, $v_2$ with 2 and 3, and $v_3$ with 3. Suppose we want to use the selected subgraph to simulate edge deletions. Then, in the unordered case one cannot distinguish between the different edges in the 3-clique, and hence they will be all treated as deleted. In contrast, in the ordered case we can, e.g., only delete edges with end points labeled 1 and 2, and 2 and 3, leaving the edge labeled 1 and 3 intact.

## F.2  Proofs

In the following, we show that $k$-OSANs capture most recently proposed subgraph-enhanced GNNs, implying Propositions 4 to 6.

**Marked GNNs, dropout GNNs and reconstruction GNNs**  Dropout GNNs [Papp et al., 2021] generate vertex embeddings by running classical MPNNs on $k$-vertex deleted subgraphs and then aggregating the obtained embeddings. Dropout GNNs were generalized to *Marked GNNs* ($k$-mGNNs) [Papp and Wattenhofer, 2022] in which the $k$ vertices are just marked, in contrast to always being deleted. For efficiency reasons, a random strategy is used to select the $k$ vertices to be marked or deleted [Papp et al., 2021, Papp and Wattenhofer, 2022].

Here, we consider the deterministic variant of $k$-mGNNs in which all possible sets of $k$ vertices are considered to be marked (as this provides the maximum distinguishing power) as is also used in Cotta et al. [2021] in the context of reconstruction GNNs. The marking process in $k$-mGNNs naturally relates to the selection of unordered $k$-vertex subgraphs, as we will illustrate.

Let $G$ be a graph and let $M \subseteq V(G)$, with $|M| = k$, be a set of $k$ marked vertices. Let $N_G^M(v) := N_G(v) \cap M$ be the set of marked neighbors of $v$. As described in Papp and Wattenhofer [2022], when running an MPNN on a graph with $k$ marked vertices $M$, features are computed in layer $i \geq 0$ as

$$\mathbf{h}_v^{(i+1)} := \mathsf{AGG}_{\mathsf{marked}}^{(i+1)}\big(\{\!\!\{\mathbf{h}_u^{(i)} \mid u \in N_G^M(v)\}\!\!\}\big) + \mathsf{AGG}_{\mathsf{unmarked}}^{(i+1)}\big(\{\!\!\{\mathbf{h}_u^{(i)} \mid u \in N_G(v) \backslash N_G^M(v)\}\!\!\}\big). \quad (12)$$

In other words, during neighbor aggregation, MPNNs can distinguish between marked and unmarked neighbors. Furthermore, for $k$-mGNNs one first obtains vertex features for all markings $M$, which are subsequently aggregated into a single vertex feature. Finally, these vertex features are aggregated to obtain a single graph feature.

Inspecting Equation (12), we see that we can replace the two aggregation functions by one aggregation function provided that $\mathbf{h}_u^{(i)}$ contains information indicating whether or not $u$ is marked. In other words, we can replace Equation (12) by

$$\mathbf{h}_v^{(i+1)} := \mathsf{AGG}^{(i+1)}\big(\{\!\!\{(\mathbf{h}_u^{(i)}, \mathbb{1}_{u \in M}) \mid u \in N_G(v)\}\!\!\}\big),$$

for a given set $M$ of markings. We use this observation for casting $k$-mGNNs as $k$-OSANs. Indeed, each marking $M$ corresponds to an ordered $k$-vertex subgraph $\mathbf{g} \in G_k$. Furthermore, we set the update function in $\mathbf{h}_{v,\mathbf{g}}^{(0)} := \mathsf{UPD}(\mathsf{atp}(v, \mathsf{t}(\mathbf{g})))$ such that it returns the label $l(v)$ of $v$ and the indicator function $\mathbb{1}_{v \in \mathbf{g}}$. We ensure that all update functions propagate the indicator function to the next layers such that aggregation functions have this information at their disposal in every layer. As mentioned, this suffices to perform the aggregation carried out by $k$-mGNNs. Moreover, all possible markings are considered in $k$-mGNNs. Hence, the $k$-OSAN will select all possible $k$-vertex graphs as well. We will capture this by setting $\boldsymbol{\pi}_{v,\mathbf{g}} = 1$ below.

More precisely, the following $k$-OSAN correspond to $k$-mGNNs:

$$\begin{aligned}
\mathbf{h}_{v,\mathbf{g}}^{(0)} &= (l(v), \mathbb{1}_{v \in \mathbf{g}}) \\
\boldsymbol{\pi}_{v,\mathbf{g}} &= 1 \\
\mathbf{h}_{v,\mathbf{g}}^{(i+1)} &= \mathsf{UPD}^{(i+1)}\Big(\mathbf{h}_{v,\mathbf{g}}^{(i)}, \mathsf{AGG}^{(i+1)}\big(\{\!\!\{\mathbf{h}_{u,\mathbf{g}}^{(i)} \mid u \in N_G(v)\}\!\!\}\big)\Big) \\
\mathbf{h}_v^{(T)} &= \mathsf{SAGG}\big(\{\!\!\{\mathbf{h}_{v,\mathbf{g}}^{(T)} \mid \mathbf{g} \in G_k \text{ s.t. } \boldsymbol{\pi}_{v,\mathbf{g}} \neq 0\}\!\!\}\big) = \mathsf{SAGG}\big(\{\!\!\{\mathbf{h}_{v,\mathbf{g}}^{(T)} \mid \mathbf{g} \in G_k\}\!\!\}\big) \\
\mathbf{h}_G &= \mathsf{AGG}\big(\{\!\!\{\mathbf{h}_v^{(T)} \mid v \in V(G)\}\!\!\}\big),
\end{aligned}$$

where the aggregation functions $\mathsf{AGG}^{(i+1)}$ are those from the marked GNN under consideration, and the update functions are such that they propagate the indicator function $\mathbb{1}_{v \in \mathbf{g}}$ to the next iteration, as explained before. Finally, $\mathbf{h}_G$ is obtained by aggregating over vertex embeddings, which in turn are defined in terms of aggregating over vertex embedding $\mathbf{h}_{v,\mathbf{g}}^{(T)}$ for $\mathbf{g} \in G_k$. This is in accordance with how marked GNNs operate.

We also note that $k$-mGNNs in Papp and Wattenhofer [2022] are guaranteed to be stronger than MPNNs because they run a classical MPNN alongside. This is not shown in the $k$-OSAN description given above as any architecture can be made at least as strong as MPNNs in this way.

We next consider $k$-*reconstruction GNNs* ($k$-recGNNs) [Cotta et al., 2021] which for each set $S$ of $k$ vertices, compute a vectorial representation (using an MPNN) of $G[S]$, then concatenate all the obtained representations (for all $S$), followed by the application of a permutation invariant update function to obtain a graph representation. The difference with marked GNN is thus that the order of aggregation is different. And indeed, $k$-recGNNs are captured by vertex-subgraph $k$-OSANs, as we will see shortly.

Clearly, the $S$ of $k$ vertices and, more specifically, $G[S]$ corresponds to a vertex-ordered subgraph $\mathbf{g} \in G_k$. Then, in order to compute a representation of $G[S]$ using $\mathbf{g}$, we proceed as follows: We run an MPNN on $G[S]$ by ensuring that the update and aggregation functions in the vertex-subgraph $k$-OSANs know which vertices belong to $\mathbf{g}$ (i.e., $G[S]$). This is done in the same way as for $k$-mGNNs by including this information in the initial features. In contrast to $k$-mGNNs, we perform vertex aggregation for each $\mathbf{g}$ to obtain a representation of $\mathbf{g}$ ($G[S]$). Then, we aggregate over all $\mathbf{g}$ (i.e., all $S$ and thus $G[S]$) using concatenation as an aggregation function, and finally apply an update function, following how $k$-recGNNs operate. We have thus shown the following.

**Proposition 28** (Proposition 4 in the main text)**.** For $k \geq 1$, $k$-OSANs capture $k$-mGNNs and vertex-subgraph $k$-OSANs capture $k$-recGNNs.

Our results on expressive power of $k$-OSANs now imply that these architectures are bounded by (and are strictly weaker than) $(k+1)$-WL.

**Identity-aware GNNs**   We next consider identity-aware GNNs (idGNNs) [You et al., 2021], an extension of MPNNs in which message functions can differentiate whether the vertices visited are equal or different from a given center vertex and vertex exploration only happens inside the $h$-hop egonet of the center vertices.

More specifically, let us denote by $N_G^h(v)$ the set of $h$-hop neighbors of the "center" vertex $v$. Then, for each $v \in V(G)$, idGNNs compute vertex features of $u \in N_G^h(v)$ in layer $i > 0$ as

$$\mathbf{h}_{u,v}^{(i+1)} := \mathsf{UPD}^{(i+1)}\big(\mathbf{h}_{u,v}^{(i)}, \mathsf{AGG}^{(i+1)}(\{\!\!\{(\mathbf{h}_{w,v}, \mathbb{1}_{w=v}) \mid w \in N_G(u) \cap N_G^h(v)\}\!\!\})\big)$$

and then after layer $h$, one lets $\mathbf{h}_v := \mathbf{h}_{v,v}^{(h)}$ and $\mathbf{h}_G := \mathsf{AGG}(\{\!\!\{\mathbf{h}_v \mid v \in V(G)\}\!\!\})$. In other words, vertex features are computed by means of a local message passing neural network, centered around each vertex, in which the aggregation functions can distinguish between the center and other vertices.

We next show how to model idGNNs as 1-OSANs. We first observe that 1-OSANs can extract information related to $h$-hop distance neighbors. More precisely, let $g \in G_1 = V(G)$ be a single-vertex subgraph. We first compute the function $\mu_{u,g}^{(i)}$ for $0 \le i \le h$, indicating if the vertex $u \in N_G^i(g)$. We can compute $\mu_{u,g}^{(i)}$ using $i$ 1-OSANs layers as follows:

$$\mu_{u,g}^{(0)} = \mathsf{UPD}^{(0)}\Big(\mathsf{atp}(u,g)\Big) = \mathbb{1}_{u=g}$$

$$\mu_{u,g}^{(i+1)} = \mathsf{UPD}^{(i+1)}\Big(\mu_{u,g}^{(i)}, \mathsf{AGG}^{(i+1)}\big(\{\!\!\{\mu_{w,g}^{(i)} \mid w \in N_G(u)\}\!\!\}\big)\Big),$$

where the update and aggregation functions are such that $\mu_{u,g}^{(i+1)} = 1$ if and only if $\mu_{u,g}^{(i)} = 1$ or there exists a $w \in N_G(u)$ with $\mu_{w,g}^{(i)} = 1$. We will use these layers for computing the indicator function $\mathbb{1}_{u \in N_G^h(g)}$ in other architectures below.

We can now model idGNNs as 1-OSANs, as follows. We let the center vertices correspond to 1-vertex subgraphs $g \in V(G)$, and ensure that the initial features $\mathbf{h}_{v,g}^{(0)}$ carry around $\mathbb{1}_{v=g}$ and $\mu_{v,g}^{(h)}$ (i.e., $\mathbb{1}_{v \in N_G^h(g)}$). As before, we assume that all update functions propagate this information to the next layer such that aggregation functions can take into account whether or not a vertex is equal $g$ or lies in $N_G^h(g)$.

In contrast to 1-mGNNs and 1-recGNNs, where $\boldsymbol{\pi}_{v,g}$ did not restrict the subgraphs, idGNNs obtain vertex features for $v$ only using the subgraph $g = v$ (recall $\mathbf{h}_v := \mathbf{h}_{v,v}^{(h)}$). Hence, we will impose that $\boldsymbol{\pi}_{v,g} = 1$ iff $v = g$. More specifically, idGNNs correspond to 1-OSANs of the form:

$$\mathbf{h}_{v,g}^{(0)} = (l(v), \mathbb{1}_{v=g}, \mu_{v,g}^{(h)})$$

$$\boldsymbol{\pi}_{v,g} = \mathsf{UPD}(\mathsf{atp}(v,g)) = \mathbb{1}_{v=g}$$

$$\mathbf{h}_{v,g}^{(i+1)} = \mathsf{UPD}^{(i+1)}\Big(\mathbf{h}_{v,g}^{(i)}, \mathsf{AGG}^{(i+1)}\big(\{\!\!\{\mathbf{h}_{u,g}^{(i)} \mid u \in N_G(v)\}\!\!\}\big)\Big)$$

$$\mathbf{h}_v = \mathsf{SAGG}\big(\{\!\!\{\mathbf{h}_{v,g}^{(h)} \mid g \in G_1 \text{ s.t. } \boldsymbol{\pi}_{v,g} \ne 0\}\!\!\}\big) = \mathsf{SAGG}\big(\{\!\!\{\mathbf{h}_{v,v}^{(h)}\}\!\!\}\big)$$

$$\mathbf{h}_G = \mathsf{AGG}\big(\mathbf{h}_v \mid v \in V(G)\big),$$

where $\mathsf{AGG}^{(i+1)}$ only takes into account those $u \in N_G(v) \cap N_G^h(g)$ (using $\mu_{u,g}^{(h)} = 1$) and also uses the availability of $\mathbb{1}_{u=g}$ to simulate the aggregation function used in idGNNs. The definitions of $\mathbf{h}_v$ and $\mathbf{h}_G$ are as in the description of idGNNs given earlier.

You et al. [2021] showed that idGNNs can distinguish more graphs than 1-WL based on the counting of cycles. By viewing idGNNs as 1-OSANs our results provide an upper bound by 2-WL for idGNNs. This is consistent with their ability to count cycles, as this can be done in 2-WL.

Although not considered in You et al. [2021], one could extend idGNNs to $k$-idGNNs by allowing checking for identity with vertices on a previously explored path of length $k-1$, as follows:

$$\mathbf{h}_{v,\mathbf{g}}^{(0)} = (l(v), \mathbb{1}_{v \in \mathbf{g}}, \mu_{v,g_1}^{(h)})$$

$$\boldsymbol{\pi}_{v,\mathbf{g}} = \mathsf{UPD}(\mathsf{atp}(v,\mathbf{g})) = \begin{cases} 1 & \text{if } v = g_1, g_2, \dots, g_k \text{ from a path in } G \\ 0 & \text{otherwise} \end{cases}$$

$$\mathbf{h}_{v,\mathbf{g}}^{(i+1)} = \mathsf{UPD}^{(i+1)}\Big(\mathbf{h}_{v,\mathbf{g}}^{(i)}, \mathsf{AGG}^{(i+1)}\big(\{\!\!\{\mathbf{h}_{u,\mathbf{g}}^{(i)} \mid u \in N_G(v)\}\!\!\}\big)\Big)$$

$$\mathbf{h}_v = \mathsf{SAGG}\big(\{\!\!\{\mathbf{h}_{v,\mathbf{g}}^{(h)} \mid \mathbf{g} \in G_k \text{ s.t. } \boldsymbol{\pi}_{v,\mathbf{g}} \neq 0\}\!\!\}\big)$$

$$= \mathsf{SAGG}\big(\{\!\!\{\mathbf{h}_{v,\mathbf{g}} \mid \mathbf{g} \text{ is a walk in } G \text{ of length } k \text{ starting from } v\}\!\!\}\big)$$

$$\mathbf{h}_G = \mathsf{AGG}\big(\{\!\!\{\mathbf{h}_v \mid v \in V(G)\}\!\!\}\big)$$

Our results show that such $k$-idGNNs are bounded by $(k+1)$-WL in expressive power. It is also readily verified that $k$-idGNNs for $k > 1$ can detect more complex substructures than cycles.

**Nested GNNs**  We next consider Nested GNNs (nestedGNNs) [Zhang and Li, 2021] that obtain vertex embeddings based on the aggregation over vertex embeddings in the $h$-hop egonets. In their notation, $G_w^h$ denotes the subgraph in $G$, rooted at $w$ of "height" $h$. Then, for any vertex $v \in V(G_w^h)$ they compute:

$$\mathbf{h}_{v,G_w^h}^{(i+1)} = \mathsf{UPD}_1^{(i+1)}\Big(\mathbf{h}_{v,G_w^h}^{(i)}, \sum_{u \in N(v|G_w^h)} \mathsf{UPD}_2^{(i+1)}\big(\mathbf{h}_{u,G_w^h}^{(i)}\big)\Big)$$

where $N(v \mid G_w^h)$ is the set of neighbors of $v$ within $G_w^h$. Pooling happens after layer $T$:

$$\mathbf{h}_w = \mathsf{AGG}(\{\!\!\{\mathbf{h}_{v,G_w^h}^{(T)} \mid v \in V(G_w^h)\}\!\!\})$$

and then $\mathbf{h}_G = \mathsf{AGG}(\{\!\!\{\mathbf{h}_v \mid v \in V(G)\}\!\!\})$.

We can formulate nestedGNNs as 1-OSANs as follows. Similarly as for idGNNs, the center vertices correspond to 1-vertex subgraphs $g$ and we again include the information $\mu_{v,g}^{(h)}$ as initial feature in order to aggregate over vertices in $N_G^h(g)$, i.e., those in $G_g^h$. We assume that the update function propagate this information to higher layers, as before. As aggregation functions $\mathsf{AGG}^{(i+1)}$, we use summation but only over those features for which the $\mu_{u,g}^{(h)}$ component is 1 and in this way simulate the aggregation step used in nestedGNNs. More specifically, we have:

$$\mathbf{h}_{v,g}^{(0)} = (l(v), \mu_{v,g}^{(h)}) \tag{13}$$

$$\boldsymbol{\pi}_{v,g} = \mathsf{UPD}(\mathsf{atp}(v,g)) = \mathbb{1}_{v=g} \tag{14}$$

$$\mathbf{h}_{v,g}^{(i+1)} = \mathsf{UPD}^{(i)}\Big(\mathbf{h}_{v,g}^{(i)}, \mathsf{AGG}^{(i+1)}\big(\{\!\!\{\mathbf{h}_{u,g}^{(i)} \mid u \in N_G(v)\}\!\!\}\big)\Big), \tag{15}$$

which similar as to how idGNNs, viewed as 1-OSANs, operate. The main difference with idGNNs is how vertex features are computed. Indeed, nestedGNNs assign to vertex $v$ the representation of $G_v^h$. We use $\mathsf{AGG}^{(T+1)}$ to aggregate over all vertices $u$ in $G_g^h$ by leveraging $\mu_{u,g}^{(h)}$. More precisely, instead of aggregating over neighbors we aggregate over the entire vertex set but ensure that $\mathsf{AGG}^{(T+1)}$ only takes into account those vertices in $N_G^h(g)$ using $\mu_{u,g}^{(h)}$:

$$\mathbf{h}_{v,g} = \mathsf{AGG}^{(T+1)}\big(\{\!\!\{\mathbf{h}_{u,g}^{(T)} \mid w \in V(G)\}\!\!\}\big)$$

$$\mathbf{h}_v = \mathsf{SAGG}\big(\{\!\!\{\mathbf{h}_{v,g} \mid g \in G_1 \text{ s.t. } \boldsymbol{\pi}_{v,g} \neq 0\}\!\!\}\big) = \mathsf{SAGG}\big(\{\!\!\{\mathbf{h}_{v,v}\}\!\!\}\big) \tag{16}$$

$$\mathbf{h}_G = \mathsf{AGG}\big(\{\!\!\{\mathbf{h}_v \mid v \in V(G)\}\!\!\}\big), \tag{17}$$

where the final steps are in place to create a graph representation in accordance with how nestedGNNs operate.

Zhang and Li [2021] observe that nestedGNNs are more powerful than 1-WL and raise the question whether nestedGNNs can be more powerful than 2-WL. By viewing nestedGNNs as 1-OSANs, our general results about expressive power, show that nestedGNNs are bounded by 2-WL in expressive power. Moreover, Zhang and Li [2021] allude to deeper nested GNNs in their paper. It seems natural to conjecture that these can be cast as $k$-OSANs when $k$ levels of nesting are used. We leave the verification of this conjecture for future work.

**GNN As Kernel**  Very related to nestedGNNs are GNN as kernel (kernelGNNs) [Zhao et al., 2021]. Indeed, the only difference is that once the $\mathbf{h}_v$ are defined for $v \in V(G)$ in Equation (16), kernelGNNs restart the message passing over egonets (Equation (13)-Equation (15)), but this time with the initial features $\mathbf{h}_{v,g}^{(0)}$ replaced by $(\mathbf{h}_v, \mu_{v,g}^{(h)})$. This is then repeated a number of times, after which a graph representation is obtained, just as for nestedGNNs (Equation (17)). It is now readily verified that we can express kernelGNNs as 1-OSANs in the same as we showed for nestedGNNs.

**Proposition 29** (Proposition 5 in the main text). The 1-OSANs capture idGNNs, kernelGNNs, and nestedGNNs.

*Proof.* The result follows from the above equations. ☐

We note again that our results show that idGNNs, kernelGNNs and nestedGNNs are all bounded in expressive power by 2-WL, yet are less expressive than 2-WL.

**DS-GNN with the $k$-vertex-deleted policy**   In the following, we define an instance of a vertex-subgraph $k$-OSAN which captures DS-GNNs with the $k$-vertex-deleted policy [Bevilacqua et al., 2021]. In a nutshell, DS-GNNs generate MPNN-based representations of a collection of subgraphs and then aggregate those to obtain a representation of the original graph. In general, a policy is in place in DS-GNNs to select the subgraphs. Here, we consider the $k$-vertex-deleted policy in which all $k$-vertex deleted subgraphs $S$ are considered. The deletion of $k$-vertices used to obtain a subgraph $S$ will be simulated by considering $k$-vertex subgraphs $\mathbf{g}$ and by treating the vertices in $\mathbf{g}$ to be marked. In other words, DS-GNNs act like a $k$-mGNNs except that graph representations are obtained by aggregating subgraph representations. More specifically:

$$\mathbf{h}_{v,\mathbf{g}}^{(0)} = (l(v), \mathbb{1}_{v \in \mathbf{g}})$$
$$\boldsymbol{\pi}_{v,\mathbf{g}} = 1$$
$$\mathbf{h}_{v,\mathbf{g}}^{(i+1)} = \mathsf{UPD}^{(i+1)}\Big(\mathbf{h}_{v,\mathbf{g}}^{(i)}, \mathsf{AGG}^{(i+1)}\big(\{\!\!\{\mathbf{h}_{u,\mathbf{g}}^{(i)} \mid u \in N_G(v)\}\!\!\}\big)\Big)$$
$$\mathbf{h}_{\mathbf{g}} = \mathsf{AGG}\big(\{\!\!\{\mathbf{h}_{v,\mathbf{g}}^{(T)} \mid v \in V(G)\}\!\!\}\big)$$
$$\mathbf{h}_G = \mathsf{SAGG}\big(\{\!\!\{\mathbf{h}_{\mathbf{g}} \mid \mathbf{g} \in G_k, \exists v \in V(G)\, \boldsymbol{\pi}_{v,\mathbf{g}} \neq \mathbf{0}\}\!\!\}\big) = \mathsf{SAGG}\big(\{\!\!\{\mathbf{h}_{\mathbf{g}} \mid \mathbf{g} \in G_k\}\!\!\}\big),$$

where update functions propagate $\mathbb{1}_{v \in \mathbf{g}}$ and aggregation functions treat vertices in $\mathbf{g}$ as marked (or to be deleted).

**Proposition 30** (Proposition 6 in the main text). Vertex-subgraph $k$-OSANs capture DS-GNNs with the $k$-vertex-deleted policy.

*Proof sketch.* We argue that the above $k$-OSAN instance can simulate DS-WL [Bevilacqua et al., 2021] which upper bounds any possible DS-WL in terms of distinguishing non-isomorphic graphs.

As noted by Papp and Wattenhofer [2022], see also above paragraph on $k$-mGNNs, marking vertices is at least as powerful as removing them. The markings enable the aggregation function to distinguish between deleted and non-deleted vertices. By choosing injective instances of UPD and AGG, we can simulate the coloring function $c_{v,\mathbf{g}}^{(i)}$, for $i \geq 0$, of the DS-WL. That is, if $\mathbf{h}_{v,\mathbf{g}}^{(i)} = \mathbf{h}_{w,\mathbf{g}}^{(i)}$ holds, it follows that $c_{v,\mathbf{g}}^{(i)} = c_{w,\mathbf{g}}^{(i)}$ for all vertices $v$ and $w$ of a given graph $G$ and $\mathbf{g} \in G_k$ holds. The existence of such instances follows directly from the proof of Theorem 2 in [Morris et al., 2019]. Similarly, by choosing injective instances of SAGG and AGG for computing the single graph feature, the resulting architecture has at least the same expressive power as the DS-WL in distinguishing non-isomorphic graphs. The reasoning is analogous to the proof Lemma 19. Hence, the resulting architecture has at least the expressive power of DS-WL, implying the result. ☐

Bevilacqua et al. [2021] also consider the 1-edge-deleted policy in which the subgraphs $S$ considered are those obtained by deleting a single edge. The deletion of an edge can be simulated by marking two vertices, which can be simulated using message and update functions having access to 2-vertex subgraphs $\mathbf{g} \in G_2$. Hence, DS-GNNs with the 1-edge-deleted policy can be captured by vertex-subgraph 2-OSANs. As a consequence, such DS-GNNs are bounded in expressive power by 3-WL. Combined with the discussion in Appendix F.1 it should be clear that DS-GNNs with $k$-edge-deleted policy can be captured by vertex-subgraph $2k$-OSANs. As argued in Appendix F.1 the use of ordered graphs is crucial to simulate multiple edge deletions. Finally, Bevilacqua et al. [2021] also consider two variants of $k$-hop ego-net policies. In the first, the subgraphs $S$ consist of all $k$-hop ego-net subgraphs, one for each vertex in the graph. In the second variant, equality with the center vertex in each ego-net can be checked. It should be clear from our treatment of nestedGNNs and kernelGNNs that the ego-net extraction can be simulated in vertex-subgraph 1-OSANs and that the distinction between two variants pours down to include $\mathbb{1}_{v=g}$ as an initial feature (just as for idGNNs). Hence DS-GNNs with ego-net policies are bounded by 2-WL.