# OpenReview forum: "Ordered Subgraph Aggregation Networks"
_NeurIPS.cc/2022/Conference — NeurIPS 2022 Accept_

### Official Review · Reviewer_n3h8 · 2022-07-03

**Rating:** 7
**Confidence:** 3
**Soundness:** 3 good
**Presentation:** 4 excellent
**Contribution:** 4 excellent

**Summary:**

This work develops a general family of subgraph-based GNN architectures that generalizes and unifies many of the recently proposed subgraph-enhanced GNNs. $k$-OSANs are based on message passing on all ordered $k$-vertex subgraphs of an input graph. Expressive power of this architecture is analyzed, showing that $k$-OSANs incomparable to $k$-WL, but always less powerful than $(k+1)$-WL. Further, a data-driven subgraph sampling procedure is developed.


**Questions:**

1. I think it is worth explicitly stating that your theoretical results give upper bounds on the expressive power of many existing methods, say at the very end of Section 3. This is a nice result!
2. At the end of the related work (line 84), it would be better to say that the above works *mostly* "do not study the approaches" expressive power beyond 2-WL. For instance, some works like the reconstruction GNN (Cotta et al, 2021) demonstrate ability to compute functions on graphs like cycles, that 2-WL cannot count (beyond a certain size). Also, the edge deletion ESAN (Bevilacqua et al. 2022) goes past 2-WL on strongly regular graphs.
3. Could you clarify, what is the O(n+ klog k ) algorithm for computing the MAP of the subgraph sampling distribution for ordered subgraphs?
4. It's worth mentioning the diversity loss (Appendix C) used in I-MLE sampling in the main paper.
5. The MAE numbers reported here for Alchemy and QM9 appear to be in different units than that of other work. In particular, these are much larger. Can you clarify how to compare these to that of other works?



**Limitations:**

Unless I missed some things, this work does not discuss limitations much at all in the main paper. I also do not see this in the Appendix, though the Appendix is long, so difficult to search.


**Strengths And Weaknesses:**

Strengths:
1. The theoretical results are interesting. In particular, the upper bound of $k$-OSANs (and hence many other methods) by $(k+1)$-WL is a nice result.
2. That the architecture uses message passing, and can independently process subgraphs is great for efficiency; future works can probably take advantage of this even further.
3. The use of ordered subgraphs in an overall permutation invariant architecture is pretty novel and smart.
4. The data-driven subgraph sampling seems mostly reasonable, and it tackles the important problem of how to get good performance without using all subgraphs / a uniformly sampled subset of the exponentially many subgraphs.

Weaknesses:
1. Experimental results are not that strong. Most comparisons are between random sampling and data-driven sampling. Assuming you use edge features on ZINC (is this true?), the results are not great, and the methods get outperformed by some message passing GNNs.

**Note:** I have not checked the proofs of expressive power, and thus I have not read much of the Appendix.

---

> ### Author Response · Authors · 2022-08-02
> **Replies to Reviewer n3h8**
>
> We thank the reviewer for the careful and constructive review.
>
> **Q1:** "*Experimental results are not that strong. Most comparisons are between random sampling and data-driven sampling. Assuming you use edge features on ZINC (is this true?), the results are not great, and the methods get outperformed by some message passing GNNs.*"
>
> **A1:** See Table 1 in our comment above for improved numbers on the ZINC dataset using the PNA architecture.
>
> **Q2:** "*I think it is worth explicitly stating that your theoretical results give upper bounds on the expressive power of many existing methods, say at the very end of Section 3. This is a nice result!*"
>
> **A2:** We agree. We will make this more explicit in the final version.
>
> **Q3:** "*At the end of the related work (line 84), it would be better to say that the above works mostly "do not study the approaches" expressive power beyond 2-WL. For instance, some works like the reconstruction GNN (Cotta et al, 2021) demonstrate ability to compute functions on graphs like cycles, that 2-WL cannot count (beyond a certain size). Also, the edge deletion ESAN (Bevilacqua et al. 2022) goes past 2-WL on strongly regular graphs.*"
>
> **A3:** We will slightly refine our statement here.
>
> We note here that we use the folklore or non-oblivious variant of the $k$-WL, which is equal to the non-folklore or oblivious $(k+1)$-WL, see https://arxiv.org/abs/2104.14624 for details.
>
> Cotta et al. showed that deleting two vertices can distinguish certain strongly regular graphs. Hence their method is incomparable to folklore $2$-WL, which is in line with our results.
>
> (Bevilacqua et al. 2022) uses the non-folklore version, however, their results also align with our results.
>
> However, no paper studies their methods abilities with regard to the folklore 3-WL or beyond.
>
> **Q4:** "*Could you clarify, what is the $O(n+ k \log k )$ algorithm for computing the MAP of the subgraph sampling distribution for ordered subgraphs?*"
>
> **A4:** First, a selection algorithm (https://en.m.wikipedia.org/wiki/Selection_algorithm) is used to find the $k$th largest element $E$ in the list of weights. This takes $O(n)$ time, e.g., using Quickselect. Now, go through the list again and select all entries larger or equal to $E$ in $O(n)$. Finally, sort the $k$ values in $k \log k$. We will make this clear in the final version.
>
> **Q5:** "*It's worth mentioning the diversity loss (Appendix C) used in I-MLE sampling in the main paper.*"
>
> **A5:** Good point; we will incorporate a  description of the loss into "Neural architectures and experimental protocol" of Section 5.
>
> **Q6:** "*The MAE numbers reported here for Alchemy and QM9 appear to be in different units than that of other work. In particular, these are much larger. Can you clarify how to compare these to that of other works?*"
>
> **A6:** Some works, e.g., Morris 2020 and 2022, report normalized test scores (with regard to mean and standard deviations). We report re-normalized test scores.
>
> **Q7:** "*Unless I missed some things, this work does not discuss limitations much at all in the main paper.*"
>
> **A7:** This is due to space constraints, we will add a section "Shortcoming and possible roadmaps" to the end of the final paper, discussing shortcomings and possible future work.

---

> > ### Comment · Reviewer_n3h8 · 2022-08-07
> > **Reply to the authors**
> >
> > We thank the authors for the quick and thorough response to the reviews. These discussed clarifications will make the paper more clear. Also, the added experiments are a plus as ablations and for understanding the properties of various proposed aspects of the models. I think this work is a great contribution, and I suggest acceptance.

---

### Official Review · Reviewer_XNdm · 2022-07-07

**Rating:** 7
**Confidence:** 4
**Soundness:** 3 good
**Presentation:** 2 fair
**Contribution:** 3 good

**Summary:**

The paper proposed a subgraph based GNN approach, k-OSAN. The authors showed that increasing subgraph size always increases the expressive power, and k-OSAN is strictly less powerful than the (k + 1)-WL while being incomparable to the k-WL. To circumvent random or heuristic subgraph selection, this paper devised a data-driven variant of k-OSANs. In experiments, the authors verified that data-driven subgraph selection is superior to previously used random sampling in predictive performance. Recently, subgraph GNN is proposed as a promissing approach, and in general, this is a paper of strong theoretical results.

**Questions:**

1. The paper in its current form is hard to follow, and I suggest the authors can first have a high-level overview of the proposed approach and then going into the detailed mathematical formulations.
2. It is proved that k-OSAN has a higher expressive power, can the authors provide some empirical results? even on some synthetic data.
Minors: Please define t(g) at line 154

**Limitations:**

 The authors have adequately addressed the limitations and potential negative societal impact of their work.

**Strengths And Weaknesses:**

Strength: A new subgraph GNN model with theoretical expressive power guarantee.
Weakness: Though this paper has strong theoretical results, the experiments is kind of weak. The auhors only showed the data driven approach is better than the random approach, but it is not clear how the proposed subgraph GNN compares with other GNN implementations, given the introduced addtional complexity of proposed method.

---

> ### Author Response · Authors · 2022-08-02
> **Replies to Reviewer XNdm**
>
> We thank the reviewer for the fair and constructive review.
>
> In the following, we address the reviewer's questions and remarks.
>
> **Q1:** "*The authors only showed the data-driven approach is better than the random approach, but it is not clear how the proposed subgraph GNN compares with other GNN implementations, given the introduced addtional complexity of the proposed method.*"
>
> **A1:** For example, see Table 1 in our comment above for improved numbers on the ZINC dataset using the PNA architecture.
>
> **Q2:** "*The paper in its current form is hard to follow, and I suggest the authors can first have a high-level overview of the proposed approach and then going into the detailed mathematical formulations.*"
>
> **A2:** Due to the limited space, some paper parts are quite dense. Since the final version allows for an additional page, we will use it to convey better a high-level overview of the theoretical results, also intuitively clear and approachable to the wider GNN and ML community.
>
> **Q3:** "*It is proved that $k$-OSAN has a higher expressive power, can the authors provide some empirical results? even on some synthetic data.*"
>
> **A3:** We note that $k$-OSANs provably encompass existing architectures for which the increase in power has already been demonstrated. Moreover, also the experimental results in Table 1 in our comment above, show that $k$-OSANs expressivity translates into real-world predictive performance boosts.

---

### Official Review · Reviewer_AFUc · 2022-07-08

**Rating:** 7
**Confidence:** 5
**Soundness:** 3 good
**Presentation:** 2 fair
**Contribution:** 4 excellent

**Summary:**

This paper provides a unifying framework for a variety of Graph Neural Networks (GNNs) that have been recently introduced and are based on a common principle: providing to the network subgraph information in various forms. The authors show that most of the existing works can be expressed via the so called Ordered Subgraph Aggregation Network (OSAN) framework, where one runs a GNN (or a 1-WL algorithm) over a marked/anchored version of the initial graph. In particular, first a k-vertex tuple is selected as anchor, which in turn induces a k-ordered subgraph. Secondly, each vertex in the graph is initially coloured by computing the atom type (similar to isomorphism type) of the subgraph induced by the vertex along with the k-tuple anchor - this is contrary to classical WL where all vertices initially have the same colour. Then a conventional GNN (or 1-WL) is run over the graph. The processes is repeated for all possible k-tuple anchors and finally the results are aggregated into a final representation.

Using this general framework, the authors prove that k-OSANs form an hierarchy and, by using some specific graph constructions, they have been able to contrast this hierarchy with the corresponding k-WL. In particular, they show that k-OSANs are incomparable with k-WL, but strictly less expressive than k+1-WL. They further proceed by showing how k-OSANs can simulate popular subgraph enhanced GNN architectures, and discuss the potential consequences of their theory in characterising the expressive power of these specific architectures. Finally, given the high computational complexity of k-OSANs, they propose a practical implementation, where only a handful of anchor subgraphs is used, and their selection is done in a data-driven manner by sampling from a learnable subgraph distribution using the perturb-and-MAP framework. The proposed algorithm is evaluated experimentally, where it is shown in several datasets that data-driven sampling can provide benefits against sampling uniformly at random, while providing improvements against non subgraph-enhanced GNNs.

**Questions:**

- Since the “upstream model” is a GNN, I can imagine that the end-to-end model will inherit its expressivity issues. Although I understand that the downstream model will do all the heavy-lifting, I am wondering how important is the upstream  model, i.e., what kind of subgraphs does it select? A discussion here would be useful.
- Regarding the terminology isomorphism type and atomic type: In several papers on k-WL and equivalent higher-order GNNs the term isomorphism type is used to refer to order-preserving bijections between k-tuples (e.g., Chen et al., NeurIPS’20). Perhaps using the term isomorphism class to refer to the equivalence class induced by isomorphism might be more appropriate? Also, I am a bit confused by the atomic type definition: this is not order preserving, right?
- L260: I found the explanation of the ordered subgraph encoding a bit confusing. Could the authors elaborate more on what they mean by position/rank?
- L331: “In addition…ease of implementation”: I am not really sure what this sentence refers to. What are the “initial features for the subgraphs”?

### Minor:
- L179: perhaps use a different symbol for the two different embedding functions (UPD) to avoid confusion.
Why does the selection vector $\pi_{v, \mathbf{g}}$ depend also on the vertex and not only on the subgraph? I noticed that further down in the text this is instatiated with a subgraph only dependence.
- Section F.2 SM - id GNNs: It is mentioned that id GNNs can count cycles and this is consistent with the 2-WL expressivity upper bound. Probably this is just a subtle detail, but 2-WL can count cycles up to 7 vertices - perhaps a clarification is needed here.


**Limitations:**

- A more elaborate discussion on the limitations of the data-driven subgraph sampling and on the choice of the subgraph parameters would be helpful (see above).
- No immediate negative societal impact

**Strengths And Weaknesses:**

To begin with, I believe that the present manuscript provides an important contribution to the GNN community and will be of interest to various readers working with GNNs and exploring their expressive power, as well as to readers working on graph theory and graph isomorphism. However, I found the presentation unsatisfactory - in my opinion the paper is hard to follow (densely written and notation heavy) and has several clarity issues, which unfortunately now might limit its impact to a small audience. Moreover, the experimental section could be improved, mainly to make the data-driven subgraph sampling more convincing. In detail:

# Strengths
- Subgraph-enhanced GNNs have become one of the most popular families of GNN architectures in the last couple of years. However, many of these innovations have been proposed almost in parallel and this has created a confusing landscape of methods and it was unclear how to compare them. What I liked the most about this paper is that it brings order to this spectrum of methods, paving the path for future work within this framework.
- Secondly, the theoretical results are general and intuitive (please see below for my reservations) and clarify the limitations of subgraph-enhanced GNNs.
- A very appealing aspect of the method is that it relies on 1-WL, which implies that vertex representations are the ones that get updated instead of k-tuple representations.
- Subgraph sampling has been also a long standing question for these and related architectures (such as the ones that use subgraph features, e.g., GSN, Bouritsas et al., TPAMI'22 and F-MPNNs, Barcelo et al., NeurIPS'21), therefore proposing a learnable algorithm is definitely towards the right direction (also see below for some reservations).

# Weaknesses
## 1. Exposition and Clarity
### Theory
- The exposition of the theoretical results needs to be improved. Right now, the authors do not provide any intuition or proof sketches in the main text, while the proofs are complicated (in some cases based on intricate graph constructions along the lines of the CFI gadgets), which will make it very hard for future readers to obtain an actual understanding of the presented theory. For example:
   - Proposition 2 and Theorem 3 are important results because they settle an open question regarding the comparison of k-WL with subgraph-enhanced GNNs, but unfortunately are poorly explained, while the proofs in the supplementary material are confusing in certain parts. Moreover, there are some small inconsistencies between the main paper and the proofs, e.g. Theorem 3 says that k+1-OSANs can distinguish non-isomorphic graphs that k-OSANs deem isomorphic, while the proof in the supplementary (Theorem 10) claims a strict inclusion, which is a stronger result. Also, the proof is very brief and points to a theorem in a different paper without adding sufficient explanations. Could the authors provide a more extensive explanation in the rebuttal so as to be able to validate this claim? Also, the chain of arguments to prove these two claims in the supplementary is tangled. Could the authors clearly explain the chain of arguments in the rebuttal and in the main paper?
   - I am a bit confused by section E.3. supplementary material. How are these results different from section D? In the title the authors mention a comparison between the vertex-subgraph k-OSWL variant and k-WL, yet in the proofs below they mention k-OSWL. Moreover, I think I am missing something in Lemma 23 (which perhaps I am also missing in section D): you mention that k-OSWL is strictly less expressive than (k+1)-WL and that (k+1)-OSWL distinguishes graphs that (k+1)-WL deems isomorphic. However, I don’t see why this immediately implies that k-OSWL is strictly less expressive than (k+1)-OSWL, since there might be graphs that k-OSWL distinguishes, therefore also (k+1)-WL, but are deemed isomoprhic by (k+1)-OSWL.

### Gradient computation
- The presentation of the I-MLE is also quite elusive. I believe the authors should provide clarifications and improve a lot the exposition in this part, first because this is probably going to be knew to the interested readers from the GNN community, and second because data-driven subgraph sampling is a novel contribution that can be potentially impactful, but currently doesn’t do its complete justice.
- Furthermore, it is not immediately clear to me why the authors chose this specific parametrisation for the subgraph distribution. Another potential and simpler choice could have been to iteratively sample without replacement k vertices M times (one for each subgraph) and then estimate the gradients with a variant of the REINFORCE algorithm (as in Bouritsas et al., NeurIPS'21). Another idea that comes to my mind is the Plackett-Luce distribution (M times, one for each subgraph), as in the NeuralSort (Grover et al., ICLR'19) or more recent ones. Note that this can be made deterministic (essentialy sampling the top-k vertices with the highest scores) and therefore the overall pipeline can become permutation invariant, contrary to the current version, which is permutation sensitive due to sampling. Is there any particular advantage of the algorithm chosen?

## 2. Experiments:
I believe that the experimental section should be extended and improved in some respects:
- First: it would have been useful to include in your tables more baselines, and especially baselines falling under the k-OSAN framework (e.g., as with ESAN). The results in Table 2b raise some concerns that perhaps the data-driven subgraph sampling is not yet at a level that can be reliably used in practice, so more extensive comparisons and/or an explanatory discussion with the potential reasons for this and remedies, would help. Also, note that the performance of the current baseline is not immediately visible in the tables.
- Second: even though subgraph sampling is learnable, the authors still have to make several design choices that seem arbitrary, i.e., the number of subgraphs to sample and the number of vertices per subgraph. Can the authors discuss any potential strategies to circumvent ad-hoc selections? An ablation study in a controlled setup, showing how the performance changes when increasing the number of subgraphs and the number of vertices would have been very helpful.
- Another potential interesting experiment would be to select a more informative baseline, than the random one, for subgraph sampling, e.g., according to certain subgraph properties that either (1) are potentially related to the task at hand, or (2) can break graph symmetries and make the vertices more identifiable.
- I found it a bit puzzling that the authors experimentally used unordered subgraphs, although all the theoretical analysis refers to ordered subgraphs. If I am not mistaken, unordered subgraphs are sufficient to simulate the architectures from previous works, but I am wondering if the theoretical results comparing k-OSANs to k-WL also hold for unordered subgraphs.
- Subgraph sampling is not permutation equivariant w.r.t. the subgraphs. The authors mention in appendix C that they use an auxiliary loss to prevent degenerate solutions, but I believe more emphasis should be given to this and this part should be ablated.
- The tables are not quite self-explanatory. What are the # and #Subg. Columns? Also, I think it is going to be helpful to explain how the operation used is implemented within the k-OSAN framework (I understand that vertex deletion means that the anchor subgraph has n-1 vertices, while vertex selection means a 1-vertex subgraph, is that correct?).

---

> ### Author Response · Authors · 2022-08-02
> **Replies to reviewer AFUc**
>
> We thank the reviewer for the detailed, careful, and constructive review.
>
> **Q1:** "*The exposition of the theoretical results needs to be improved. Right now, the authors do not provide any intuition or proof sketches in the main text, while the proofs are complicated (in some cases based on intricate graph constructions along the lines of the CFI gadgets), which will make it very hard for future readers to obtain an actual understanding of the presented theory.*"
>
> **A1:** Due to the limited space, some paper parts are quite dense. Since the final version allows for an additional page, we will use it to convey better a high-level overview of the theoretical results, also intuitively clear and approachable to the wider GNN and ML community. We will also add an intuitive description before each of the proofs in the appendix.
>
> **Q2:** "*Proposition 2 and Theorem 3 are important results because they settle an open question regarding the comparison of k-WL with subgraph-enhanced GNNs, but unfortunately are poorly explained, while the proofs in the supplementary material are confusing in certain parts.*"
>
> **A2:** We will more carefully structure the line of reasoning in the appendix, guiding the reader more carefully, in the final version.
>
> **Q3:** "*Theorem 3 says that $k+1$-OSANs can distinguish non-isomorphic graphs that $k$-OSANs deem isomorphic, while the proof in the supplementary (Theorem 10) claims a strict inclusion, which is a stronger result. Also, the proof is very brief and points to a theorem in a different paper without adding sufficient explanations. Could the authors provide a more extensive explanation in the rebuttal so as to be able to validate this claim?*"
>
> **A3:** Good catch, we indeed prove the strict inclusion. We will state Theorem 3 more precisely, making the strict inclusion clear.
>
> The proof of Theorem 10 is indeed rather brief. We here provide a more detailed explanation. In particular,  from Proposition 1 we know that we can equivalently show that $(k+1)$-OSWL is strictly more powerful than $k$-OSWL. This in turn follows from the following:
>
> 1. $(k+1)$-OSWL is at least as powerful as $k$-OSWL.
> (This is implicit in Theorem 10 in the appendix; we will expand the proof of this inclusion in our final version as per the reviewer's remarks. See also below for the proof.)
>
> 2. $(k+1)$-WL is at least as powerful as $k$-OSWL. (From Proposition 1 $k$-OSWL and $k$-OSAN are equivalent, and $k$-OSAN can be easily seen to be $k$-MPNNs. These $k$-MPNNs are defined in Geerts & Reutter 2022 in terms of update and aggregation operations on $(k+1)$-vertex embeddings, which is precisely the machinery needed to define $k$-OSANs.  Finally, $k$-MPNNs are bounded by $(k+1)$-WL in expressive power by Geerts & Reutter 2022).
>
> 3. $k$-WL cannot distinguish CFI graphs while $k$-OSWL can. (Lemma 15 in the paper).
>
> Hence, (3) applied for $(k+1)$-WL implies that there exist graphs than can be distinguished by $(k+1)$-OSWL but not by $(k+1)$-WL. Then, because of (2), these graphs cannot be distinguished by $k$-OSWL either. Combined with (1), we get that $(k+1)$-OSWL is strictly more powerful than $k$-OSWL. Proposition 1 implies the same for (k+1)-OSAN and $k$-OSAN.
>
> In the appendix,  we further complement (2) by showing that $(k+1)$-WL is strictly more powerful than $k$-OSWL (Lemma 13 + Theorem 14 in the paper). This result may be of independent interest.
>
> Intuitively, $(k+1)$-OSWL is at least as powerful as $k$-OSWL because ordered $k$-subgraphs can be simulated by ordered $(k+1)$-subgraphs.
>
> A more formal argument is based on the following stronger statement: for every $r>0$, the partition of $V(G)$ produced by $r$ rounds of $k$-OSWL is no finer than the partition of $V(G)$ produced by $r$ rounds of $(k+1)$-OSWL.
>
> The proof of this statement is by induction on $r$ by using the following observations. The set $S$ of ordered $k$-subgraphs is in bijective correspondence with the set $T$ of those ordered $(k+1)$-subgraphs which have the $k$th and $k$st labels on the same vertex. This atomic type constraint, i.e., that the last two labeled vertices are identical, ensures that the colors aggregated using ordered $(k+1)$-subgraphs from the set $T$ are always distinct from the colors aggregated using ordered $(k+1)$-subgraphs not in $T$.
> Hence, if $k$-OSWL distinguishes two vertices $u$ and $v$, then $(k+1)$-OSWL will also distinguish the vertices $u$ and $v$, simply
> by looking at the sub-multiset of colors aggregated using the ordered subgraphs from the set $T$.

---

> > ### Author Response · Authors · 2022-08-02
> > **Replies to reviewer AFUc, continued**
> >
> > **Q4:** "*The presentation of the I-MLE is also quite elusive. I believe the authors should provide clarifications and improve a lot the exposition in this part, first because this is probably going to be knew to the interested readers from the GNN community, and second because data-driven subgraph sampling is a novel contribution that can be potentially impactful, but currently doesn’t do its complete justice.*"
> >
> > **A4:** We will add a comprehensive background section on IMLE and gradient estimation in the appendix.
> >
> > **Q5:** "*Furthermore, it is not immediately clear to me why the authors chose this specific parametrization for the subgraph distribution [...]*"
> >
> > **A5:** The advantage of the chosen algorithm is its general purpose nature. In contrast to top-k (SoftSub), sorting (NeuralSort), and other specialized algorithms, IMLE can be used with any graph algorithm including maximum-weight covering subgraph (ensuring all nodes are covered by the subgraphs, we have implemented this) or spanning tree. Hence, in addition to being ordered and of a particular size k additional constraints can be imposed on the selected graphs. Results for these methods can be found in Table 4.
> >
> > While REINFORCE (aka the score function estimator) is also general purpose, it is very difficult to use due to its high variance and even when using control variates. Hence, we decided to use a single low-variance general purpose gradient estimation method for the OSAN framework. However, future work might also investigate specialized gradient estimation techniques in the context of subgraph sampling.
> >
> > We believe that sampling is important as it allows the model to trade off exploration ("trying out new subgraphs") and exploitation during training. Following prior work (ESAN), we also sample at test time. However, we could also take the most probable subgraphs at test time. We found that the sampling strategy works empirically the best.*"
> >
> > **Q6:** "*It would have been useful to include in your tables more baselines, and especially baselines falling under the k-OSAN framework (e.g., as with ESAN). The results in Table 2b raise some concerns that perhaps the data-driven subgraph sampling is not yet at a level that can be reliably used in practice, so more extensive comparisons and/or an explanatory discussion with the potential reasons for this and remedies, would help. Also, note that the performance of the current baseline is not immediately visible in the tables.*"
> >
> > **A6:** We added more baseline results in our tables, e.g., Table 1 above. The main selling point of our method is its efficiency. As seen in Table 2b in the main paper, our model is significantly faster because we sample much fewer subgraphs than ESAN. Note that ESAN keeps a set of all the possible subgraphs in memory, then they sample from the set or use all of the subgraphs, quickly becoming infeasible even for small $k$. If the subgraphs are sampled randomly on the fly, it refers to the _random_ rows of Table 2b. Our method also samples on the fly but in a data-driven manner, and our method outperforms random sampling.
> >
> > **Q7:** "*Even though subgraph sampling is learnable, the authors still have to make several design choices that seem arbitrary, i.e., the number of subgraphs to sample and the number of vertices per subgraph. Can the authors discuss any potential strategies to circumvent ad-hoc selections? [...]*"
> >
> > **A7:** We note here that these hyperparameters also appear when considering naive sampling, e.g., uniform, or considering all subgraphs. That is, they are not specific to our sampling architecture. Finding the right hyperparameters can be achieved with any standard hyperparameter selection algorithm/architecture.
> >
> > We will add a discussion to the final paper, and include a more carefully designed ablation study with regard to the different hyperparameters.
> >
> > **Q8:** "*I found it a bit puzzling that the authors experimentally used unordered subgraphs, although all the theoretical analysis refers to ordered subgraphs. If I am not mistaken, unordered subgraphs are sufficient to simulate the architectures from previous works, but I am wondering if the theoretical results comparing k-OSANs to k-WL also hold for unordered subgraphs.*"
> >
> > **A8:** On request we have added ablation experiments where we use the ordered method for sampling, please see Table 2 and Table 3 in the joint reply. The ordered methods show even better results than the unordered methods.

---

> > > ### Author Response · Authors · 2022-08-02
> > > **Replies to reviewer AFUc, continued**
> > >
> > > **Q9:** "*Subgraph sampling is not permutation equivariant w.r.t. the subgraphs. The authors mention in appendix C that they use an auxiliary loss to prevent degenerate solutions, but I believe more emphasis should be given to this and this part should be ablated.*"
> > >
> > > **A9:** We have done the ablation experiments without auxiliary loss. The description of auxiliary loss and the corresponding results can be found in the joint reply and Table 5. We can see through the table that no auxiliary loss leads to worse performance in almost all the cases.
> > >
> > > **Q10:** "*The tables are not quite self-explanatory. What are the # and #Subg. Columns?*"
> > >
> > > **A10:** # = "number", so number of vertices and number of sampled subgraphs, respectively. We will make this clear in the final version.
> > >
> > > **Q11:** "*Since the “upstream model” is a GNN, I can imagine that the end-to-end model will inherit its expressivity issues. Although I understand that the downstream model will do all the heavy-lifting, I am wondering how important is the upstream model, i.e., what kind of subgraphs does it select? A discussion here would be useful.*"
> > >
> > > **A11:** We agree, and we will add a discussion regarding the expressivity of the "upstream model" and its relationship to the expressivity of the overall architecture in the final version.
> > >
> > > **Q12:** "*Also, I am a bit confused by the atomic type definition: this is not order-preserving, right?*"
> > >
> > > **A12:** The atomic type definition is order-preserving, which is equivalent to considering the isomorphism type of vertex-ordered k-vertex subgraphs, e.g., labeled from 1 to k.
> > >
> > > **Q13:** "*I found the explanation of the ordered subgraph encoding a bit confusing. Could the authors elaborate more on what they mean by position/rank?*"
> > >
> > > **A13:** The rank is the position of the node in the ordered subgraph. When the ordered subgraphs have $k$ nodes, then the rank is the position in this order. Rank = $0$ means not part of the ordered subgraph. Rank = $1$ means the lowest rank in the ordered subgraph, ... Rank=$k$ means the highest rank in the ordered subgraph.
> > >
> > > **Q14:** "*perhaps use a different symbol for the two different embedding functions (UPD) to avoid confusion. Why does the selection vector depend also on the vertex and not only on the subgraph? I noticed that further down in the text this is instantiated with a subgraph only dependence.*"
> > >
> > > **A14:** We agree and make will this more clear in the final version.
> > >
> > > **Q15:** "*Probably this is just a subtle detail, but $2$-WL can count cycles up to 7 vertices - perhaps a clarification is needed here.*"
> > >
> > > **A15:** The restriction of $2$-WL to count up to 7 cycles is for counting subgraph isomorphisms of cycles. Id-GNNs count homomorphism counts of cycles for which there is no restriction on the size of cycles.

---

### Official Review · Reviewer_5AyY · 2022-07-18

**Rating:** 7
**Confidence:** 3
**Soundness:** 3 good
**Presentation:** 3 good
**Contribution:** 3 good

**Summary:**

Graph Representation Learning, and specifically GNNs have generated tremendous over the past few years. In this work, the authors propose a framework, which builds on recent works on subgraph enhanced (aggregated) GNNs to obtain expressive representations.  Here, the authors firstly propose a unifying framework (which ties most recently proposed works), and secondly introduces a way to 'learn to sample' from the subgraphs leveraging techniques which can back-propagate through discrete distributions.

**Questions:**

1. Please provide timings for other datasets as well.
2. Page 4, line 161 - do you mean Union? or both strategies work? What do you see in practice - performance wise?

**Limitations:**

Yes, the authors do talk about computation costs and ways to scale them down in practice.

**Strengths And Weaknesses:**

Strengths:
1. The idea to learn using data to 'select' subgraphs is interesting
2. The paper is largely well written and easy to understand.
3. The propositions and theorems appear valid


Weaknesses:
1. The results are relatively worse on the ZINC dataset than ESAN, and comparison is not performed on other datasets performed in comparable works (e.g. MUTAG, Proteins, PTC, etc). Is there an underlying reason - which I may have missed.
2. Per, section 5 (and appendix F.1), sampling subgraphs - the authors use unordered subgraphs rather than the proposed theoretical strategy. Can you provide results atleast on a sample synthetic result, for the proposed framework?







Post Rebuttal Updates:

Thank you very much for the rebuttal authors -- for adding the  ordered experimental results, justification for the smaller datasets, etc. After reading the response and the other reviews (and corresponding replies), I am happy to increase my scores.


Best,

Reviewer 5AyY

---

> ### Author Response · Authors · 2022-08-02
> **Replies to Reviewer 5AyY**
>
> In the following, we address the reviewer's questions and remarks.
>
> **Q1:** "*The results are relatively worse on the ZINC dataset than ESAN, and comparison is not performed on other datasets performed in comparable works (e.g. MUTAG, Proteins, PTC, etc). Is there an underlying reason - which I may have missed.*"
>
> **A1:** See Table 1 in our comment above for improved numbers on the ZINC dataset using the PNA architecture. We did not use the TUDatasets (e.g., MUTAG, Proteins, PTC, ...) due to their small number of graphs, leading to high standard deviations, making it hard to make meaningful comparisons. Nevertheless, we, see Table 7 above, added results for the PROTEINS dataset, showing that our data-driven approach slightly outperforms the random-sampling approach, while beating the standard baselines.
>
> **Q2:** "*Please provide timings for other datasets as well.*"
>
> **A2:** Good point; we will add timings for all considered datasets in the final version. Unfortunately, the short rebuttal time did not allow us to compute meaningful timings.
>
> **Q3:** "*Page 4, line 161 - do you mean Union? or both strategies work?*"
>
> **A3:** The white box acts as a placeholder for either $N_G(v)$ or $V(G)$. That is, replace the white box with either $N_G(v)$ or $V(G)$, resulting in two different definitions of $C_{i+1}$.

---

> > ### Comment · Reviewer_5AyY · 2022-08-05
> > **Response to authors**
> >
> > Post Rebuttal Updates:
> >
> > Thank you very much for the rebuttal authors -- for adding the ordered experimental results, justification for the smaller datasets, etc. After reading the response and the other reviews (and corresponding replies), I am happy to increase my scores.
> >
> > Best,
> >
> > Reviewer 5AyY

---

### Author Response · Authors · 2022-08-02
**Joint reply: Improved experimental results.**

In response to the reviewers, we added additional experiments and present them here in a single answer. We further reference the listed tables throughout the individual responses; see below.

Unless specifically pointed out, the training hyperparameters are identical to the ones in the appendix, i.e., we did not tune them.

# New base model

We added a new base model, PNA [6], for the ZINC dataset. We use the official implementation of PyG for the ZINC dataset. The training hyperparameters are the same as those used for the GINE model; see the appendix. The table below compares the MAE error between our method, random sampling, and additional baselines. As shown in the table, our method outperforms the baselines and random sampling methods.

## ZINC with PNA model

|     Method      | Operation | Type   | #   | # Subgraph | MAE &darr;       |
|:---------------:|-----------|--------|-----|:----------:|------------------|
|     PNA [6]     | -         | -      | -   |     -      | 0.188 &pm; 0.004 |
|     GIN [1]     | -         | -      | -   |     -      | 0.252 &pm; 0.017 |
|     DGN [7]     | -         | -      | -   |     -      | 0.168 &pm; 0.003 |
|     Random      | Delete    | Vertex | 1   |     3      | 0.260 &pm; 0.001 |
| I-MLE unordered | Delete    | Vertex | 1   |     3      | 0.168 &pm; 0.005 |
|     Random      | Delete    | Vertex | 3   |     3      | 0.226 &pm; 0.007 |
| I-MLE unordered | Delete    | Vertex | 3   |     3      | 0.172 &pm; 0.001 |
|     Random      | Delete    | Edge   | 3   |     3      | 0.180 &pm; 0.007 |
| I-MLE unordered | Delete    | Edge   | 3   |     3      | 0.159 &pm; 0.008 |
|     Random      | Delete    | Edge   | 10  |     3      | 0.174 &pm; 0.009 |
| I-MLE unordered | Delete    | Edge   | 10  |     3      | 0.161 &pm; 0.003 |
|     Random      | Delete    | 1-Ego  | -   |     3      | 0.325 &pm; 0.001 |
| I-MLE unordered | Delete    | 1-Ego  | -   |     3      | 0.167 &pm; 0.005 |

__Table 1__

# Ordered/unordered methods

We conducted additional experiments for the ordered subgraph setting, aligning with the theory in the paper, as reviewers `5AyY` and `AFUc` suggested.

For the ordered subgraph setting, we sorted the nodes according to the weights given by the upstream model in descending order and obtained a re-ordered graph, isomorphic to the original one, and take the (re-ordered) adjacency matrix as an extra feature. In the downstream model, we embed these extra features together with (reordered) node features. Different from the unordered method, we aggregated the node embeddings in different subgraphs first, then aggregated the various nodes into a graph embedding.

See below for a comparison of ordered and unordered subgraph aggregation methods, together with random sampling. In both datasets, the ordered method shows on-par or even better performance than the unordered counterpart.

## ogbg-molesol

|     Method      | Operation | Type   | #   | # Subgraph | MSE    &darr;    |
|:---------------:|-----------|--------|-----|:----------:|------------------|
|   No sampling   | -         | -      | -   |     -      | 1.193 &pm; 0.083 |
|     Random      | Delete    | Vertex | 1   |     3      | 1.215 &pm; 0.095 |
| I-MLE unordered | Delete    | Vertex | 1   |     3      | 1.053 &pm; 0.080 |
|  I-MLE ordered  | Delete    | Vertex | 1   |     3      | 0.835 &pm; 0.079 |
|     Random      | Delete    | Vertex | 2   |     3      | 1.132 &pm; 0.020 |
| I-MLE unordered | Delete    | Vertex | 2   |     3      | 1.081 &pm; 0.021 |
|  I-MLE ordered  | Delete    | Vertex | 2   |     3      | 0.850 &pm; 0.106 |
|     Random      | Delete    | Vertex | 5   |     3      | 0.992 &pm; 0.115 |
| I-MLE unordered | Delete    | Vertex | 5   |     3      | 1.115 &pm; 0.076 |
|  I-MLE ordered  | Delete    | Vertex | 5   |     3      | 0.853 &pm; 0.043 |

__Table 2__

## ZINC

|     Method      | Operation | Type   | #   | # Subgraph | MAE &darr;       |
|:---------------:|-----------|--------|-----|:----------:|------------------|
|   No sampling   | -         | -      | -   |     -      | 0.207 &pm; 0.006 |
|     Random      | Delete    | Vertex | 1   |     3      | 0.283 &pm; 0.003 |
| I-MLE unordered | Delete    | Vertex | 1   |     3      | 0.194 &pm; 0.007 |
|  I-MLE ordered  | Delete    | Vertex | 1   |     3      | 0.187 &pm; 0.004 |

__Table 3__

---

> ### Author Response · Authors · 2022-08-02
> **continued**
>
> # Arbitrary subgraph selection methods
>
> As suggested by reviewer `AFUc`, we added some more sophisticated subgraph selection methods.
>
> For node selection, we leveraged the Google OR package and solved a combinatorial optimization problem. The objective function was to maximize the sum of weights of the nodes selected. The constraints were chosen so that the nodes of a given graph are covered, i.e., each node is contained in at least one subgraph.
>
> For the edge selection method, we searched for a maximum spanning tree (MST) with the weights given by the upstream model. In comparison, the random sampling method assigns identical weights to each edge and searches for a random spanning tree.
>
> ## ogbg-molesol
>
> |     Method      | Operation | Type   | #   | # Subgraph | MSE &darr;       |
> |:---------------:|-----------|--------|-----|:----------:|------------------|
> |   No sampling   | -         | -      | -   |     -      | 1.193 &pm; 0.083 |
> |     Random      | Delete    | Vertex | 1   |     3      | 1.215 &pm; 0.095 |
> | I-MLE unordered | Delete    | Vertex | 1   |     3      | 1.053 &pm; 0.080 |
> |  I-MLE covered  | Delete    | Vertex | 1   |     3      | 1.074 &pm; 0.115 |
> |     Random      | Delete    | Vertex | 2   |     3      | 1.132 &pm; 0.020 |
> | I-MLE unordered | Delete    | Vertex | 2   |     3      | 1.081 &pm; 0.021 |
> |  I-MLE covered  | Delete    | Vertex | 2   |     3      | 1.081 &pm; 0.068 |
> |     Random      | Delete    | Vertex | 5   |     3      | 0.992 &pm; 0.115 |
> | I-MLE unordered | Delete    | Vertex | 5   |     3      | 1.115 &pm; 0.076 |
> |  I-MLE covered  | Delete    | Vertex | 5   |     3      | 0.946 &pm; 0.058 |
> |   Random MST    | Delete    | Edge   | -   |     3      | 1.095 &pm; 0.021 |
> |    I-MLE MST    | Delete    | Edge   | -   |     3      | 1.070 &pm; 0.005 |
>
> __Table 4__
>
> # Auxiliary loss ablation
>
> In response to reviewer `AFUc`, we conducted some ablation studies without the auxiliary loss.
>
> __Note that the auxiliary loss is improved and different from that in the paper.__ We define the auxiliary as follows: consider a set of subgraphs, we use a mask matrix $M \in \{0, 1\} ^ {n \times k}$ to represent the nodes selected or unselected, where $n$ is the number of nodes in the original graph, and $k$ the number of subgraphs. We would like the subgraphs to differ from each other, therefore we take the sum $\mathbf{s} = \sum_{j=1}^{k} M_{i,j} \in \mathbb{N}^{n}$, and calculate a KL divergence of the normalized vector $\hat{\mathbf{s}}$ and the normalized ones vector $\hat{1}$ as $KL = \sum_{i=1}^{n} \dfrac{1}{n} \log \dfrac{\frac{1}{n}}{\hat{\mathbf{s}}_i}$ as the auxiliary loss.
>
> The following ablation study shows that this auxiliary loss actually helps in almost all cases.
>
> ## ogbg-molesol
>
> |     Method      | Operation | Type   | #   | # Subgraph | MSE &darr;       |
> |:---------------:|-----------|--------|-----|:----------:|------------------|
> |   No sampling   | -         | -      | -   |     -      | 1.193 &pm; 0.083 |
> |     Random      | Delete    | Vertex | 1   |     3      | 1.215 &pm; 0.095 |
> | I-MLE unordered | Delete    | Vertex | 1   |     3      | 1.053 &pm; 0.080 |
> | I-MLE ablation  | Delete    | Vertex | 1   |     3      | 1.120 &pm; 0.092 |
> |     Random      | Delete    | Vertex | 2   |     3      | 1.132 &pm; 0.020 |
> | I-MLE unordered | Delete    | Vertex | 2   |     3      | 1.081 &pm; 0.021 |
> | I-MLE ablation  | Delete    | Vertex | 2   |     3      | 1.137 &pm; 0.146 |
> |     Random      | Delete    | Vertex | 5   |     3      | 0.992 &pm; 0.115 |
> | I-MLE unordered | Delete    | Vertex | 5   |     3      | 1.115 &pm; 0.076 |
> | I-MLE ablation  | Delete    | Vertex | 5   |     3      | 1.247 &pm; 0.126 |
>
> __Table 5__
>
> ## ZINC
> |     Method      | Operation | Type   | #   | # Subgraph | MAE &darr;       |
> |:---------------:|-----------|--------|-----|:----------:|------------------|
> |   No sampling   | -         | -      | -   |     -      | 0.207 &pm; 0.006 |
> |     Random      | Delete    | Vertex | 1   |     3      | 0.283 &pm; 0.003 |
> | I-MLE unordered | Delete    | Vertex | 1   |     3      | 0.194 &pm; 0.007 |
> | I-MLE ablation  | Delete    | Vertex | 1   |     3      | 0.194 &pm; 0.004 |
> |     Random      | Delete    | Vertex | 3   |     3      | 0.265 &pm; 0.003 |
> | I-MLE unordered | Delete    | Vertex | 3   |     3      | 0.184 &pm; 0.006 |
> | I-MLE ablation  | Delete    | Vertex | 3   |     3      | 0.184 &pm; 0.004 |
> |     Random      | Delete    | Edge   | 3   |     3      | 0.192 &pm; 0.002 |
> | I-MLE unordered | Delete    | Edge   | 3   |     3      | 0.176 &pm; 0.006 |
> | I-MLE ablation  | Delete    | Edge   | 3   |     3      | 0.178 &pm; 0.008 |
> |     Random      | Delete    | Edge   | 10  |     10     | 0.169 &pm; 0.013 |
> | I-MLE unordered | Delete    | Edge   | 10  |     10     | 0.155 &pm; 0.004 |
> | I-MLE ablation  | Delete    | Edge   | 10  |     10     | 0.162 &pm; 0.001 |
>
> __Table 6__

---

> > ### Author Response · Authors · 2022-08-02
> > **continued**
> >
> > # New dataset
> >
> > We added a new dataset PROTEINS as required, and show comparisons with variants of our method and random sampling. We use a GIN model, whose architecture and hyperparameters follow ESAN [8]. And we split the dataset with 80% training set, 10% validation set, and 10% test set, the same as DGL implementation for PROTEINS.
> >
> > ## PROTEINS
> >
> > |      Method      | Operation | Type   | #   | # Subgraph | Accuracy &uarr;    |
> > |:----------------:|-----------|--------|-----|:----------:|------------------|
> > |     GIN [1]      | -         | -      | -   |     -      | 0.762 &pm; 0.028 |
> > | GIN + ID-GNN [2] | -         | -      | -   |     -      | 0.754 &pm; 0.027 |
> > |   DropEdge [3]   | -         | -      | -   |     -      | 0.735 &pm; 0.045 |
> > |     CIN [4]      | -         | -      | -   |     -      | 0.770 &pm; 0.043 |
> > |     PPGN [5]     | -         | -      | -   |     -      | 0.772 &pm; 0.047 |
> > |   No sampling    | -         | -      | -   |     -      | 0.775 &pm; 0.034 |
> > |      Random      | Delete    | Vertex | 1   |     3      | 0.760 &pm; 0.011 |
> > | I-MLE unordered  | Delete    | Vertex | 1   |     3      | 0.775 &pm; 0.014 |
> > |      Random      | Delete    | Vertex | 3   |     3      | 0.769 &pm; 0.019 |
> > | I-MLE unordered  | Delete    | Vertex | 3   |     3      | 0.783 &pm; 0.012 |
> > |      Random      | Delete    | Edge   | 3   |     3      | 0.764 &pm; 0.024 |
> > | I-MLE unordered  | Delete    | Edge   | 3   |     3      | 0.780 &pm; 0.013 |
> >
> > __Table 7__
> >
> > # References
> >
> > [1] Xu, Keyulu, et al. "How powerful are graph neural networks?." arXiv preprint arXiv:1810.00826 (2018).
> >
> > [2] You, Jiaxuan, et al. "Identity-aware graph neural networks." Proceedings of the AAAI Conference on Artificial Intelligence. Vol. 35. No. 12. 2021.
> >
> > [3] Rong, Yu, et al. "Dropedge: Towards deep graph convolutional networks on node classification." arXiv preprint arXiv:1907.10903 (2019).
> >
> > [4] Bodnar, Cristian, et al. "Weisfeiler and Lehman go cellular: CW networks." Advances in Neural Information Processing Systems 34 (2021): 2625-2640.
> >
> > [5] Maron, Haggai, et al. "Provably powerful graph networks." Advances in neural information processing systems 32 (2019).
> >
> > [6] Corso, Gabriele, et al. "Principal neighbourhood aggregation for graph nets." Advances in Neural Information Processing Systems 33 (2020): 13260-13271.
> >
> > [7] Beaini, Dominique, et al. "Directional graph networks." International Conference on Machine Learning. PMLR, 2021.
> >
> > [8] Bevilacqua, Beatrice, et al. "Equivariant subgraph aggregation networks." arXiv preprint arXiv:2110.02910 (2021).

---

### Meta-Review · Area_Chair_FdYb · 2022-08-23

**Recommendation:** Accept
**Confidence:** Certain

**Metareview:**

The paper considered subgraph-enhanced graph neural networks that provably boost the expressive power of standard message-passing graph neural networks. It proposed a theoretical framework for studying the expressive power of these GNNs and the relation to the Weisfeiler-Leman hierarchy and extended the existing results. It then addressed the limitation of subgraph sampling in existing methods, exploring different sampling approaches using state-of-the-art data-driven methods. Empirical results showed data-driven architectures increase prediction accuracy and reduce computation time.

The work has made novel and solid contributions: theoretical analysis of the expressive power of subgraph-enhanced GNNs, data-driven approaches for sampling subgraphs, and strong performance of the proposed approaches. The authors have addressed well the comments by the reviewers during the response period and strengthened the work.


**Award:**

No

---

### Decision · Program_Chairs · 2022-09-14

Accept